# What Makes Data Suitable for a Locally Connected Neural Network? A Necessary and Sufficient Condition Based on Quantum Entanglement

**Yotam Alexander**[*], **Nimrod De La Vega**[*], **Noam Razin**, **Nadav Cohen**

Tel Aviv University

{yotama,nimrodd,noamrazin}@mail.tau.ac.il, cohennadav@cs.tau.ac.il

## Abstract

The question of what makes a data distribution suitable for deep learning is a fundamental open problem. Focusing on locally connected neural networks (a prevalent family of architectures that includes convolutional and recurrent neural networks as well as local self-attention models), we address this problem by adopting theoretical tools from quantum physics. Our main theoretical result states that a certain locally connected neural network is capable of accurate prediction over a data distribution *if and only if* the data distribution admits low quantum entanglement under certain canonical partitions of features. As a practical application of this result, we derive a preprocessing method for enhancing the suitability of a data distribution to locally connected neural networks. Experiments with widespread models over various datasets demonstrate our findings. We hope that our use of quantum entanglement will encourage further adoption of tools from physics for formally reasoning about the relation between deep learning and real-world data.[1]

## 1 Introduction

Deep learning is delivering unprecedented performance when applied to data modalities involving images, text and audio. On the other hand, it is known both theoretically and empirically [53, 1] that there exist data distributions over which deep learning utterly fails. The question of *what makes a data distribution suitable for deep learning* is a fundamental open problem in the field.

A prevalent family of deep learning architectures is that of *locally connected neural networks*. It includes, among others: *(i)* convolutional neural networks, which dominate the area of computer vision; *(ii)* recurrent neural networks, which were the most common architecture for sequence (*e.g.* text and audio) processing, and are experiencing a resurgence by virtue of S4 models [26]; and *(iii)* local variants of self-attention neural networks [46]. Conventional wisdom postulates that data distributions suitable for locally connected neural networks are those exhibiting a "local nature," and there have been attempts to formalize this intuition [66, 28, 15]. However, to the best of our knowledge, there are no characterizations providing necessary and sufficient conditions for a data distribution to be suitable to a locally connected neural network.

A seemingly distinct scientific discipline tying distributions and computational models is *quantum physics*. There, distributions of interest are described by *tensors*, and the associated computational models are *tensor networks*. While there is shortage in formal tools for assessing the suitability of data distributions to deep learning architectures, there exists a widely accepted theory that allows for assessing the suitability of tensors to tensor networks. The theory is based on the notion of *quantum*

---

[*]Equal contribution

[1]Due to lack of space, a significant portion of the paper is deferred to the appendices. We refer the reader to [2] for a self-contained version of the text.

37th Conference on Neural Information Processing Systems (NeurIPS 2023).

*entanglement*, which quantifies dependencies that a tensor admits under partitions of its axes (for a given tensor $\mathcal{A}$ and a partition of its axes to sets $\mathcal{K}$ and $\mathcal{K}^c$, the entanglement is a non-negative number quantifying the dependence that $\mathcal{A}$ admits between $\mathcal{K}$ and $\mathcal{K}^c$).

In this paper, we apply the foregoing theory to a tensor network equivalent to a certain locally connected neural network, and derive theorems by which fitting a tensor is possible if and only if the tensor admits low entanglement under certain *canonical partitions* of its axes. We then consider the tensor network in a machine learning context, and find that its ability to attain low approximation error, *i.e.* to express a solution with low population loss, is determined by its ability to fit a particular tensor defined by the data distribution, whose axes correspond to features. Combining the latter finding with the former theorems, we conclude that a *locally connected neural network is capable of accurate prediction over a data distribution if and only if the data distribution admits low entanglement under canonical partitions of features*. Experiments with different datasets corroborate this conclusion, showing that the accuracy of common locally connected neural networks (including modern convolutional, recurrent, and local self-attention neural networks) is inversely correlated to the entanglement under canonical partitions of features in the data (the lower the entanglement, the higher the accuracy, and vice versa).

The above results bring forth a recipe for enhancing the suitability of a data distribution to locally connected neural networks: given a dataset, search for an arrangement of features which leads to low entanglement under canonical partitions, and then arrange the features accordingly. Unfortunately, the above search is computationally prohibitive. However, if we employ a certain correlation-based measure as a surrogate for entanglement, *i.e.* as a gauge for dependence between sides of a partition of features, then the search converts into a succession of *minimum balanced cut* problems, thereby admitting use of well-established graph theoretical tools, including ones designed for large scale [29, 57]. We empirically evaluate this approach on various datasets, demonstrating that it substantially improves prediction accuracy of common locally connected neural networks (including modern convolutional, recurrent, and local self-attention neural networks).

The data modalities to which deep learning is most commonly applied — namely ones involving images, text and audio — are often regarded as natural (as opposed to, for example, tabular data fusing heterogeneous information). We believe the difficulty in explaining the suitability of such modalities to deep learning may be due to a shortage in tools for formally reasoning about natural data. Concepts and tools from physics — a branch of science concerned with formally reasoning about natural phenomena — may be key to overcoming said difficulty. We hope that our use of quantum entanglement will encourage further research along this line.

## 2   Preliminaries

For simplicity, the main text treats locally connected neural networks whose input data is one dimensional (*e.g.* text or audio). We defer to Appendix J an extension of the analysis and experiments to models intaking data of arbitrary dimension (*e.g.* two-dimensional images). Due to lack of space, we also defer our review of related work to Appendix A.

We use $\|\cdot\|$ and $\langle \cdot, \cdot \rangle$ to denote the Euclidean (Frobenius) norm and inner product, respectively. We shorthand $[N] := \{1, \ldots, N\}$, where $N \in \mathbb{N}$. The complement of $\mathcal{K} \subseteq [N]$ is denoted by $\mathcal{K}^c$, *i.e.* $\mathcal{K}^c := [N] \setminus \mathcal{K}$.

### 2.1   Tensors and Tensor Networks

For our purposes, a *tensor* is an array with multiple axes $\mathcal{A} \in \mathbb{R}^{D_1 \times \cdots \times D_N}$, where $N \in \mathbb{N}$ is its *order* and $D_1, \ldots, D_N \in \mathbb{N}$ are its *axes lengths*. The $(d_1, \ldots, d_N)$'th entry of $\mathcal{A}$ is denoted $\mathcal{A}_{d_1, \ldots, d_N}$.

*Contraction* between tensors is a generalization of multiplication between matrices. Two matrices $\mathbf{A} \in \mathbb{R}^{D_1 \times D_2}$ and $\mathbf{B} \in \mathbb{R}^{D'_1 \times D'_2}$ can be multiplied if $D_2 = D'_1$, in which case we get a matrix in $\mathbb{R}^{D_1 \times D'_2}$ holding $\sum_{d=1}^{D_2} \mathbf{A}_{d_1, d} \cdot \mathbf{B}_{d, d'_2}$ in entry $(d_1, d'_2) \in [D_1] \times [D'_2]$. More generally, two tensors $\mathcal{A} \in \mathbb{R}^{D_1 \times \cdots \times D_N}$ and $\mathcal{B} \in \mathbb{R}^{D'_1 \times \cdots \times D'_{N'}}$ can be contracted along axis $n \in [N]$ of $\mathcal{A}$ and $n' \in [N']$ of $\mathcal{B}$ if $D_n = D'_{n'}$, in which case we get a tensor of size $D_1 \times \cdots D_{n-1} \times D_{n+1} \times \cdots \times D_N \times D'_1 \times \cdots \times D'_{n'-1} \times D'_{n'+1} \cdots \times D'_{N'}$ holding $\sum_{d=1}^{D_n} \mathcal{A}_{d_1, \ldots, d_{n-1}, d, d_{n+1}, \ldots, d_N} \cdot \mathcal{B}_{d'_1, \ldots, d'_{n'-1}, d, d'_{n'+1}, \ldots, d'_{N'}}$ in the entry indexed by $\{d_k \in [D_k]\}_{k \in [N] \setminus \{n\}}$ and $\{d'_k \in [D'_k]\}_{k \in [N'] \setminus \{n'\}}$.

*Tensor networks* are prominent computational models for fitting (*i.e.* representing) tensors. More specifically, a tensor network is a weighted graph that describes formation of a (typically high-order)

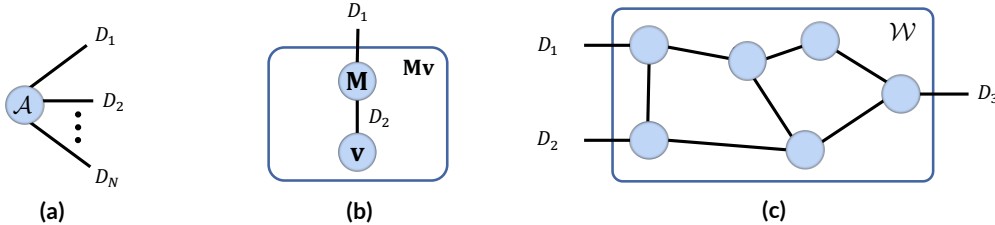

**(a)**           **(b)**           **(c)**

Figure 1: Tensor networks form a graphical language for fitting (*i.e.* representing) tensors through tensor contractions. **Tensor network definition:** Every node in a tensor network is associated with a tensor, whose order is equal to the number of edges emanating from the node. An edge connecting two nodes specifies contraction between the tensors associated with the nodes (Section 2.1), where the weight of the edge signifies the respective axes lengths. Tensor networks may also contain open edges, *i.e.* edges that are connected to a node on one side and are open on the other. The number of such open edges is equal to the order of the tensor produced by contracting the tensor network. **Illustrations:** Presented are exemplar tensor network diagrams of: **(a)** an order $N$ tensor $\mathcal{A} \in \mathbb{R}^{D_1 \times \cdots \times D_N}$; **(b)** a vector-matrix multiplication between $\mathbf{M} \in \mathbb{R}^{D_1 \times D_2}$ and $\mathbf{v} \in \mathbb{R}^{D_2}$, which results in the vector $\mathbf{Mv} \in \mathbb{R}^{D_1}$; and **(c)** a tensor network generating $\mathcal{W} \in \mathbb{R}^{D_1 \times D_2 \times D_3}$.

tensor via contractions between (typically low-order) tensors. As customary (*cf.* [42]), we will present tensor networks via graphical diagrams to avoid cumbersome notation — see Figure 1 for details.

## 2.2 Quantum Entanglement

In quantum physics, the distribution of possible states for a multi-particle ("many body") system is described by a tensor, whose axes are associated with individual particles. A key property of the distribution is the dependence it admits under a given partition of the particles (*i.e.* between a given set of particles and its complement). This dependence is formalized through the notion of *quantum entanglement*, defined using the distribution's description as a tensor — see Definition 1 below.

Quantum entanglement lies at the heart of a widely accepted theory which allows assessing the ability of a tensor network to fit a given tensor (*cf.* [16, 35]). In Section 3 we specialize this theory to a tensor network equivalent to a certain locally connected neural network. The specialized theory will be used (in Section 4) to establish our main theoretical contribution: a necessary and sufficient condition for when the locally connected neural network is capable of accurate prediction over a data distribution.

**Definition 1.** For a tensor $\mathcal{A} \in \mathbb{R}^{D_1 \times \cdots \times D_N}$ and subset of its axes $\mathcal{K} \subseteq [N]$, let $[\![\mathcal{A}; \mathcal{K}]\!] \in \mathbb{R}^{\prod_{n \in \mathcal{K}} D_n \times \prod_{n \in \mathcal{K}^c} D_n}$ be the arrangement of $\mathcal{A}$ as a matrix where rows correspond to axes $\mathcal{K}$ and columns correspond to the remaining axes $\mathcal{K}^c := [N] \setminus \mathcal{K}$. Denote by $\sigma_1 \geq \cdots \geq \sigma_{D_{\mathcal{K}}} \in \mathbb{R}_{\geq 0}$ the singular values of $[\![\mathcal{A}; \mathcal{K}]\!]$, where $D_{\mathcal{K}} := \min\{\prod_{n \in \mathcal{K}} D_n, \prod_{n \in \mathcal{K}^c} D_n\}$. The *quantum entanglement*[2] of $\mathcal{A}$ with respect to the partition $(\mathcal{K}, \mathcal{K}^c)$ is the entropy of the distribution $\{\rho_d := \sigma_d^2 / \sum_{d'=1}^{D_{\mathcal{K}}} \sigma_{d'}^2\}_{d=1}^{D_{\mathcal{K}}}$, *i.e.* $QE(\mathcal{A}; \mathcal{K}) := -\sum_{d=1}^{D_{\mathcal{K}}} \rho_d \ln(\rho_d)$. By convention, if $\mathcal{A} = 0$ then $QE(\mathcal{A}; \mathcal{K}) = 0$.

# 3 Low Entanglement Under Canonical Partitions Is Necessary and Sufficient for Fitting Tensor

In this section, we prove that a tensor network equivalent to a certain locally connected neural network can fit a tensor if and only if the tensor admits low entanglement under certain canonical partitions of its axes. We begin by introducing the tensor network (Section 3.1). Subsequently, we establish the necessary and sufficient condition required for it to fit a given tensor (Section 3.2). For conciseness, the treatment in this section is limited to one-dimensional (sequential) models; see Appendix J.1 for an extension to arbitrary dimensions.

## 3.1 Tensor Network Equivalent to a Locally Connected Neural Network

Let $N \in \mathbb{N}$, and for simplicity suppose that $N = 2^L$ for some $L \in \mathbb{N}$. We consider a tensor network with an underlying perfect binary tree graph of height $L$, which generates $\mathcal{W}_{\text{TN}} \in \mathbb{R}^{D_1 \times \cdots \times D_N}$. Figure 2(a) provides its diagrammatic definition. For simplicity of presentation, the lengths of axes corresponding to inner (non-open) edges are taken to all be equal to some $R \in \mathbb{N}$, referred to as the *width* of the tensor network.

---

[2]There exist multiple notions of entanglement in quantum physics (see, *e.g.*, [35]). The one we consider is the most common, known as *entanglement entropy*.

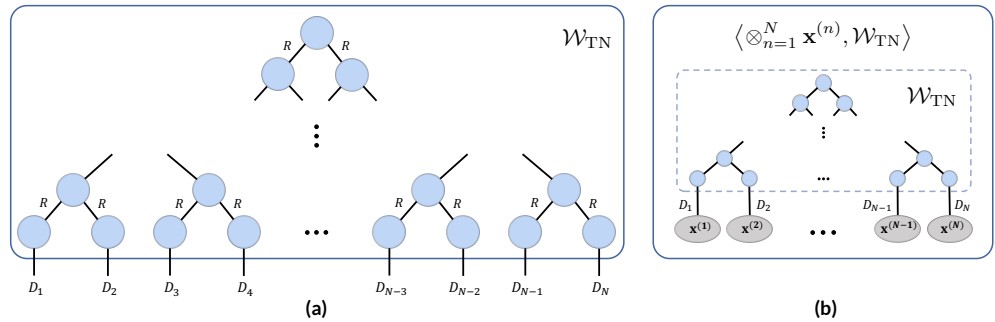

Figure 2: The analyzed tensor network equivalent to a locally connected neural network. **(a)** We consider a tensor network adhering to a perfect binary tree connectivity with $N = 2^L$ leaf nodes, for $L \in \mathbb{N}$, generating $\mathcal{W}_{\mathrm{TN}} \in \mathbb{R}^{D_1 \times \cdots \times D_N}$. Axes corresponding to open edges are indexed such that open edges descendant to any node of the tree have contiguous indices. The lengths of axes corresponding to inner (non-open) edges are equal to $R \in \mathbb{N}$, referred to as the *width* of the tensor network. **(b)** Contracting $\mathcal{W}_{\mathrm{TN}}$ with vectors $\mathbf{x}^{(1)} \in \mathbb{R}^{D_1}, \ldots, \mathbf{x}^{(N)} \in \mathbb{R}^{D_N}$ produces $\langle \otimes_{n=1}^{N} \mathbf{x}^{(n)}, \mathcal{W}_{\mathrm{TN}} \rangle$. Performing these contractions from leaves to root can be viewed as a forward pass of a data instance $(\mathbf{x}^{(1)}, \ldots, \mathbf{x}^{(N)})$ through a certain locally connected neural network (with polynomial non-linearity; see, *e.g.*, [12, 10, 35, 49]). Accordingly, we call the tensor network generating $\mathcal{W}_{\mathrm{TN}}$ a *locally connected tensor network*.

As identified by previous works, the tensor network depicted in Figure 2(a) is equivalent to a certain locally connected neural network (with polynomial non-linearity — see, *e.g.*, [12, 10, 35, 49]). In particular, contracting the tensor network with vectors $\mathbf{x}^{(1)} \in \mathbb{R}^{D_1}, \ldots, \mathbf{x}^{(N)} \in \mathbb{R}^{D_N}$, as illustrated in Figure 2(b), can be viewed as a forward pass of the data instance $(\mathbf{x}^{(1)}, \ldots, \mathbf{x}^{(N)})$ through a locally connected neural network, whose hidden layers are of width $R$. This computation results in a scalar equal to $\langle \otimes_{n=1}^{N} \mathbf{x}^{(n)}, \mathcal{W}_{\mathrm{TN}} \rangle$, where $\otimes$ stands for the outer product.[3] In light of its equivalence to a locally connected neural network, we will refer to the tensor network as a *locally connected tensor network*. We note that for the equivalent neural network to be practical (in terms of memory and runtime), the width $R$ needs to be of moderate size (typically no more than a few thousands). Specifically, $R$ cannot be exponential in the order $N$, meaning $\ln(R)$ needs to be much smaller than $N$.

By virtue of the locally connected tensor network's equivalence to a deep neural network, it has been paramount for the study of expressiveness and generalization in deep learning [12, 9, 10, 13, 14, 54, 34, 35, 3, 30, 31, 36, 47, 48, 49, 50]. Although the equivalent deep neural network (which has polynomial non-linearity) is less common than other neural networks (*e.g.*, ones with ReLU non-linearity), it has demonstrated competitive performance in practice [8, 11, 55, 58, 22]. More importantly, its theoretical analyses, through the equivalence to the locally connected tensor network, brought forth numerous insights that were demonstrated empirically and led to development of practical tools for common locally connected architectures. Continuing this line, we will demonstrate our theoretical insights through experiments with widespread convolutional, recurrent and local self-attention architectures (Section 4.3), and employ our theory for deriving an algorithm that enhances the suitability of a data distribution to said architectures (Section 5).

### 3.2   Necessary and Sufficient Condition for Fitting Tensor

Herein we show that the ability of the locally connected tensor network (defined in Section 3.1) to fit (*i.e.* represent) a given tensor is determined by the entanglements that the tensor admits under the below-defined *canonical partitions* of $[N]$. Note that each canonical partition comprises a subset of contiguous indices, so low entanglement under canonical partitions can be viewed as a formalization of locality.

**Definition 2.** The *canonical partitions* of $[N] = [2^L]$ (illustrated in Figure 4 of Appendix B) are:

$$\mathcal{C}_N := \Big\{ (\mathcal{K}, \mathcal{K}^c) : \mathcal{K} = \big\{ 2^{L-l} \cdot (n-1) + 1, \ldots, 2^{L-l} \cdot n \big\}, \, l \in \{0, \ldots, L\}, \, n \in [2^l] \Big\}.$$

By appealing to known upper bounds on the entanglements that a given tensor network supports [16, 35], we now establish that if the locally connected tensor network can fit a given tensor, that tensor

---

[3]For any $\{\mathbf{x}^{(n)} \in \mathbb{R}^{D_n}\}_{n=1}^{N}$, the outer product $\otimes_{n=1}^{N} \mathbf{x}^{(n)} \in \mathbb{R}^{D_1 \times \cdots \times D_N}$ is defined element-wise by $[\otimes_{n=1}^{N} \mathbf{x}^{(n)}]_{d_1, \ldots, d_N} = \prod_{n=1}^{N} \mathbf{x}_{d_n}^{(n)}$, where $d_1 \in [D_1], \ldots, d_N \in [D_N]$.

must admit low entanglement under the canonical partitions of its axes. Namely, suppose that $\mathcal{W}_{\text{TN}}$ — the tensor generated by the locally connected tensor network — well-approximates an order $N$ tensor $\mathcal{A}$. Then, Theorem 1 below shows that the entanglement of $\mathcal{A}$ with respect to every canonical partition $(\mathcal{K}, \mathcal{K}^c) \in \mathcal{C}_N$ cannot be much larger than $\ln(R)$ (recall that $R$ is the width of the locally connected tensor network, and that in practical settings $\ln(R)$ is much smaller than $N$), whereas the expected entanglement of a random tensor with respect to $(\mathcal{K}, \mathcal{K}^c)$ is on the order of $\min\{|\mathcal{K}|, |\mathcal{K}^c|\}$ (which is linear in $N$ for most canonical partitions).

In the other direction, Theorem 2 below implies that low entanglement under the canonical partitions is not only necessary for a tensor to be fit by the locally connected tensor network, but also sufficient.

**Theorem 1.** *Let $\mathcal{W}_{\text{TN}} \in \mathbb{R}^{D_1 \times \cdots \times D_N}$ be a tensor generated by the locally connected tensor network defined in Section 3.1, and let $\mathcal{A} \in \mathbb{R}^{D_1 \times \cdots \times D_N}$. For any $\epsilon \in [0, \|\mathcal{A}\|/4]$, if $\|\mathcal{W}_{\text{TN}} - \mathcal{A}\| \leq \epsilon$, then for all canonical partitions $(\mathcal{K}, \mathcal{K}^c) \in \mathcal{C}_N$ (Definition 2) it holds that $QE(\mathcal{A}; \mathcal{K}) \leq \ln(R) + \frac{2\epsilon}{\|\mathcal{A}\|} \cdot \ln(D_\mathcal{K}) + 2\sqrt{\frac{2\epsilon}{\|\mathcal{A}\|}}$, where $D_\mathcal{K} := \min\{\prod_{n \in \mathcal{K}} D_n, \prod_{n \in \mathcal{K}^c} D_n\}$.[4] In contrast, a random tensor $\mathcal{A}' \in \mathbb{R}^{D_1 \times \cdots \times D_N}$, drawn according to the uniform distribution over the set of unit norm tensors, satisfies $\mathbb{E}[QE(\mathcal{A}'; \mathcal{K})] \geq \min\{|\mathcal{K}|, |\mathcal{K}^c|\} \cdot \ln(\min_{n \in [N]} D_n) + \ln(\frac{1}{2}) - \frac{1}{2}$ for all canonical partitions $(\mathcal{K}, \mathcal{K}^c) \in \mathcal{C}_N$.*

*Proof sketch (proof in Appendix M.2).* In general, the entanglements that a tensor network supports can be upper bounded through cuts in its graph [16, 35]. For the locally connected tensor network, these bounds imply that $QE(\mathcal{W}_{\text{TN}}; \mathcal{K}) \leq \ln(R)$ for any canonical partition $(\mathcal{K}, \mathcal{K}^c)$. The upper bounds on the entanglements of $\mathcal{A}$ then follow by showing that if $\mathcal{W}_{\text{TN}}$ and $\mathcal{A}$ are close, then so are their entanglements. The lower bounds on the expected entanglements of a random tensor are derived based on a characterization from [52]. □

**Theorem 2.** *Let $\mathcal{A} \in \mathbb{R}^{D_1 \times \cdots \times D_N}$ and $\epsilon > 0$. Suppose that for all canonical partitions $(\mathcal{K}, \mathcal{K}^c) \in \mathcal{C}_N$ (Definition 2) it holds that $QE(\mathcal{A}; \mathcal{K}) \leq \frac{\epsilon^2}{(2N-3)\|\mathcal{A}\|^2} \cdot \ln(R)$.[5] Then, there exists an assignment for the tensors constituting the locally connected tensor network (defined in Section 3.1) such that it generates $\mathcal{W}_{\text{TN}} \in \mathbb{R}^{D_1 \times \cdots \times D_N}$ satisfying $\|\mathcal{W}_{\text{TN}} - \mathcal{A}\| \leq \epsilon$.*

*Proof sketch (proof in Appendix M.3).* We show that if $\mathcal{A}$ has low entanglement under a canonical partition $(\mathcal{K}, \mathcal{K}^c) \in \mathcal{C}_N$, then the singular values of $[\![\mathcal{A}; \mathcal{K}]\!]$ must decay rapidly (recall that $[\![\mathcal{A}; \mathcal{K}]\!]$ is the arrangement of $\mathcal{A}$ as a matrix where rows correspond to axes indexed by $\mathcal{K}$ and columns correspond to the remaining axes). The approximation guarantee is then obtained through a construction from [25], which is based on truncated singular value decompositions of every $[\![\mathcal{A}; \mathcal{K}]\!]$ for $(\mathcal{K}, \mathcal{K}^c) \in \mathcal{C}_N$. □

## 4 Low Entanglement Under Canonical Partitions Is Necessary and Sufficient for Accurate Prediction

In this section we consider the locally connected tensor network from Section 3.1 in a machine learning setting. We show that attaining low population loss amounts to fitting a tensor defined by the data distribution, whose axes correspond to features (Section 4.1). Applying the theorems of Section 3.2, we then conclude that the locally connected tensor network is capable of accurate prediction if and only if the data distribution admits low entanglement under canonical partitions of features (Section 4.2). This conclusion is corroborated through experiments, demonstrating that the performance of common locally connected neural networks (including convolutional, recurrent, and local self-attention neural networks) is inversely correlated with the entanglement under canonical partitions of features in the data (Section 4.3). For conciseness, the treatment in this section is limited to one-dimensional (sequential) models and data; see Appendix J.2 for an extension to arbitrary dimensions.

---

[4]If $\mathcal{A} = 0$, then $\epsilon = 0$. In this case, the expression $\epsilon/\|\mathcal{A}\|$ is by convention equal to zero.

[5] When the approximation error $\epsilon$ tends to zero, the sufficient condition in Theorem 2 requires entanglements to approach zero, unlike the necessary condition in Theorem 1 which requires entanglements to become no greater than $\ln(R)$. This is unavoidable. However, if for all canonical partitions $(\mathcal{K}, \mathcal{K}^c) \in \mathcal{C}_N$ the singular values of $[\![\mathcal{A}; \mathcal{K}]\!]$ trailing after the $R$'th one are small, then we can also guarantee an assignment for the locally connected tensor network satisfying $\|\mathcal{W}_{\text{TN}} - \mathcal{A}\| \leq \epsilon$, while $QE(\mathcal{A}; \mathcal{K})$ can be on the order of $\ln(R)$ for all $(\mathcal{K}, \mathcal{K}^c) \in \mathcal{C}_N$. See Appendix C for details.

## 4.1 Accurate Prediction Is Equivalent to Fitting Data Tensor

As discussed in Section 3.1, the locally connected tensor network generating $\mathcal{W}_{\mathrm{TN}} \in \mathbb{R}^{D_1 \times \cdots \times D_N}$ is equivalent to a locally connected neural network, whose forward pass over a data instance $(\mathbf{x}^{(1)}, \ldots, \mathbf{x}^{(N)})$ yields $\langle \otimes_{n=1}^{N} \mathbf{x}^{(n)}, \mathcal{W}_{\mathrm{TN}} \rangle$, where $\mathbf{x}^{(1)} \in \mathbb{R}^{D_1}, \ldots, \mathbf{x}^{(N)} \in \mathbb{R}^{D_N}$. Motivated by this fact, we consider a binary classification setting, in which the label $y$ of the instance $(\mathbf{x}^{(1)}, \ldots, \mathbf{x}^{(N)})$ is either $1$ or $-1$, and the prediction $\hat{y}$ is taken to be the sign of the output of the neural network, *i.e.* $\hat{y} = \mathrm{sign}\big( \langle \otimes_{n=1}^{N} \mathbf{x}^{(n)}, \mathcal{W}_{\mathrm{TN}} \rangle \big)$.

Suppose we are given a training set of labeled instances $\big\{ \big( (\mathbf{x}^{(1,m)}, \ldots, \mathbf{x}^{(N,m)}), y^{(m)} \big) \big\}_{m=1}^{M}$ drawn i.i.d. from some distribution, and we would like to learn the parameters of the neural network through the soft-margin support vector machine (SVM) objective, *i.e.* by optimizing:

$$\min_{\|\mathcal{W}_{\mathrm{TN}}\| \leq B} \frac{1}{M} \sum_{m=1}^{M} \max \Big\{ 0, 1 - y^{(m)} \langle \otimes_{n=1}^{N} \mathbf{x}^{(n,m)}, \mathcal{W}_{\mathrm{TN}} \rangle \Big\}, \tag{1}$$

for a predetermined constant $B > 0$. We assume instances are normalized, *i.e.* the distribution is such that all vectors constituting an instance have norm no greater than one. We also assume that $B \leq 1$. In this case $\big| y^{(m)} \langle \otimes_{n=1}^{N} \mathbf{x}^{(n,m)}, \mathcal{W}_{\mathrm{TN}} \rangle \big| \leq 1$, so our optimization problem can be expressed as $\min_{\|\mathcal{W}_{\mathrm{TN}}\| \leq B} 1 - \langle \mathcal{D}_{\mathrm{emp}}, \mathcal{W}_{\mathrm{TN}} \rangle$, where

$$\mathcal{D}_{\mathrm{emp}} := \frac{1}{M} \sum_{m=1}^{M} y^{(m)} \cdot \otimes_{n=1}^{N} \mathbf{x}^{(n,m)} \tag{2}$$

is referred to as the *empirical data tensor*. This means that the accuracy over the training data is determined by how large the inner product $\langle \mathcal{D}_{\mathrm{emp}}, \mathcal{W}_{\mathrm{TN}} \rangle$ is.

Disregarding the degenerate case of $\mathcal{D}_{\mathrm{emp}} = 0$ (in which the optimized objective is constant), the inner products $\langle \mathcal{D}_{\mathrm{emp}}, \mathcal{W}_{\mathrm{TN}} \rangle$ and $\langle \mathcal{D}_{\mathrm{emp}}/\|\mathcal{D}_{\mathrm{emp}}\|, \mathcal{W}_{\mathrm{TN}} \rangle$ differ by only a multiplicative (positive) constant, so fitting the training data amounts to optimizing $\max_{\|\mathcal{W}_{\mathrm{TN}}\| \leq B} \langle \mathcal{D}_{\mathrm{emp}}/\|\mathcal{D}_{\mathrm{emp}}\|, \mathcal{W}_{\mathrm{TN}} \rangle$. If $\mathcal{W}_{\mathrm{TN}}$ can represent some $\mathcal{W}$, then it can also represent $c \cdot \mathcal{W}$ for every $c \in \mathbb{R}$. Thus, we may equivalently optimize $\max_{\mathcal{W}_{\mathrm{TN}}} \langle \mathcal{D}_{\mathrm{emp}}/\|\mathcal{D}_{\mathrm{emp}}\|, \mathcal{W}_{\mathrm{TN}}/\|\mathcal{W}_{\mathrm{TN}}\| \rangle$ and multiply the result by $B$. Fitting the training data therefore boils down to minimizing $\big\| \frac{\mathcal{W}_{\mathrm{TN}}}{\|\mathcal{W}_{\mathrm{TN}}\|} - \frac{\mathcal{D}_{\mathrm{emp}}}{\|\mathcal{D}_{\mathrm{emp}}\|} \big\|$. In other words, the accuracy achievable over the training data is determined by the extent to which $\frac{\mathcal{W}_{\mathrm{TN}}}{\|\mathcal{W}_{\mathrm{TN}}\|}$ can fit the normalized empirical data tensor $\frac{\mathcal{D}_{\mathrm{emp}}}{\|\mathcal{D}_{\mathrm{emp}}\|}$.

The arguments above are independent of the training set size $M$, and in fact apply to the population loss as well, in which case $\mathcal{D}_{\mathrm{emp}}$ is replaced by the *population data tensor*:

$$\mathcal{D}_{\mathrm{pop}} := \mathbb{E}_{(\mathbf{x}^{(1)}, \ldots, \mathbf{x}^{(N)}), y} \big[ y \cdot \otimes_{n=1}^{N} \mathbf{x}^{(n)} \big]. \tag{3}$$

Disregarding the degenerate case of $\mathcal{D}_{\mathrm{pop}} = 0$ (*i.e.* that in which the population loss is constant), it follows that the achievable accuracy over the population is determined by the extent to which $\frac{\mathcal{W}_{\mathrm{TN}}}{\|\mathcal{W}_{\mathrm{TN}}\|}$ can fit the normalized population data tensor $\frac{\mathcal{D}_{\mathrm{pop}}}{\|\mathcal{D}_{\mathrm{pop}}\|}$. We refer to the minimal distance from it as the *suboptimality in achievable accuracy*.

**Definition 3.** In the context of the classification setting above, the *suboptimality in achievable accuracy* is $\mathrm{SubOpt} := \min_{\mathcal{W}_{\mathrm{TN}}} \big\| \frac{\mathcal{W}_{\mathrm{TN}}}{\|\mathcal{W}_{\mathrm{TN}}\|} - \frac{\mathcal{D}_{\mathrm{pop}}}{\|\mathcal{D}_{\mathrm{pop}}\|} \big\|$.

## 4.2 Necessary and Sufficient Condition for Accurate Prediction

In the classification setting of Section 4.1, by invoking Theorems 1 and 2 from Section 3.2, we conclude that the suboptimality in achievable accuracy is small if and only if the population data tensor $\mathcal{D}_{\mathrm{pop}}$ admits low entanglement under the canonical partitions of its axes (Definition 2). This is formalized in Corollary 1 below. The quantum entanglement of $\mathcal{D}_{\mathrm{pop}}$ with respect to an arbitrary partition of its axes $(\mathcal{K}, \mathcal{K}^c)$, where $\mathcal{K} \subseteq [N]$, is a measure of dependence between the data features indexed by $\mathcal{K}$ and those indexed by $\mathcal{K}^c$ — see Appendix D for intuition behind this. Thus, Corollary 1 implies that the suboptimality in achievable accuracy is small if and only if the data admits low dependence under the canonical partitions of features. Since canonical partitions comprise a subset with contiguous indices, we obtain a formalization of the intuition by which data distributions suitable for locally connected neural networks are those exhibiting a "local nature."

Directly evaluating the conditions required by Corollary 1 — low entanglement under canonical partitions for $\mathcal{D}_{\mathrm{pop}}$ — is impractical, since: *(i)* $\mathcal{D}_{\mathrm{pop}}$ is defined via an unknown data distribution

(Equation (3)); and *(ii)* computing the entanglements involves taking singular value decompositions of matrices with size exponential in the number of input variables $N$. Fortunately, as Proposition 2 in Appendix E shows, $\mathcal{D}_{\text{pop}}$ is with high probability well-approximated by the empirical data tensor $\mathcal{D}_{\text{emp}}$. Moreover, the entanglement of $\mathcal{D}_{\text{emp}}$ under any partition can be computed efficiently, without explicitly storing or manipulating an exponentially large matrix — see Appendix F for an algorithm (originally proposed in [40]). Overall, we obtain an efficiently computable criterion (low entanglement under canonical partitions for $\mathcal{D}_{\text{emp}}$), that with high probability is both necessary and sufficient for low suboptimality in achievable accuracy (see Corollary 2 in Appendix E for a formalization).

**Corollary 1.** *Consider the classification setting of Section 4.1, and let $\epsilon \in [0, 1/4]$. If there exists a canonical partition $(\mathcal{K}, \mathcal{K}^c) \in \mathcal{C}_N$ (Definition 2) under which $QE(\mathcal{D}_{\text{pop}}; \mathcal{K}) > \ln(R) + 2\epsilon \cdot \ln(D_{\mathcal{K}}) + 2\sqrt{2}\epsilon$, where $R$ is the width of the locally connected tensor network and $D_{\mathcal{K}} := \min\{\prod_{n \in \mathcal{K}} D_n, \prod_{n \in \mathcal{K}^c} D_n\}$, then $\text{SubOpt} > \epsilon$. Conversely, if for all $(\mathcal{K}, \mathcal{K}^c) \in \mathcal{C}_N$ it holds that $QE(\mathcal{D}_{\text{pop}}; \mathcal{K}) \leq \frac{\epsilon^2}{8N-12} \cdot \ln(R)$, then $\text{SubOpt} \leq \epsilon$.*

*Proof sketch (proof in Appendix M.4).* The result follows from Theorems 1 and 2 after accounting for the normalization of $\mathcal{W}_{\text{TN}}$ in the definition of $\text{SubOpt}$ (Definition 3). $\qquad\square$

### 4.3 Empirical Demonstration

Corollary 2 establishes that, with high probability, the locally connected tensor network (from Section 3.1) can achieve high prediction accuracy if and only if the empirical data tensor (Equation (2)) admits low entanglement under canonical partitions of its axes. We corroborate our formal analysis through experiments, demonstrating that its conclusions carry over to common locally connected architectures. Namely, applying convolutional neural networks, S4 (a popular recurrent neural network; see [26]), and a local self-attention model [46] to different datasets, we show that the achieved test accuracy is inversely correlated with the entanglements of the empirical data tensor under canonical partitions. Below is a description of experiments with one-dimensional (*i.e.* sequential) models and data. Additional experiments with two-dimensional (imagery) models and data are given in Appendix J.2.3.

Discerning the relation between entanglements of the empirical data tensor and performance (prediction accuracy) of locally connected neural networks requires datasets admitting different entanglements. A potential way to acquire such datasets is as follows. First, select a dataset on which locally connected neural networks perform well, in the hopes that it admits low entanglement under canonical partitions; natural candidates are datasets comprising images, text or audio. Subsequently, create "shuffled" variants of the dataset by repeatedly swapping the position of two features chosen at random.[6] This erodes the original arrangement of features in the data, and is expected to yield higher entanglement under canonical partitions.

We followed the blueprint above for a binary classification version of the Speech Commands audio dataset [64]. Figure 3 presents test accuracies achieved by a convolutional neural network, S4, and a local self-attention model, as well as average entanglement under canonical partitions of the empirical data tensor, against the number of random feature swaps performed to create the dataset. As expected, when the number of swaps increases, the average entanglement under canonical partitions becomes higher. At the same time, in accordance with our theory, the prediction accuracies of the locally connected neural networks substantially deteriorate, showing an inverse correlation with the entanglement under canonical partitions.

## 5 Enhancing Suitability of Data to Locally Connected Neural Networks

Our analysis (Sections 3 and 4) suggests that a data distribution is suitable for locally connected neural networks if and only if it admits low entanglement under canonical partitions of features. Motivated by this observation, we derive a preprocessing algorithm aimed to enhance the suitability of a data distribution to locally connected neural networks (Section 5.1 and Appendix G). Empirical evaluations demonstrate that it significantly improves prediction accuracies of common locally connected neural networks on various datasets (Section 5.2). For conciseness, the treatment in this section is limited to one-dimensional (sequential) models and data; see Appendix J.3 for an extension to arbitrary dimensions.

---

[6]It is known that as the number of random position swaps goes to infinity, the arrangement of the features converges to a random permutation [18].

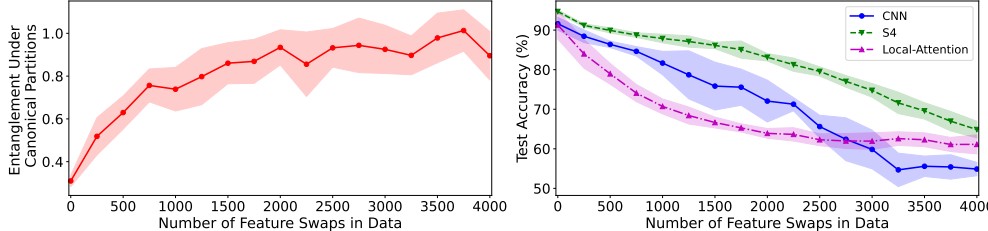

Figure 3: The prediction accuracies of common locally connected neural networks are inversely correlated with the entanglements of the data under canonical partitions of features, in compliance with our theory (Sections 4.1 and 4.2). **Left:** Average entanglement under canonical partitions (Definition 2) of the empirical data tensor (Equation (2)), for binary classification variants of the Speech Commands audio dataset [64] obtained by performing random position swaps between features. **Right:** Test accuracies achieved by a convolutional neural network (CNN) [17], S4 (a popular class of recurrent neural networks; see [26]), and a local self-attention model [46], against the number of random feature swaps performed to create the dataset. **All:** Reported are the means and standard deviations of the quantities specified above, taken over ten different random seeds. See Appendix J.2.3 for experiments over (two-dimensional) image data and Appendix L for implementation details.

## 5.1 Search for Feature Arrangement With Low Entanglement Under Canonical Partitions

Our analysis naturally leads to a recipe for enhancing the suitability of a data distribution to locally connected neural networks: given a dataset, search for an arrangement of features which leads to low entanglement under canonical partitions, and then arrange the features accordingly. Formally, suppose we have $M \in \mathbb{N}$ training instances $\left\{ \left( (\mathbf{x}^{(1,m)}, \ldots, \mathbf{x}^{(N,m)}), y^{(m)} \right) \right\}_{m=1}^{M}$, where $y^{(m)} \in \{1, -1\}$ and $\mathbf{x}^{(n,m)} \in \mathbb{R}^D$ for $n \in [N], m \in [M]$, with $D \in \mathbb{N}$. Assume without loss of generality that $N$ is a power of two (if this is not the case we may add constant features as needed). The aforementioned recipe boils down to a search for a permutation $\pi : [N] \to [N]$, which when applied to feature indices leads the empirical data tensor $\mathcal{D}_{\mathrm{emp}}$ (Equation (2)) to admit low entanglement under the canonical partitions of its axes (Definition 2).

A greedy realization of the foregoing search is as follows. Initially, partition the features into two equally sized sets $\mathcal{K}_{1,1} \subset [N]$ and $\mathcal{K}_{1,2} := [N] \setminus \mathcal{K}_{1,1}$ such that the entanglement of $\mathcal{D}_{\mathrm{emp}}$ with respect to $(\mathcal{K}_{1,1}, \mathcal{K}_{1,2})$ is minimal. That is, find $\mathcal{K}_{1,1} \in \mathrm{argmin}_{\mathcal{K} \subset [N], |\mathcal{K}|=N/2} QE(\mathcal{D}_{\mathrm{emp}}; \mathcal{K})$. The permutation $\pi$ will map $\mathcal{K}_{1,1}$ to coordinates $\{1, \ldots, \frac{N}{2}\}$ and $\mathcal{K}_{1,2}$ to $\{\frac{N}{2}+1, \ldots, N\}$. Then, partition $\mathcal{K}_{1,1}$ into two equally sized sets $\mathcal{K}_{2,1} \subset \mathcal{K}_{1,1}$ and $\mathcal{K}_{2,2} := \mathcal{K}_{1,1} \setminus \mathcal{K}_{2,1}$ such that the average of entanglements induced by these sets is minimal, *i.e.* $\mathcal{K}_{2,1} \in \mathrm{argmin}_{\mathcal{K} \subset \mathcal{K}_{1,1}, |\mathcal{K}|=|\mathcal{K}_{1,1}|/2} \frac{1}{2}\big[ QE(\mathcal{D}_{\mathrm{emp}}; \mathcal{K}) + QE(\mathcal{D}_{\mathrm{emp}}; \mathcal{K}_{1,1} \setminus \mathcal{K})\big]$. The permutation $\pi$ will map $\mathcal{K}_{2,1}$ to coordinates $\{1, \ldots, \frac{N}{4}\}$ and $\mathcal{K}_{2,2}$ to $\{\frac{N}{4}+1, \ldots, \frac{N}{2}\}$. A partition of $\mathcal{K}_{1,2}$ into two equally sized sets $\mathcal{K}_{2,3}$ and $\mathcal{K}_{2,4}$ is obtained similarly, where $\pi$ will map $\mathcal{K}_{2,3}$ to coordinates $\{\frac{N}{2}+1, \ldots, \frac{3N}{4}\}$ and $\mathcal{K}_{2,4}$ to $\{\frac{3N}{4}+1, \ldots, N\}$. Continuing in the same fashion, until we reach subsets $\mathcal{K}_{L,1}, \ldots, \mathcal{K}_{L,N}$ consisting of a single feature index each, fully specifies the permutation $\pi$.

Unfortunately, the step lying at the heart of the above scheme — finding a balanced partition that minimizes average entanglement — is computationally prohibitive, and we are not aware of any tools that alleviate the computational difficulty. However, as discussed in Appendix G, if one replaces entanglement with an appropriate surrogate measure, then each search for a balanced partition minimizing average entanglement converts into a *minimum balanced cut problem*, which enjoys a wide array of established approximation tools [29]. We thus obtain a practical algorithm for enhancing the suitability of a data distribution to locally connected neural networks.

## 5.2 Experiments

We empirically evaluate our feature rearrangement method, detailed in Appendix G, using common locally connected neural networks — a convolutional neural network, S4 (popular recurrent neural network; see [26]), and a local self-attention model [46] — over randomly permuted audio datasets (Section 5.2.1) and several tabular datasets (Section 5.2.2). For brevity, we defer some experiments and implementation details to Appendices K and L. Additional experiments with two-dimensional data are given in Appendix J.3.3.

### 5.2.1 Randomly Permuted Audio Datasets

Section 4.3 demonstrated that audio data admits low entanglement under canonical partitions of features, and that randomly permuting the position of features leads this entanglement to increase,

Table 1: Arranging features of randomly permuted audio data via our method (detailed in Appendix G) significantly improves the prediction accuracies of locally connected neural networks. Reported are test accuracies (mean and standard deviation over ten random seeds) of a convolutional neural network (CNN), S4 (a popular recurrent neural network; see [26]), and a local self-attention model [46], over the Speech Commands dataset [64] subject to different arrangements of features: *(i)* a random arrangement; *(ii)* an arrangement provided by applying our method to the random arrangement; and *(iii)* an arrangement provided by applying an adaptation of IGTD [67] — a heuristic scheme designed for convolutional neural networks — to the random arrangement. For each model, we highlight (in boldface) the highest mean accuracy if the difference between that and the second-highest mean accuracy is statistically significant (namely, is larger than the standard deviation corresponding to the former). Our method leads to significant improvements in prediction accuracies, surpassing the improvements brought forth by IGTD. See Appendix K for experiments demonstrating its scalability and implementation details.

|  | Randomly Permuted | Our Method | IGTD |
|---|---|---|---|
| CNN | $5.5 \pm 0.5$ | $\mathbf{18} \pm 1.6$ | $5.7 \pm 0.6$ |
| S4 | $9.4 \pm 0.8$ | $\mathbf{30.1} \pm 1.8$ | $12.8 \pm 1.3$ |
| Local-Attention | $7.8 \pm 0.5$ | $\mathbf{12.9} \pm 0.6$ | $7 \pm 0.6$ |

while substantially degrading the prediction accuracy of locally connected neural networks. A sensible test for our method is to evaluate its ability to recover performance lost due to the random permutation of features.

For the Speech Commands dataset [64], Table 1 compares prediction accuracies of locally connected neural networks on the data: *(i)* subject to a random permutation of features; *(ii)* attained after rearranging the randomly permuted features via our method; and *(iii)* attained after rearranging the randomly permuted features via IGTD [67] — a heuristic scheme designed for convolutional neural networks (see Appendix A). As can be seen, our method leads to significant improvements, surpassing those brought forth by IGTD. Note that the performance lost due to the random permutation of features is not entirely recovered.[7] We believe this relates to phenomena outside the scope of the theory underlying our method (Sections 3 and 4), for example translation invariance in data being beneficial in terms of generalization. Investigation of such phenomena and suitable modification of our method are regarded as promising directions for future work.

The number of features in the audio dataset used in the experiment above is 2048. We demonstrate the scalability of Algorithm 2 by including in Appendix K an experiment over audio data with 50,000 features. In this experiment, instances of the minimum balanced cut problem encountered in Algorithm 2 entail graphs with up to $25 \cdot 10^8$ edges. They are solved using the well-known edge sparsification algorithm of [57] that preserves weights of cuts, allowing for configurable compute and memory consumption (the more resources are consumed, the more accurate the solution will be).

### 5.2.2 Tabular Datasets

The prediction accuracies of locally connected neural networks on tabular data, *i.e.* on data in which features are arranged arbitrarily, is known to be subpar [56]. Tables 2 and 5 report results of experiments with locally connected neural networks over standard tabular benchmarks (namely "semeion", "isolet" and "dna" [60]), demonstrating that arranging features via our method leads to significant improvements in prediction accuracies, surpassing improvements brought forth by IGTD (a heuristic scheme designed for convolutional neural networks [67]). Note that our method does not lead to state of the art prediction accuracies on the evaluated benchmarks.[8] However, the results suggest that it renders locally connected neural networks a viable option for tabular data. This option is particularly appealing in when the number of features is large settings, where many alternative approaches (*e.g.* ones involving fully connected neural networks) are impractical.

## 6 Conclusion

### 6.1 Summary

The question of what makes a data distribution suitable for deep learning is a fundamental open problem. Focusing on locally connected neural networks — a prevalent family of deep learning architectures that includes as special cases convolutional neural networks, recurrent neural networks (in particular the recent S4 models) and local self-attention models — we address this problem by

---

[7]Accuracies on the original data are 59.8, 69.6 and 48.1 for CNN, S4 and Local-Attention, respectively.

[8]XGBoost, *e.g.*, achieves prediction accuracies 91, 95.2 and 96 over semeion, isolet and dna, respectively.

Table 2: Arranging features of tabular datasets via our method (detailed in Appendix G) significantly improves the prediction accuracies of locally connected neural networks. Reported are results of experiments analogous to those of Table 1, but with the "semeion" and "isolet" tabular classification datasets [60]. Since to the arrangement of features in a tabular dataset is intended to be arbitrary, we regard as a baseline the prediction accuracies attained with a random permutation of features. For each combination of dataset and model, we highlight (in boldface) the highest mean accuracy if the difference between that and the second-highest mean accuracy is statistically significant (namely, is larger than the standard deviation corresponding to the former). As in the experiment of Table 1, rearranging the features according to our method leads to significant improvements in prediction accuracies, surpassing the improvements brought forth by IGTD. See Appendix K for experiments with an additional tabular dataset ("dna") and implementation details.

| | Dataset: semeion | | | Dataset: isolet | | |
| --- | --- | --- | --- | --- | --- | --- |
| | Baseline | Our Method | IGTD | Baseline | Our Method | IGTD |
| CNN | $77.7 \pm 1.4$ | $80.0 \pm 1.8$ | $78.9 \pm 1.9$ | $91.0 \pm 0.6$ | $\mathbf{92.5} \pm 0.4$ | $92.0 \pm 0.6$ |
| S4 | $82.5 \pm 1.1$ | $\mathbf{89.7} \pm 0.5$ | $86.0 \pm 0.7$ | $92.3 \pm 0.4$ | $\mathbf{93.4} \pm 0.3$ | $92.7 \pm 0.5$ |
| Local-Attention | $60.9 \pm 4.9$ | $\mathbf{78.0} \pm 1.7$ | $67.8 \pm 2.6$ | $82.0 \pm 1.6$ | $\mathbf{89.0} \pm 0.6$ | $85.7 \pm 1.9$ |

adopting theoretical tools from quantum physics. Our main theoretical result states that a certain locally connected neural network is capable of accurate prediction (*i.e.* can express a solution with low population loss) over a data distribution *if and only if* the data distribution admits low quantum entanglement under certain canonical partitions of features. Experiments with widespread locally connected neural networks corroborate this finding.

Our theory suggests that the suitability of a data distribution to locally connected neural networks may be enhanced by arranging features such that low entanglement under canonical partitions is attained. Employing a certain surrogate for entanglement, we show that this arrangement can be implemented efficiently, and that it leads to substantial improvements in the prediction accuracies of common locally connected neural networks on various datasets.

## 6.2 Limitations and Future Work

**Neural network architecture**  We theoretically analyzed a locally connected neural network with polynomial non-linearity, by employing its equivalence to a tensor network (*cf.* Section 3.1). Accounting for neural networks with connectivities beyond those considered (*e.g.* connectivities that are non-local or ones involving skip connections) is an interesting topic for future work. It requires modification of the equivalent tensor network and corresponding modification of the definition of canonical partitions, similarly to the analysis in Appendix J. Another valuable direction is to account for neural networks with other non-linearities, *e.g.* ReLU. We believe this may be achieved through a generalized notion of tensor networks, successfully used in past work to analyze such architectures [9].

**Objective function**  The analysis in Section 4 assumes a binary soft-margin SVM objective. Extending it to other objective functions, *e.g.* multi-class SVM, may shed light on the relation between the objective and the requirements for a data distribution to be suitable to neural networks.

**Textual data**  Our experiments (in Section 4.3 and Appendix J.2.3) show that the necessary and sufficient condition we derived for a data distribution to be suitable to a locally connected neural network — namely, low quantum entanglement under canonical partitions of features — is upheld by audio and image datasets. This falls in line with the excellent performance of locally connected neural networks over these data modalities. In contrast, high performant architectures for textual data are typically non-local [61]. Investigating the quantum entanglements that textual data admits and, in particular, under which partitions they are low, may allow designing more efficient architectures with connectivity tailored to textual data.

## 6.3 Outlook

The data modalities to which deep learning is most commonly applied — namely ones involving images, text and audio — are often regarded as natural (as opposed to, for example, tabular data fusing heterogeneous information). We believe the difficulty in explaining the suitability of such modalities to deep learning may be due to a shortage in tools for formally reasoning about natural data. Concepts and tools from physics — a branch of science concerned with formally reasoning about natural phenomena — may be key to overcoming said difficulty. We hope that our use of quantum entanglement will encourage further research along this line.

## Acknowledgments and Disclosure of Funding

This work was supported by a Google Research Scholar Award, a Google Research Gift, the Yandex Initiative in Machine Learning, the Israel Science Foundation (grant 1780/21), the Tel Aviv University Center for AI and Data Science, the Adelis Research Fund for Artificial Intelligence, Len Blavatnik and the Blavatnik Family Foundation, and Amnon and Anat Shashua. NR is supported by the Apple Scholars in AI/ML PhD fellowship.

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

# A  Related Work

Characterizing formal properties of data distributions that make them suitable for deep learning is a major open problem in the field. A number of papers provide sufficient conditions on a data distribution which imply that it is learnable by certain neural networks [5, 65, 39, 19, 20, 43, 62, 6]. However, these sufficient conditions are restrictive, and are not argued to be necessary for any aspect of learning (*i.e.* for expressiveness, optimization or generalization). To the best of our knowledge, this paper is the first to derive a verifiable condition on a data distribution that is both necessary and sufficient for aspects of learning to be achievable by a neural network. We note that the condition we derive resembles past attempts to quantify the structure of data via quantum entanglement and mutual information [40, 15, 38, 7, 66, 28, 21]. However, such quantifications have not been formally related to learnability by neural networks.

The current paper follows a long line of research employing tensor networks as theoretical models for studying deep learning. This line includes works analyzing the expressiveness of different neural network architectures [12, 55, 9, 10, 54, 14, 34, 3, 35, 30, 36, 31, 50], their generalization [37], and the implicit regularization induced by their optimization [47, 48, 49, 63, 24]. Similarly to prior works we focus on expressiveness, yet our approach differs in that we incorporate the data distribution into the analysis and tackle the question of what makes data suitable for deep learning.

The algorithm we propose for enhancing the suitability of data to locally connected neural networks can be considered a form of representation learning. Representation learning is a vast field, far too broad to survey here (for an overview see [41]). Our algorithm, which learns a representation via rearrangement of features in the data, is complementary to most representation learning methods in the literature. A notable method that is also based on feature rearrangement is IGTD [67] — a heuristic scheme designed for convolutional neural networks. In contrast to IGTD, our algorithm is theoretically grounded. Moreover, we demonstrate empirically in Section 5 that it leads to higher prediction accuracies.

# B  Illustration of Canonical Partitions

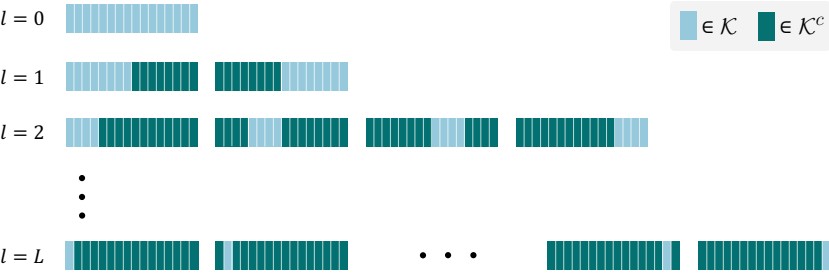

Figure 4: The canonical partitions of $[N]$, for $N = 2^L$ with $L \in \mathbb{N}$. Every $l \in \{0, \dots, L\}$ contributes $2^l$ canonical partitions, the $n$'th one induced by $\mathcal{K} = \{2^{L-l} \cdot (n-1) + 1, \dots, 2^{L-l} \cdot n\}$.

# C  Impossibility Result for Improving the Sufficient Condition in Theorem 2

The sufficient condition in Theorem 2 (from Section 3.2) for approximating $\mathcal{A} \in \mathbb{R}^{D_1 \times \cdots \times D_N}$ requires entanglements to approach zero as the desired approximation error $\epsilon$ does, in contrast to the necessary condition in Theorem 1 where they approach $\ln(R)$. As Proposition 1 shows, this is unavoidable in the absence of further knowledge regarding $\mathcal{A}$. However, if for all canonical partitions $(\mathcal{K}, \mathcal{K}^c) \in \mathcal{C}_N$ the singular values of $[\![\mathcal{A}; \mathcal{K}]\!]$ trailing after the $R$'th one are small, then we can also guarantee an assignment for the locally connected tensor network satisfying $\|\mathcal{W}_{\text{TN}} - \mathcal{A}\| \leq \epsilon$, while $QE(\mathcal{A}; \mathcal{K})$ can be on the order of $\ln(R)$ for all $(\mathcal{K}, \mathcal{K}^c) \in \mathcal{C}_N$. Indeed, this follows directly from a result in [25], which we restate as Lemma 7 for convenience.

**Proposition 1.** *Let* $f : \mathbb{R}^2 \to \mathbb{R}_{\geq 0}$ *be monotonically increasing in its second variable. Suppose that the following statement holds: if a tensor* $\mathcal{A} \in \mathbb{R}^{D_1 \times \cdots \times D_N}$ *satisfies* $QE(\mathcal{A}; \mathcal{K}) \leq f(\|\mathcal{A}\|, \epsilon)$ *for all canonical partitions* $(\mathcal{K}, \mathcal{K}^c) \in \mathcal{C}_N$ *(Definition 2), then there exists an assignment for the tensors constituting the locally connected tensor network (defined in Section 3.1) for which* $\mathcal{W}_{\text{TN}} \in$

$\mathbb{R}^{D_1 \times \cdots \times D_N}$ *upholds:*

$$\|\mathcal{W}_{\text{TN}} - \mathcal{A}\| \leq \epsilon.$$

*Then, for any $\mathcal{A} \in \mathbb{R}^{D_1 \times \cdots \times D_N}$, as the desired approximation error $\epsilon$ goes to zero, so does the sufficient condition on entanglements, i.e.:*

$$\lim_{\epsilon \to 0} f(\|\mathcal{A}\|, \epsilon) = 0.$$

*Proof.* Suppose otherwise, *i.e.* that there exists some $a > 0$ such that $\lim_{\epsilon \to 0} f(a, \epsilon) = \inf_{\epsilon > 0} f(a, \epsilon) = c > 0$. Let $\mathcal{A}$ be a tensor with $\|\mathcal{A}\| = a$ such that $QE(A; \mathcal{K}) < c$ for all canonical partitions $(\mathcal{K}, \mathcal{K}^c) \in \mathcal{C}_N$. Then by assumption, for all $\epsilon > 0$ there exist a tensor generated by the locally connected tensor network, which we denote by $\mathcal{W}_{\text{TN}}(\epsilon) \in \mathbb{R}^{D_1 \times \cdots \times D_N}$, that satisfies:

$$\|\mathcal{W}_{\text{TN}}(\epsilon) - \mathcal{A}\| \leq \epsilon.$$

By Lemma 5, for all $\epsilon > 0$ we have that

$$\text{rank}(\llbracket \mathcal{W}_{\text{TN}}(\epsilon); \mathcal{K} \rrbracket) \leq R,$$

so by the lower semicontinuity of the matrix rank, we have that $\text{rank}(\llbracket \mathcal{A}; \mathcal{K} \rrbracket) \leq R$ as well. But, there are tensors for which this leads to a contradiction, namely tensors with arbitrary low entanglement across all partitions but nearly maximal rank. Indeed, consider tensors of the form

$$\mathcal{Q}(\delta) = a \cdot \frac{\otimes_{n=1}^N \mathbf{v}^{(n)} + \delta \cdot \mathcal{B}}{\|\otimes_{n=1}^N \mathbf{v}^{(n)} + \delta \cdot \mathcal{B}\|} \in \mathbb{R}^{D_1 \times \cdots \times D_N},$$

where $\{\mathbf{v}^{(n)} \in \mathbb{R}^{D_n}\}_{n=1}^N$ are (non-zero) vectors, $\delta > 0$ and $\mathcal{B} \in \mathbb{R}^{D_1 \times \cdots \times D_N}$ is some tensor with matricizations of maximal rank across all canonical partitions (for a proof of the existence of such a tensor see Claim 3 in [35]). Note that $\|\mathcal{Q}(\delta)\| = a$ for all $\delta > 0$. By the triangle inequality have

$$\left\| \frac{\mathcal{Q}(\delta)}{a} - \frac{\otimes_{n=1}^N \mathbf{v}^{(n)}}{\|\otimes_{n=1}^N \mathbf{v}^{(n)}\|} \right\| \leq \frac{\delta \cdot \|\mathcal{B}\|}{\|\otimes_{n=1}^N \mathbf{v}^{(n)} + \delta \cdot \mathcal{B}\|} + \left\| \frac{\otimes_{n=1}^N \mathbf{v}^{(n)}}{\|\otimes_{n=1}^N \mathbf{v}^{(n)} + \delta \cdot \mathcal{B}\|} - \frac{\otimes_{n=1}^N \mathbf{v}^{(n)}}{\|\otimes_{n=1}^N \mathbf{v}^{(n)}\|} \right\|,$$

and so

$$\lim_{\delta \to 0} \left\| \frac{\mathcal{Q}(\delta)}{a} - \frac{\otimes_{n=1}^N \mathbf{v}^{(n)}}{\|\otimes_{n=1}^N \mathbf{v}^{(n)}\|} \right\| = 0.$$

Thus, from Lemma 9 we know that

$$\lim_{\delta \to 0} \left| QE\left( \frac{\mathcal{Q}(\delta)}{a}; \mathcal{K} \right) - QE\left( \frac{\otimes_{n=1}^N \mathbf{v}^{(n)}}{\|\otimes_{n=1}^N \mathbf{v}^{(n)}\|}; \mathcal{K} \right) \right| = 0$$

for all $(\mathcal{K}, \mathcal{K}^c) \in \mathcal{C}_N$. Furthermore, for any $(\mathcal{K}, \mathcal{K}^c) \in \mathcal{C}_N$:

$$QE\left( \frac{\otimes_{n=1}^N \mathbf{v}^{(n)}}{\|\otimes_{n=1}^N \mathbf{v}^{(n)}\|}; \mathcal{K} \right) = 0,$$

and therefore by Theorem 1 for sufficiently small $\delta > 0$ we have

$$QE(\mathcal{Q}(\delta); \mathcal{K}) < c,$$

for all $(\mathcal{K}, \mathcal{K}^c) \in \mathcal{C}_N$. However, $\text{rank}(\llbracket \mathcal{Q}(\delta); \mathcal{K} \rrbracket)$ is (nearly) maximal for all canonical partitions. Indeed, by the triangle inequality for matrix rank we get

$$\text{rank}(\llbracket \mathcal{Q}(\delta); \mathcal{K} \rrbracket) \geq \text{rank}\left( \llbracket \mathcal{B}; \mathcal{K} \rrbracket \right) - \text{rank}\left( \llbracket \otimes_{n=1}^N \mathbf{v}^{(n)}; \mathcal{K} \rrbracket \right) = D_{\mathcal{K}} - 1,$$

where $D_{\mathcal{K}} := \min\{\prod_{n \in \mathcal{K}} D_n, \prod_{n \in \mathcal{K}^c} D_n\}$. $\qquad\square$

## D   Quantum Entanglement as a Measure of Dependence Between Features

In this appendix, we provide intuition behind the following statement made in Section 4.2: the quantum entanglement of $\mathcal{D}_{\text{pop}}$ with respect to an arbitrary partition of its axes $(\mathcal{K}, \mathcal{K}^c)$, where $\mathcal{K} \subseteq [N]$, is a measure of dependence between the data features indexed by $\mathcal{K}$ and those indexed by $\mathcal{K}^c$. Consider the case where the features indexed by $\mathcal{K}$ are statistically independent of those indexed by $\mathcal{K}^c$, and all features are statistically independent of the label $y$. Then $QE(\mathcal{D}_{\text{pop}}; \mathcal{K}) = 0$ (see Lemma 1 below for proof of this fact). If the features are not statistically independent of $y$, but when conditioned on $y$ the features indexed by $\mathcal{K}$ are statistically independent of those indexed by $\mathcal{K}^c$, then $QE(\mathcal{D}_{\text{pop}}; \mathcal{K}) \leq \ln(2)$ (this fact is also established by Lemma 1 below). In general, the higher $QE(\mathcal{D}_{\text{pop}}; \mathcal{K})$ is, the farther the distribution is from the aforementioned situations of independence, implying stronger dependence between the features indexed by $\mathcal{K}$ and those indexed by $\mathcal{K}^c$.

**Lemma 1.** *Consider the classification setting of Section 4.1, and let $\mathcal{K} \subseteq [N]$. If the features indexed by $\mathcal{K}$ are independent of those indexed by $\mathcal{K}^c$ (i.e. $(\mathbf{x}^{(n)})_{n \in \mathcal{K}}$ are independent of $(\mathbf{x}^{(n)})_{n \in \mathcal{K}^c}$), conditioned on the label $y$, then $QE(\mathcal{D}_{\text{pop}}; \mathcal{K}) \leq \ln(2)$. Furthermore, if the features are also independent of $y$, then $QE(\mathcal{D}_{\text{pop}}; \mathcal{K}) = 0$.*

*Proof.* For brevity, denote $\mathbf{X} := (\mathbf{x}^{(1)}, \ldots, \mathbf{x}^{(N)})$, $\mathbf{X}_{\mathcal{K}} := (\mathbf{x}^{(n)})_{n \in \mathcal{K}}$, and $\mathbf{X}_{\mathcal{K}^c} := (\mathbf{x}^{(n)})_{n \in \mathcal{K}^c}$. By the law of total expectation, the entry of $\mathcal{D}_{\text{pop}}$ corresponding to an index tuple $(i_1, \ldots, i_N) \in [D_1] \times \cdots \times [D_N]$ satisfies:

$$(\mathcal{D}_{\text{pop}})_{i_1, \ldots, i_N} = \mathbb{E}_{\mathbf{X}, y}\left[ y \cdot \prod\nolimits_{n=1}^{N} \mathbf{x}_{i_n}^{(n)} \right]$$
$$= \mathbb{P}(y = 1) \cdot \mathbb{E}_{\mathbf{X}}\left[ \prod\nolimits_{n=1}^{N} \mathbf{x}_{i_n}^{(n)} \big| y = 1 \right] - \mathbb{P}(y = -1) \cdot \mathbb{E}_{\mathbf{X}}\left[ \prod\nolimits_{n=1}^{N} \mathbf{x}_{i_n}^{(n)} \big| y = -1 \right].$$

Thus, if the features indexed by $\mathcal{K}$ are independent of those indexed by $\mathcal{K}^c$ given $y$, it holds that:

$$(\mathcal{D}_{\text{pop}})_{i_1, \ldots, i_N} = \mathbb{P}(y = 1) \cdot \mathbb{E}_{\mathbf{X}_{\mathcal{K}}}\left[ \prod\nolimits_{n \in \mathcal{K}} \mathbf{x}_{i_n}^{(n)} \big| y = 1 \right] \cdot \mathbb{E}_{\mathbf{X}_{\mathcal{K}^c}}\left[ \prod\nolimits_{n \in \mathcal{K}^c} \mathbf{x}_{i_n}^{(n)} \big| y = 1 \right]$$
$$- \mathbb{P}(y = -1) \cdot \mathbb{E}_{\mathbf{X}_{\mathcal{K}}}\left[ \prod\nolimits_{n \in \mathcal{K}} \mathbf{x}_{i_n}^{(n)} \big| y = -1 \right] \cdot \mathbb{E}_{\mathbf{X}_{\mathcal{K}^c}}\left[ \prod\nolimits_{n \in \mathcal{K}^c} \mathbf{x}_{i_n}^{(n)} \big| y = -1 \right].$$
$$(4)$$

This implies that we can write $[\![\mathcal{D}_{\text{pop}}; \mathcal{K}]\!] \in \mathbb{R}^{\Pi_{n \in \mathcal{K}} D_n \times \Pi_{n \in \mathcal{K}^c} D_n}$ — the arrangement of $\mathcal{D}_{\text{pop}}$ as a matrix whose rows correspond to axes indexed by $\mathcal{K}$ and columns correspond to the remaining axes — as a weighted sum of two outer products. Specifically, define $\mathbf{v}_{\mathcal{K}}, \mathbf{u}_{\mathcal{K}} \in \mathbb{R}^{\Pi_{n \in \mathcal{K}} D_n}$ to be the vectors holding for each possible indexing tuple $(i_n)_{n \in \mathcal{K}}$ the values $\mathbb{E}_{\mathbf{X}_{\mathcal{K}}}\left[ \prod_{n \in \mathcal{K}} \mathbf{x}_{i_n}^{(n)} \big| y = 1 \right]$ and $\mathbb{E}_{\mathbf{X}_{\mathcal{K}}}\left[ \prod_{n \in \mathcal{K}} \mathbf{x}_{i_n}^{(n)} \big| y = -1 \right]$, respectively (with the arrangement of entries in the vectors being consistent with the rows of $[\![\mathcal{D}_{\text{pop}}; \mathcal{K}]\!]$). Furthermore, let $\mathbf{v}_{\mathcal{K}^c}, \mathbf{u}_{\mathcal{K}^c} \in \mathbb{R}^{\Pi_{n \in \mathcal{K}^c} D_n}$ be defined analogously by replacing $\mathcal{K}$ with $\mathcal{K}^c$ (with the arrangement of entries in the vectors being consistent with the columns of $[\![\mathcal{D}_{\text{pop}}; \mathcal{K}]\!]$). Then, by Equation (4):

$$[\![\mathcal{D}_{\text{pop}}; \mathcal{K}]\!] = \mathbb{P}(y = 1) \cdot \mathbf{v}_{\mathcal{K}} \otimes \mathbf{v}_{\mathcal{K}^c} - \mathbb{P}(y = -1) \cdot \mathbf{u}_{\mathcal{K}} \otimes \mathbf{u}_{\mathcal{K}^c}.$$

Since each outer product forms a rank one matrix, the subaddititivty of rank implies that the rank of $[\![\mathcal{D}_{\text{pop}}; \mathcal{K}]\!]$ is at most two. As a result, $[\![\mathcal{D}_{\text{pop}}; \mathcal{K}]\!]$ has at most two non-zero singular values and $QE(\mathcal{D}_{\text{pop}}; \mathcal{K}) \leq \ln(2)$ (the entropy of a distribution is at most the natural logarithm of the size of its support).

If, in addition, all features in the data are independent of the label $y$, then $\mathbf{v}_{\mathcal{K}} = \mathbf{u}_{\mathcal{K}}$ and $\mathbf{v}_{\mathcal{K}^c} = \mathbf{u}_{\mathcal{K}^c}$, meaning $[\![\mathcal{D}_{\text{pop}}; \mathcal{K}]\!] = (\mathbb{P}(y = 1) - \mathbb{P}(y = -1)) \cdot \mathbf{v}_{\mathcal{K}} \otimes \mathbf{v}_{\mathcal{K}^c}$. In this case, the rank of $[\![\mathcal{D}_{\text{pop}}; \mathcal{K}]\!]$ is at most one, so $QE(\mathcal{D}_{\text{pop}}; \mathcal{K}) = 0$. $\square$

# E  Efficiently Computable Criterion for Low Suboptimality in Achievable Accuracy

In this appendix, we provide the formal claims deferred from Section 4.2. Specifically, Proposition 2 shows that $\mathcal{D}_{\text{pop}}/\|\mathcal{D}_{\text{pop}}\|$ is, with high probability, well-approximated by the normalized empirical data tensor $\mathcal{D}_{\text{emp}}/\|\mathcal{D}_{\text{emp}}\|$. Corollary 2 establishes the efficiently computable criterion (low entanglement under canonical partitions for $\mathcal{D}_{\text{emp}}$), that with high probabilty is both necessary and sufficient for low suboptimality in achievable accuracy.

**Proposition 2.** *Consider the classification setting of Section 4.1, and let $\delta \in (0, 1)$ and $\gamma > 0$. If the training set size $M$ satisfies $M \geq \frac{2 \ln(\frac{2}{\delta})}{\|\mathcal{D}_{\text{pop}}\|^2 \gamma^2}$, then with probability at least $1 - \delta$:*

$$\left\| \frac{\mathcal{D}_{\text{pop}}}{\|\mathcal{D}_{\text{pop}}\|} - \frac{\mathcal{D}_{\text{emp}}}{\|\mathcal{D}_{\text{emp}}\|} \right\| \leq \gamma$$

*Proof sketch (proof in Appendix M.5).* A standard generalization of the Hoeffding inequality to random vectors in a Hilbert space yields a high probability bound on $\|\mathcal{D}_{\text{emp}} - \mathcal{D}_{\text{pop}}\|$, which is then translated to a bound on the normalized tensors. $\square$

---

**Algorithm 1** Entanglement Computation for the Empirical Data Tensor

---

1: **Input:** $\mathcal{X} := \big\{(\mathbf{x}^{(1,m)}, \ldots, \mathbf{x}^{(N,m)})\big\}_{m=1}^{M}$ — $M \in \mathbb{N}$ data instances comprising $N \in \mathbb{N}$ features each, $\mathcal{K} \subseteq [N]$ — subset of feature indices

2: **Output:** $QE(\mathcal{D}_{\mathrm{emp}}; \mathcal{K})$

---

3: Compute $\mathbf{G}^{(\mathcal{K})}, \mathbf{G}^{(\mathcal{K}^c)} \in \mathbb{R}^{M \times M}$ given element-wise by:

$$\forall i, j \in [M]: \ \mathbf{G}_{i,j}^{(\mathcal{K})} = y^{(i)} y^{(j)} \cdot \big\langle \otimes_{n \in \mathcal{K}} \mathbf{x}^{(n,i)}, \otimes_{n \in \mathcal{K}} \mathbf{x}^{(n,j)} \big\rangle = y^{(i)} y^{(j)} \cdot \prod_{n \in \mathcal{K}} \big\langle \mathbf{x}^{(n,i)}, \mathbf{x}^{(n,j)} \big\rangle$$

$$\forall i, j \in [M]: \ \mathbf{G}_{i,j}^{(\mathcal{K}^c)} = \big\langle \otimes_{n \in \mathcal{K}^c} \mathbf{x}^{(n,i)}, \otimes_{n \in \mathcal{K}^c} \mathbf{x}^{(n,j)} \big\rangle = \prod_{n \in \mathcal{K}^c} \big\langle \mathbf{x}^{(n,i)}, \mathbf{x}^{(n,j)} \big\rangle$$

4: Compute eigenvalue decompositions of $\mathbf{G}^{(\mathcal{K})}$ and $\mathbf{G}^{(\mathcal{K}^c)}$, *i.e.*:

$$\mathbf{G}^{(\mathcal{K})} = \mathbf{U}^{(\mathcal{K})} \mathbf{S}^{(\mathcal{K})} \big(\mathbf{U}^{(\mathcal{K})}\big)^{\top}$$

$$\mathbf{G}^{(\mathcal{K}^c)} = \mathbf{U}^{(\mathcal{K}^c)} \mathbf{S}^{(\mathcal{K}^c)} \big(\mathbf{U}^{(\mathcal{K}^c)}\big)^{\top}$$

where $\mathbf{U}^{(\mathcal{K})}, \mathbf{U}^{(\mathcal{K}^c)} \in \mathbb{R}^{M \times M}$ are orthogonal matrices and $\mathbf{S}^{(\mathcal{K})}, \mathbf{S}^{(\mathcal{K}^c)} \in \mathbb{R}^{M \times M}$ are diagonal holding the eigenvalues of $\mathbf{G}^{(\mathcal{K})}$ and $\mathbf{G}^{(\mathcal{K}^c)}$, respectively

5: Compute $\mathbf{Q} = \big(\mathbf{S}^{(\mathcal{K})}\big)^{\frac{1}{2}} \big(\mathbf{U}^{(\mathcal{K})}\big)^{\top} \mathbf{U}^{(\mathcal{K}^c)} \big(\mathbf{S}^{(\mathcal{K}^c)}\big)^{\frac{1}{2}} \in \mathbb{R}^{M \times M}$

6: Compute a singular value decomposition of $\mathbf{Q}$ to obtain its singular values $\sigma_1(\mathbf{Q}), \ldots, \sigma_M(\mathbf{Q})$

7: Let $\rho_m := \sigma_m^2(\mathbf{Q}) / \sum_{m'=1}^{M} \sigma_{m'}^2(\mathbf{Q})$ for $m \in [M]$

8: **return** $QE(\mathcal{D}_{\mathrm{emp}}; \mathcal{K}) = -\sum_{m=1}^{M} \rho_m \ln(\rho_m)$ (if $\mathbf{Q} = 0$, then return 0)

---

**Corollary 2.** *Consider the setting and notation of Corollary 1, with $\epsilon \in (0, 1/6]$. For $\delta \in (0,1)$, suppose that the training set size $M$ satisfies $M \geq \frac{8 \ln(\frac{2}{\delta})}{\|\mathcal{D}_{\mathrm{pop}}\|^2 \epsilon^2}$. Then, with probability at least $1 - \delta$ the following hold. First, if there exists a canonical partition $(\mathcal{K}, \mathcal{K}^c) \in \mathcal{C}_N$ (Definition 2) under which $QE(\mathcal{D}_{\mathrm{emp}}; \mathcal{K}) > \ln(R) + 3\epsilon \cdot \ln(D_\mathcal{K}) + 2\sqrt{3}\epsilon$, then:*

$$\mathrm{SubOpt} > \epsilon.$$

*Second, if for all $(\mathcal{K}, \mathcal{K}^c) \in \mathcal{C}_N$ it holds that $QE(\mathcal{D}_{\mathrm{emp}}; \mathcal{K}) \leq \frac{\epsilon^2}{32N - 48} \cdot \ln(R)$, then:*

$$\mathrm{SubOpt} \leq \epsilon.$$

*Moreover, the conditions above on the entanglements of $\mathcal{D}_{\mathrm{emp}}$ can be evaluated efficiently (in $\mathcal{O}(DNM^2 + NM^3)$ time $\mathcal{O}(DNM + M^2)$ and memory, where $D := \max_{n \in [N]} D_n$).*

*Proof sketch (proof in Appendix M.6).* Implied by Corollary 1, Proposition 2 with $\gamma = \frac{\epsilon}{2}$ and Algorithm 1 in Appendix F. $\qquad\square$

## F  Efficiently Computing Entanglements of the Empirical Data Tensor

For a given tensor, its entanglement with respect to a partition of axes (Definition 1) is determined by the singular values of its arrangement as a matrix according to the partition. Since the empirical data tensor $\mathcal{D}_{\mathrm{emp}}$ (Equation (2)) has size exponential in the number of features $N$, it is infeasible to naively compute its entanglement (or even explicitly store it in memory). Fortunately, as shown in [40], the specific form of the empirical data tensor admits an efficient algorithm for computing entanglements, without explicitly manipulating an exponentially large matrix. Specifically, the algorithm runs in $\mathcal{O}(DNM^2 + M^3)$ time and requires $\mathcal{O}(DNM + M^2)$ memory, where $D := \max_{n \in [N]} D_n$ is the maximal feature dimension (*i.e.* axis length of $\mathcal{D}_{\mathrm{emp}}$), $N$ is the number of features in the data (*i.e.* number of axes that $\mathcal{D}_{\mathrm{emp}}$ has), and $M$ is the number of training instances. For completeness, we outline the method in Algorithm 1 while referring the interested reader to Appendix A in [40] for further details.

# G   Practical Algorithm via Surrogate for Entanglement

To efficiently implement the scheme from Section 5.1, we replace entanglement with a surrogate measure of dependence. The surrogate is based on the Pearson correlation coefficient for multivariate features [45],[9] and its agreement with entanglement is demonstrated empirically in Appendix I. Theoretically supporting this agreement is left for future work.

**Definition 4.**   Given a set of $M \in \mathbb{N}$ instances $\mathcal{X} := \{(\mathbf{x}^{(1,m)}, \ldots, \mathbf{x}^{(N,m)}) \in (\mathbb{R}^D)^N\}_{m=1}^M$, denote by $p_{n,n'}$ the multivariate Pearson correlation between features $n, n' \in [N]$. For $\mathcal{K} \subseteq [N]$, the *surrogate entanglement* of $\mathcal{X}$ with respect to the partition $(\mathcal{K}, \mathcal{K}^c)$, denoted $SE(\mathcal{X}; \mathcal{K})$, is the sum of absolute values of Pearson correlation coefficients between pairs of features, the first belonging to $\mathcal{K}$ and the second to $\mathcal{K}^c := [N] \setminus \mathcal{K}$. That is, $SE(\mathcal{X}; \mathcal{K}) := \sum_{n \in \mathcal{K}, n' \in \mathcal{K}^c} |p_{n,n'}|$.

As shown by Proposition 3 below, replacing entanglement with surrogate entanglement in the scheme from Section 5.1 converts each search for a balanced partition minimizing average entanglement into a *minimum balanced cut problem*. Although the minimum balanced cut problem is NP-hard (see, *e.g.*, [23]), it enjoys a wide array of well-established approximation tools, particularly ones designed for large scale [29, 57]. We therefore obtain a practical algorithm for enhancing the suitability of a data distribution to locally connected neural networks — see Algorithm 2.

**Proposition 3.**   *For any $\bar{\mathcal{K}} \subseteq [N]$ of even size, the following optimization problem can be framed as a minimum balanced cut problem over a complete graph with $|\bar{\mathcal{K}}|$ vertices:*

$$\min_{\mathcal{K} \subset \bar{\mathcal{K}}, |\mathcal{K}| = |\bar{\mathcal{K}}|/2} \frac{1}{2} \Big[ SE(\mathcal{X}; \mathcal{K}) + SE(\mathcal{X}; \bar{\mathcal{K}} \setminus \mathcal{K}) \Big]. \tag{5}$$

*Specifically, there exists a complete undirected weighted graph with vertices $\bar{\mathcal{K}}$ and edge weights $w : \bar{\mathcal{K}} \times \bar{\mathcal{K}} \to \mathbb{R}$ such that for any $\mathcal{K} \subset \bar{\mathcal{K}}$, the weight of the cut in the graph induced by $\mathcal{K}$ — $\sum_{n \in \mathcal{K}, n' \in \bar{\mathcal{K}} \setminus \mathcal{K}} w(\{n, n'\})$ — is equal, up to an additive constant, to the term minimized in Equation (5), i.e. to $\frac{1}{2} \big[ SE(\mathcal{X}; \mathcal{K}) + SE(\mathcal{X}; \bar{\mathcal{K}} \setminus \mathcal{K}) \big]$.*

*Proof.*   Consider the complete undirected graph whose vertices are $\bar{\mathcal{K}}$ and where the weight of an edge $\{n, n'\} \in \bar{\mathcal{K}} \times \bar{\mathcal{K}}$ is $w(\{n, n'\}) = |p_{n,n'}|$ (recall that $p_{n,n'}$ stands for the multivariate Pearson correlation between features $n$ and $n'$ in $\mathcal{X}$). For any $\mathcal{K} \subset \bar{\mathcal{K}}$ it holds that:

$$\sum_{n \in \mathcal{K}, n' \in \bar{\mathcal{K}} \setminus \mathcal{K}} w(\{n, n'\}) = \frac{1}{2} \Big[ SE(\mathcal{X}; \mathcal{K}) + SE(\mathcal{X}; \bar{\mathcal{K}} \setminus \mathcal{K}) \Big] - \frac{1}{2} SE(\mathcal{X}; \bar{\mathcal{K}}),$$

where $\frac{1}{2} SE(\mathcal{X}; \bar{\mathcal{K}})$ does not depend on $\mathcal{K}$. This concludes the proof.   $\square$

# H   Definition of the Multivariate Pearson Correlation from [45]

Appendix G introduces a surrogate measure for entanglement based on the multivariate Pearson correlation from [45]. For completeness, this appendix provides its formal definition.

Given a set of $M \in \mathbb{N}$ instances $\mathcal{X} := \{(\mathbf{x}^{(1,m)}, \ldots, \mathbf{x}^{(N,m)}) \in (\mathbb{R}^D)^N\}_{m=1}^M$, let $\Sigma^{(n)}$ be the empirical covariance matrix of feature $n \in [N]$ and $\Sigma^{(n,n')}$ be the empirical cross-covariance matrix of features $n, n' \in [N]$, *i.e.*:

$$\Sigma^{(n)} := \frac{1}{M} \sum_{m-1}^M \big( \mathbf{x}^{(n,m)} - \mu^{(n)} \big) \otimes \big( \mathbf{x}^{(n,m)} - \mu^{(n)} \big),$$

$$\Sigma^{(n,n')} := \frac{1}{M} \sum_{m-1}^M \big( \mathbf{x}^{(n,m)} - \mu^{(n)} \big) \otimes \big( \mathbf{x}^{(n',m)} - \mu^{(n')} \big),$$

where $\mu^{(n)} := \frac{1}{M} \sum_{m=1}^M \mathbf{x}^{(n,m)}$ for $n \in [N]$. With this notation, the multivariate Pearson correlation of $n, n' \in [N]$ from [45] is defined by $p_{n,n'} := \mathrm{trace}\big( \Sigma^{(n,n')} \big) / \mathrm{trace}\big( (\Sigma^{(n)} \Sigma^{(n')})^{1/2} \big)$.

---

[9]For completeness, Appendix H provides a formal definition of the multivariate Pearson correlation.

---
**Algorithm 2** Enhancing Suitability of Data to Locally Connected Neural Networks
---
1: **Input:** $\mathcal{X} := \{(\mathbf{x}^{(1,m)}, \dots, \mathbf{x}^{(N,m)})\}_{m=1}^{M}$ — $M \in \mathbb{N}$ data instances comprising $N \in \mathbb{N}$ features

2: **Output:** Permutation $\pi : [N] \to [N]$ to apply to feature indices

---
3: Let $\mathcal{K}_{0,1} := [N]$ and denote $L := \log_2(N)$

4: # We assume for simplicity that $N$ is a power of two, otherwise one may add constant features

5: **for** $l = 0, \dots, L - 1$ , $n = 1, \dots, 2^l$ **do**

6:     Using a reduction to a minimum balanced cut problem (Proposition 3), find an approximate solution $\mathcal{K}_{l+1,2n-1} \subset \mathcal{K}_{l,n}$ for:

$$\min_{\mathcal{K} \subset \mathcal{K}_{l,n}, |\mathcal{K}| = |\mathcal{K}_{l,n}|/2} \frac{1}{2}[SE(\mathcal{X}; \mathcal{K}) + SE(\mathcal{X}; \mathcal{K}_{l,n} \setminus \mathcal{K})]$$

7:     Let $\mathcal{K}_{l+1,2n} := \mathcal{K}_{l,n} \setminus \mathcal{K}_{l+1,2n-1}$

8: **end for**

9: # At this point, $\mathcal{K}_{L,1}, \dots, \mathcal{K}_{L,N}$ each contain a single feature index

10: **return** $\pi$ that maps $k \in \mathcal{K}_{L,n}$ to $n$, for every $n \in [N]$
---

# I    Entanglement and Surrogate Entanglement Are Strongly Correlated

In Appendix G, we introduced a surrogate entanglement measure (Definition 4) to facilitate efficient implementation of the feature arrangement search scheme from Section 5.1. Figure 5 supports the viability of the chosen surrogate, demonstrating empirically that it is strongly correlated with entanglement of the empirical data tensor (Definition 1 and Equation (2)).

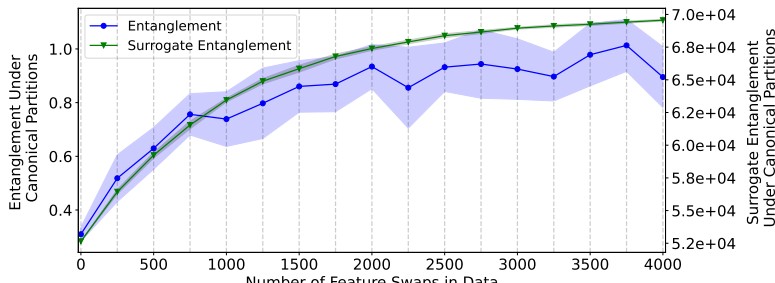

Figure 5: Surrogate entanglement (Definition 4) is strongly correlated with the entanglement (Definition 1) of the empirical data tensor. Presented are average entanglement and average surrogate entanglement under canonical partitions, admitted by the Speech Commands audio datasets [64] considered in Figure 3. Remarkably, the Pearson correlation between the quantities is 0.974. For further details see caption of Figure 3 as well as Appendix L.

# J    Extension to Arbitrary Dimensional Models and Data

In this appendix, we extend our theoretical analysis and experiments, including the algorithm for enhancing the suitability of data to locally connected neural networks, from one-dimensional (sequential) models and data to $P$-dimensional models and data (such as two-dimensional image data or three-dimensional video data), for $P \in \mathbb{N}$. Specifically, Appendix J.1 extends Section 3, Appendix J.2 extends Section 4 and Appendix J.3 extends Section 5.

To ease presentation, we consider $P$-dimensional data instances whose feature vectors are associated with coordinates $(n_1, \dots, n_P) \in [N]^P$, where $N = 2^L$ for some $L \in \mathbb{N}$ (if this is not the case we may add constant features as needed).

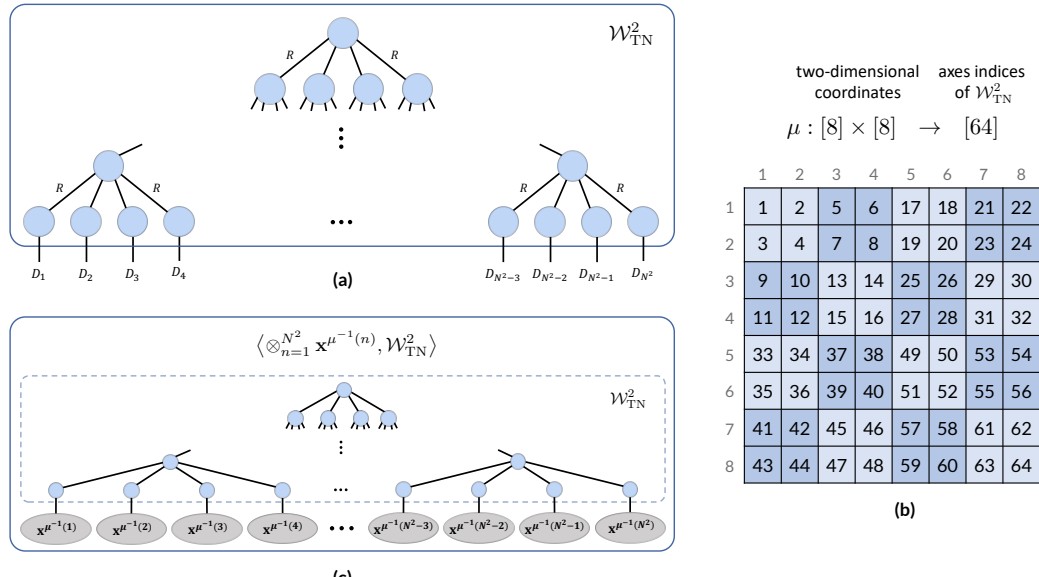

Figure 6: The analyzed tensor network equivalent to a locally connected neural network operating over $P$-dimensional data, for $P = 2$. **(a)** The tensor network adheres to a perfect $2^P$-ary tree connectivity with $N^P$ leaf nodes, where $N = 2^L$ for some $L \in \mathbb{N}$, and generates $\mathcal{W}_{\text{TN}}^P \in \mathbb{R}^{D_1 \times \cdots \times D_{N^P}}$. Axes corresponding to open edges are indexed such that open edges descendant to any node of the tree have contiguous indices. The lengths of axes corresponding to inner (non-open) edges are equal to $R \in \mathbb{N}$, referred to as the width of the tensor network. **(b)** Exemplar $\mu : [N]^P \to [N^P]$ compatible with the locally connected tensor network (Definition 5), mapping $P$-dimensional coordinates to axes indices of $\mathcal{W}_{\text{TN}}^P$. **(c)** Contracting $\mathcal{W}_{\text{TN}}^P$ with vectors $\{\mathbf{x}^{(n_1,\dots,n_P)}\}_{n_1,\dots,n_P \in [N]}$ according to a compatible $\mu$ produces $\langle \otimes_{n=1}^{N^P} \mathbf{x}^{\mu^{-1}(n)}, \mathcal{W}_{\text{TN}}^P \rangle$. Performing these contractions can be viewed as a forward pass of a certain locally connected neural network (with polynomial non-linearity) over the data instance $\{\mathbf{x}^{(n_1,\dots,n_P)}\}_{n_1,\dots,n_P \in [N]}$ (see, *e.g.*, [12, 10, 35, 49]).

## J.1 Low Entanglement Under Canonical Partitions Is Necessary and Sufficient for Fitting Tensor

We introduce the locally connected tensor network equivalent to a locally connected neural network that operates over $P$-dimensional data (Appendix J.1.1). Subsequently, we establish a necessary and sufficient condition required for it to fit a given tensor (Appendix J.1.2), generalizing the results of Section 3.2.

### J.1.1 Tensor Network Equivalent to a Locally Connected Neural Network

For $P$-dimensional data, the locally connected tensor network we consider (defined in Section 3.1 for one-dimensional data) has an underlying perfect $2^P$-ary tree graph of height $L$. We denote the tensor it generates by $\mathcal{W}_{\text{TN}}^P \in \mathbb{R}^{D_1 \times \cdots \times D_{N^P}}$. Figure 6(a) provides its diagrammatic definition. As in the one-dimensional case, the lengths of axes corresponding to inner edges are taken to be $R \in \mathbb{N}$, referred to as the width of the tensor network.

The axes of $\mathcal{W}_{\text{TN}}^P$ are associated with $P$-dimensional coordinates through a bijective function $\mu : [N]^P \to [N^P]$.

**Definition 5.** We say that a bijective function $\mu : [N]^P \to [N^P]$ is *compatible* with the locally connected tensor network if, for any node in the tensor network, the coordinates mapped to indices of $\mathcal{W}_{\text{TN}}^P$'s axes descendant to that node form a contiguous $P$-dimensional cubic block in $[N]^P$ (*e.g.*, square block when $P = 2$) — see Figure 6(b) for an illustration. With slight abuse of notation, for $\mathcal{K} \subseteq [N]^P$ we denote $\mu(\mathcal{K}) := \{\mu(n_1, \dots, n_P) : (n_1, \dots, n_P) \in \mathcal{K}\} \subseteq [N^P]$.

Contracting the locally connected tensor network with $\{\mathbf{x}^{(n_1,\dots,n_P)} \in \mathbb{R}^{D_{\mu(n_1,\dots,n_P)}}\}_{n_1,\dots,n_P \in [N]}$ according to a compatible $\mu$, as depicted in Figure 6(c), can be viewed as a forward pass of the data instance $\{\mathbf{x}^{(n_1,\dots,n_P)}\}_{n_1,\dots,n_P \in [N]}$ through a locally connected neural network (with polynomial non-linearity), which produces the scalar $\langle \otimes_{n=1}^{N^P} \mathbf{x}^{\mu^{-1}(n)}, \mathcal{W}_{\text{TN}}^P \rangle$ (see, *e.g.*, [12, 10, 35, 49]).

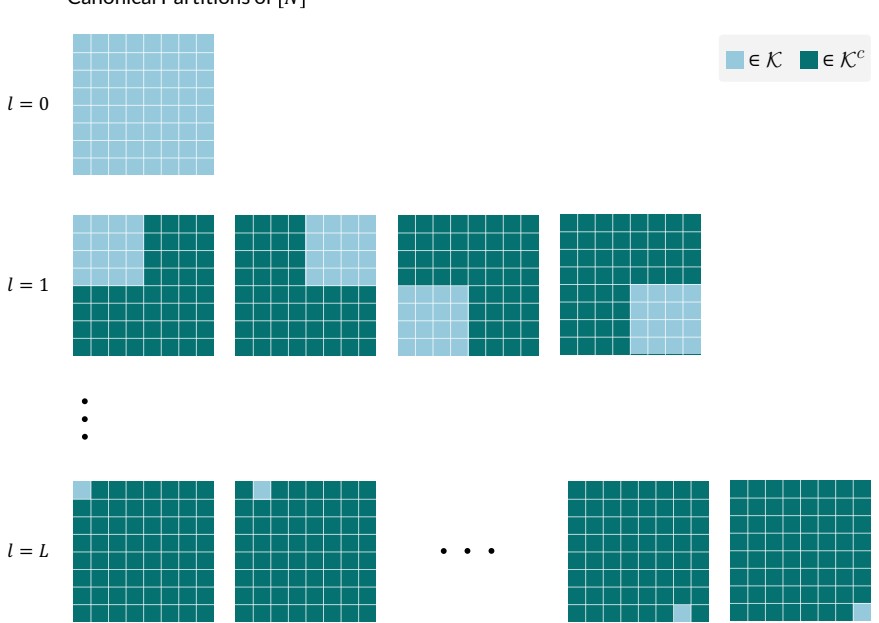

**Figure 7:** The canonical partitions of $[N]^P$, for $P = 2$ and $N = 2^L$ with $L \in \mathbb{N}$. Every $l \in \{0, \ldots, L\}$ contributes $2^{l \cdot P}$ canonical partitions, each induced by $\mathcal{K} = \times_{p=1}^{P} \{2^{L-l} \cdot (n_p - 1) + 1, \ldots, 2^{L-l} \cdot n_p\}$ for $n_1, \ldots, n_P \in [2^l]$.

### J.1.2 Necessary and Sufficient Condition for Fitting Tensor

The ability of the locally connected tensor network, defined in Appendix J.1.1, to fit (*i.e.* represent) a tensor is determined by the entanglements that the tensor admits under partitions of its axes, induced by the following canonical partitions of $[N]^P$.

**Definition 6.** The *canonical partitions* of $[N]^P$, illustrated in Figure 7 for $P = 2$, are:[10]

$$\mathcal{C}_N^P := \Big\{ (\mathcal{K}, \mathcal{K}^c) : \mathcal{K} = \times_{p=1}^{P} \big\{ 2^{L-l} \cdot (n_p - 1) + 1, \ldots, 2^{L-l} \cdot n_p \big\},$$

$$l \in \big\{0, \ldots, L\big\}, \; n_1, \ldots, n_P \in \big[2^l\big] \Big\}.$$

With the definition of canonical partitions for $P$-dimensional data in place, Theorem 3 generalizes Theorem 1. In particular, suppose that $\mathcal{W}_{\text{TN}}^P$ — the tensor generated by the locally connected tensor network — well-approximates $\mathcal{A} \in \mathbb{R}^{D_1 \times \cdots \times D_{N^P}}$. Then, given a compatible $\mu : [N]^P \to [N^P]$ (Definition 5), Theorem 3 establishes that the entanglement of $\mathcal{A}$ with respect to $(\mu(\mathcal{K}), \mu(\mathcal{K})^c)$, where $(\mathcal{K}, \mathcal{K}^c) \in \mathcal{C}_N^P$, cannot be much larger than $\ln(R)$, whereas the expected entanglement attained by a random tensor with respect to $(\mu(\mathcal{K}), \mu(\mathcal{K})^c)$ is on the order of $\min\{|\mathcal{K}|, |\mathcal{K}^c|\}$ (which is linear in $N^P$ for some canonical partitions).

In the other direction, Theorem 4 implies that low entanglement under partitions of axes induced by canonical partitions of $[N]^P$ is not only necessary for a tensor to be fit by the locally connected tensor network, but also sufficient.

**Theorem 3.** *Let $\mathcal{W}_{\text{TN}}^P \in \mathbb{R}^{D_1 \times \cdots \times D_{N^P}}$ be a tensor generated by the locally connected tensor network defined in Appendix J.1.1, and $\mu : [N]^P \to [N^P]$ be a compatible map from $P$-dimensional coordinates to axes indices of $\mathcal{W}_{\text{TN}}^P$ (Definition 5). For any $\mathcal{A} \in \mathbb{R}^{D_1 \times \cdots \times D_{N^P}}$ and $\epsilon \in [0, \|\mathcal{A}\|/4]$, if $\|\mathcal{W}_{\text{TN}}^P - \mathcal{A}\| \leq \epsilon$, then for all canonical partitions $(\mathcal{K}, \mathcal{K}^c) \in \mathcal{C}_N^P$ (Definition 6):*

$$QE(\mathcal{A}; \mu(\mathcal{K})) \leq \ln(R) + \frac{2\epsilon}{\|\mathcal{A}\|} \cdot \ln(D_{\mu(\mathcal{K})}) + 2\sqrt{\frac{2\epsilon}{\|\mathcal{A}\|}}, \tag{6}$$

---

[10]For sets $\mathcal{S}_1, \ldots, \mathcal{S}_P$, we denote their Cartesian product by $\times_{p=1}^{P} \mathcal{S}_p$.

where $D_{\mu(\mathcal{K})} := \min\{\prod_{n\in\mu(\mathcal{K})} D_n, \prod_{n\in\mu(\mathcal{K})^c} D_n\}$. *In contrast, a random* $\mathcal{A}' \in \mathbb{R}^{D_1\times\cdots\times D_{N^P}}$, *drawn according to the uniform distribution over the set of unit norm tensors, satisfies for all canonical partitions* $(\mathcal{K}, \mathcal{K}^c) \in \mathcal{C}_N^P$:

$$\mathbb{E}[QE(\mathcal{A}'; \mu(\mathcal{K}))] \geq \min\{|\mathcal{K}|, |\mathcal{K}^c|\} \cdot \ln\big(\min_{n\in[N^P]} D_n\big) + \ln\left(\frac{1}{2}\right) - \frac{1}{2}. \tag{7}$$

*Proof sketch (proof in Appendix M.7).* The proof is analogous to that of Theorem 1. $\qquad\square$

**Theorem 4.** *Let* $\mathcal{A} \in \mathbb{R}^{D_1\times\cdots\times D_{N^P}}$ *and* $\epsilon > 0$. *Suppose that for all canonical partitions* $(\mathcal{K}, \mathcal{K}^c) \in \mathcal{C}_N^P$ *(Definition 6) it holds that* $QE(\mathcal{A}; \mu(\mathcal{K})) \leq \frac{\epsilon^2}{(2N^P-3)\|\mathcal{A}\|^2} \cdot \ln(R)$, *where* $\mu : [N]^P \to [N^P]$ *is compatible with the locally connected tensor network (Definition 5). Then, there exists an assignment for the tensors constituting the locally connected tensor network (defined in Appendix J.1.1) such that it generates* $\mathcal{W}_{TN}^P \in \mathbb{R}^{D_1\times\cdots\times D_{N^P}}$ *satisfying:*

$$\|\mathcal{W}_{TN}^P - \mathcal{A}\| \leq \epsilon.$$

*Proof sketch (proof in Appendix M.8).* The claim follows through a reduction from the locally connected tensor network for $P$-dimensional data to that for one-dimensional data (defined in Section 3.1), *i.e.* from perfect $2^P$-ary to perfect binary tree tensor networks. Specifically, we consider a modified locally connected tensor network for one-dimensional data, where axes corresponding to different inner edges can vary in length (as opposed to all having length $R$). We then show, by arguments analogous to those in the proof of Theorem 2, that it can approximate $\mathcal{A}$ while having certain inner axes, related to the canonical partitions of $[N]^P$, of lengths at most $R$. The proof concludes by establishing that, any tensor represented by such a locally connected tensor network for one-dimensional data can be represented via the locally connected tensor network for $P$-dimensional data (where the length of each axis corresponding to an inner edge is $R$). $\qquad\square$

## J.2 Low Entanglement Under Canonical Partitions Is Necessary and Sufficient for Accurate Prediction

In this appendix, we consider the locally connected tensor network from Appendix J.1.1 in a machine learning setting, and extend the results and experiments of Section 4 from one-dimensional to $P$-dimensional models and data.

### J.2.1 Accurate Prediction Is Equivalent to Fitting Data Tensor

The locally connected tensor network generating $\mathcal{W}_{TN}^P \in \mathbb{R}^{D_1\times\cdots\times D_{N^P}}$ is equivalent to a locally connected neural network operating over $P$-dimensional data (see Appendix J.1.1). Specifically, a forward pass of the latter over $\{\mathbf{x}^{(n_1,\ldots,n_P)} \in \mathbb{R}^{D_{\mu(n_1,\ldots,n_P)}}\}_{n_1,\ldots,n_P\in[N]}$ yields $\langle \otimes_{n=1}^{N^P} \mathbf{x}^{\mu^{-1}(n)}, \mathcal{W}_{TN}^P\rangle$, for a compatible $\mu : [N]^P \to [N^P]$ (Definition 5). Suppose we are given a training set of labeled instances $\big\{\{\mathbf{x}^{((n_1,\ldots,n_P),m)}\}_{n_1,\ldots,n_P\in[N]}, y^{(m)}\big\}_{m=1}^M$ drawn i.i.d. from some distribution, where $y^{(m)} \in \{1, -1\}$ for $m \in [M]$. Learning the parameters of the neural network through the soft-margin support vector machine (SVM) objective amounts to optimizing:

$$\min_{\|\mathcal{W}_{TN}^P\|\leq B} \frac{1}{M} \sum_{m=1}^M \max\Big\{0, 1 - y^{(m)}\big\langle \otimes_{n=1}^{N^P}\mathbf{x}^{(\mu^{-1}(n),m)}, \mathcal{W}_{TN}^P\big\rangle\Big\},$$

for a predetermined constant $B > 0$. This objective generalizes Equation (1) from one-dimensional to $P$-dimensional model and data. Assume that instances are normalized, *i.e.* $\|\mathbf{x}^{((n_1,\ldots,n_P),m)}\| \leq 1$ for all $n_1,\ldots,n_P \in [N], m \in [M]$, and that $B \leq 1$. By a derivation analogous to that succeeding Equation (1) in Section 4.1, if follows that minimizing the SVM objective is equivalent to minimizing $\big\|\frac{\mathcal{W}_{TN}^P}{\|\mathcal{W}_{TN}^P\|} - \frac{\mathcal{D}_{emp}^P}{\|\mathcal{D}_{emp}^P\|}\big\|$, where:

$$\mathcal{D}_{emp}^P := \frac{1}{M}\sum_{m=1}^M y^{(m)} \cdot \otimes_{n=1}^{N^P}\mathbf{x}^{(\mu^{-1}(n),m)} \tag{8}$$

extends the notion of *empirical data tensor* to $P$-dimensional data. In other words, the accuracy achievable over the training data is determined by the extent to which $\frac{\mathcal{W}_{TN}^P}{\|\mathcal{W}_{TN}^P\|}$ can fit the normalized empirical data tensor $\frac{\mathcal{D}_{emp}^P}{\|\mathcal{D}_{emp}^P\|}$.

The same arguments apply to the population loss, in which case $\mathcal{D}^P_{\text{emp}}$ is replaced by the *population data tensor*:

$$\mathcal{D}^P_{\text{pop}} := \mathbb{E}_{\{\mathbf{x}^{(n_1,\ldots,n_P)}\}_{n_1,\ldots,n_P \in [N]},y}\left[y \cdot \otimes_{n=1}^{N^P} \mathbf{x}^{\mu^{-1}(n)}\right]. \tag{9}$$

The achievable accuracy over the population is therefore determined by the extent to which $\frac{\mathcal{W}^P_{\text{TN}}}{\|\mathcal{W}^P_{\text{TN}}\|}$ can fit the normalized population data tensor $\frac{\mathcal{D}^P_{\text{pop}}}{\|\mathcal{D}^P_{\text{pop}}\|}$. Accordingly, we refer to the minimal distance from it as the *supobtimality in achievable accuracy*, generalizing Definition 3 from Section 4.1.

**Definition 7.** In the context of the classification setting above, the *suboptimality in achievable accuracy* is:

$$\text{SubOpt}^P := \min_{\mathcal{W}^P_{\text{TN}}}\left\|\frac{\mathcal{W}^P_{\text{TN}}}{\|\mathcal{W}^P_{\text{TN}}\|} - \frac{\mathcal{D}^P_{\text{pop}}}{\|\mathcal{D}^P_{\text{pop}}\|}\right\|.$$

### J.2.2 Necessary and Sufficient Condition for Accurate Prediction

In the classification setting of Appendix J.2.1, by invoking Theorems 3 and 4 from Appendix J.1.2, we conclude that the suboptimality in achievable accuracy is small if and only if the population (empirical) data tensor $\mathcal{D}^P_{\text{pop}}$ ($\mathcal{D}^P_{\text{emp}}$) admits low entanglement under the canonical partitions of features (Definition 6). Specifically, we establish Corollary 3, Proposition 4 and Corollary 4, which generalize Corollary 1, Proposition 2 and Corollary 2 from Section 4.2, respectively, to $P$-dimensional model and data.

**Corollary 3.** *Consider the classification setting of Appendix J.2.1, and let $\epsilon \in [0, 1/4]$. If there exists a canonical partition $(\mathcal{K}, \mathcal{K}^c) \in \mathcal{C}^P_N$ (Definition 6) under which $QE(\mathcal{D}^P_{pop}; \mu(\mathcal{K})) > \ln(R) + 2\epsilon \cdot \ln(D_{\mu(\mathcal{K})}) + 2\sqrt{2\epsilon}$, where $D_{\mu(\mathcal{K})} := \min\{\prod_{n\in\mu(\mathcal{K})} D_n, \prod_{n\in\mu(\mathcal{K})^c} D_n\}$, then:*

$$\text{SubOpt}^P > \epsilon.$$

*Conversely, if for all $(\mathcal{K}, \mathcal{K}^c) \in \mathcal{C}^P_N$ it holds that $QE(\mathcal{D}^P_{pop}; \mu(\mathcal{K})) \leq \frac{\epsilon^2}{8N^P - 12} \cdot \ln(R)$, then:*

$$\text{SubOpt}^P \leq \epsilon.$$

*Proof.* Implied by Theorems 3 and 4 after accounting for $\mathcal{W}^P_{\text{TN}}$ being normalized in the suboptimality in achievable accuracy, as done in the proof of Corollary 1. $\square$

**Proposition 4.** *Consider the classification setting of Appendix J.2.1, and let $\delta \in (0, 1)$ and $\gamma > 0$. If the training set size $M$ satisfies $M \geq \frac{2\ln(\frac{2}{\delta})}{\|\mathcal{D}^P_{pop}\|^2 \gamma^2}$, then with probability at least $1 - \delta$:*

$$\left\|\frac{\mathcal{D}^P_{pop}}{\|\mathcal{D}^P_{pop}\|} - \frac{\mathcal{D}^P_{emp}}{\|\mathcal{D}^P_{emp}\|}\right\| \leq \gamma$$

*Proof.* The claim is established by following steps identical to those in the proof of Proposition 2. $\square$

**Corollary 4.** *Consider the setting and notation of Corollary 3, with $\epsilon \in (0, \frac{1}{6}]$. For $\delta \in (0, 1)$, suppose that the training set size $M$ satisfies $M \geq \frac{8\ln(\frac{2}{\delta})}{\|\mathcal{D}^P_{pop}\|^2 \epsilon^2}$ Then, with probability at least $1 - \delta$ the following hold. First, if there exists a canonical partition $(\mathcal{K}, \mathcal{K}^c) \in \mathcal{C}^P_N$ (Definition 6) under which $QE(\mathcal{D}^P_{emp}; \mu(\mathcal{K})) > \ln(R) + 3\epsilon \cdot \ln(D_{\mu(\mathcal{K})}) + 2\sqrt{3\epsilon}$, then:*

$$\text{SubOpt}^P > \epsilon.$$

*Second, if for all $(\mathcal{K}, \mathcal{K}^c) \in \mathcal{C}^P_N$ it holds that $QE(\mathcal{D}^P_{emp}; \mu(\mathcal{K})) \leq \frac{\epsilon^2}{32N^P - 48} \cdot \ln(R)$, then:*

$$\text{SubOpt}^P \leq \epsilon.$$

*Moreover, the conditions above on the entanglements of $\mathcal{D}^P_{emp}$ can be evaluated efficiently (in $\mathcal{O}(DN^P M^2 + N^P M^3)$ time $\mathcal{O}(DN^P M + M^2)$ and memory, where $D := \max_{n\in[N^P]} D_n$).*

*Proof.* Implied from Corollary 3, Proposition 4 with $\gamma = \frac{\epsilon}{2}$ and Algorithm 1 in Appendix F by following steps identical to those in the proof of Corollary 2. $\square$

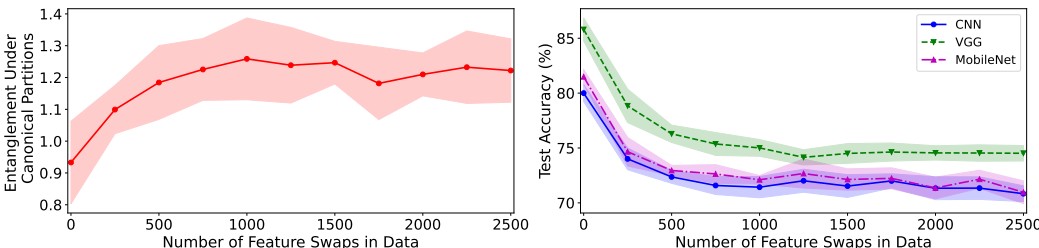

Figure 8: The prediction accuracies of convolutional neural networks are inversely correlated with the entanglements of image data under canonical partitions of features, in compliance with our theory (Appendices J.1 and J.2). This figure is identical to Figure 3, except that the measurements were carried over a binary classification version of the CIFAR10 image dataset, as opposed to a one-dimensional (sequential) audio dataset, using three different convolutional neural network architectures. For further details see caption of Figure 3 as well as Appendix L.

### J.2.3 Empirical Demonstration

Figure 8 extends the experiments reported by Figure 3 in Section 4.3 from one-dimensional (sequential) audio data to two-dimensional image data. Specifically, it demonstrates that the prediction accuracies of convolutional neural networks over a variant of CIFAR10 [33] is inversely correlated with the entanglements that the data admits under canonical partitions of features (Definition 6), *i.e.* with the entanglements of the empirical data tensor under partitions of its axes induced by canonical partitions.

### J.3 Enhancing Suitability of Data to Locally Connected Neural Networks

We extend the preprocessing algorithm from Section 5, aimed to enhance the suitability of a data distribution to locally connected neural networks, from one-dimensional to $P$-dimensional models and data (Appendices J.3.1 and J.3.2). Empirical evaluations demonstrate that it significantly improves prediction accuracy of common architectures (Appendix J.3.3).

### J.3.1 Search for Feature Arrangement With Low Entanglement Under Canonical Partitions

Suppose we have $M \in \mathbb{N}$ training instances $\left\{\{\mathbf{x}^{((n_1,...,n_P),m)}\}_{n_1,...,n_P \in [N]}, y^{(m)}\right\}_{m=1}^M$, where $\mathbf{x}^{((n_1,...,n_P),m)} \in \mathbb{R}^D$ and $y^{(m)} \in \{1, -1\}$ for $n_1, \ldots, n_P \in [N], m \in [M]$, with $D \in \mathbb{N}$. For models intaking $P$-dimensional data, the recipe for enhancing the suitability of a data distribution to locally connected neural networks from Section 5.1 boils down to finding a permutation $\pi : [N]^P \to [N]^P$, which when applied to feature coordinates leads the empirical data tensor $\mathcal{D}_{\text{emp}}^P$ (Equation (8)) to admit low entanglement under canonical partitions (Definition 6).[11]

A greedy realization, analogous to that outlined in Section 5.1, is as follows. Initially, partition the features into $2^P$ equally sized disjoint sets $\{\mathcal{K}_{1,(k_1,...,k_P)} \subset [N]^P\}_{k_1,...,k_P \in [2]}$ such that the average of $\mathcal{D}_{\text{emp}}^P$'s entanglements induced by these sets is minimal. That is, find an element of:

$$\underset{\substack{\{\mathcal{K}'_{k_1,...,k_P} \subset [N]^P\}_{k_1,...,k_P \in [2]} \\ \text{s.t. } \cup_{k_1,...,k_P \in [2]} \mathcal{K}'_{k_1,...,k_P} = [N]^P, \\ \forall k_1,...,k_P,k'_1,...,k'_P \in [2] \; |\mathcal{K}'_{k_1,...,k_P}| = |\mathcal{K}'_{k'_1,...,k'_P}|}}{\operatorname{argmin}} \frac{1}{2^P} \sum_{k_1,...,k_P \in [2]} QE\left(\mathcal{D}_{\text{emp}}^P; \mu(\mathcal{K}'_{k_1,...,k_P})\right),$$

where $\mu : [N]^P \to [N^P]$ is a compatible map from coordinates in $[N]^P$ to axes indices (Definition 5). The permutation $\pi$ will map each $\mathcal{K}_{1,(k_1,...,k_P)}$ to coordinates $\times_{p=1}^P \{\frac{N}{2} \cdot (k_p - 1) + 1, \ldots, \frac{N}{2} \cdot k_p\}$, for $k_1, \ldots, k_P \in [2]$. Then, partition similarly each of $\{\mathcal{K}_{1,(k_1,...,k_P)}\}_{k_1,...,k_P \in [2]}$ into $2^P$ equally sized disjoint sets. Continuing in the same fashion, until we reach subsets $\{\mathcal{K}_{L,(k_1,...,k_P)}\}_{k_1,...,k_P \in [N]}$ consisting of a single feature coordinate each, fully specifies the permutation $\pi$.

---

[11]Enhancing the suitability of a data distribution with instances of dimension different than $P$ to $P$-dimensional models is possible by first arbitrarily mapping features to coordinates in $[N]^P$, and then following the scheme for rearranging $P$-dimensional data.

**Algorithm 3** Enhancing Suitability of $P$-Dimensional Data to Locally Connected Neural Networks

---

1: **Input:** $\mathcal{X} := \left\{ \{ \mathbf{x}^{((n_1,\ldots,n_P),m)} \}_{n_1,\ldots,n_P \in [N]} \right\}_{m=1}^{M}$ — $M \in \mathbb{N}$ data instances comprising $N^P$ features each

2: **Output:** Permutation $\pi : [N]^P \to [N]^P$ to apply to feature coordinates

---

3: Let $\mathcal{K}_{0,(1,\ldots,1)} := [N]^P$ and denote $L := \log_2(N)$

4: # We assume for simplicity that $N$ is a power of two, otherwise one may add constant features

5: **for** $l = 0,\ldots,L-1$ , $k_1,\ldots,k_P = 1,\ldots,2^l$ **do**

6: Using a reduction to a minimum balanced $2^P$-cut problem (Proposition 5), find an approximate solution $\{\mathcal{K}_{l+1,(2k_1-i_1,\ldots,2k_P-i_P)} \subset \mathcal{K}_{l,(k_1,\ldots,k_P)}\}_{i_1,\ldots,i_P \in \{0,1\}}$ for:

$$
\min_{\substack{\{\mathcal{K}_{k_1',\ldots,k_P'} \subset \mathcal{K}_{l,(k_1,\ldots,k_P)}\}_{k_1',\ldots,k_P' \in [2]} \\ \text{s.t. } \cup_{k_1',\ldots,k_P' \in [2]} \mathcal{K}_{k_1',\ldots,k_P'} = \mathcal{K}_{l,(k_1,\ldots,k_P)}, \\ \forall k_1',\ldots,k_P',\hat{k}_1,\ldots,\hat{k}_P \in [2] \; |\mathcal{K}_{k_1',\ldots,k_P'}| = |\mathcal{K}_{\hat{k}_1,\ldots,\hat{k}_P}|}} \frac{1}{2^P} \sum_{k_1',\ldots,k_P' \in [2]} SE\big(\mathcal{X}; \mathcal{K}_{k_1',\ldots,k_P'}\big)
$$

7: **end for**

8: # At this point, $\{\mathcal{K}_{L,(k_1,\ldots,k_P)}\}_{k_1,\ldots,k_P \in [N]}$ each contain a single feature coordinate

9: **return** $\pi$ that maps $(n_1,\ldots,n_P) \in \mathcal{K}_{L,(k_1,\ldots,k_P)}$ to $(k_1,\ldots,k_P)$, for every $k_1,\ldots,k_P \in [N]$

---

As in the case of one-dimensional models and data (Section 5.1), the step lying at the heart of the above scheme — finding a balanced partition into $2^P$ sets that minimizes average entanglement — is computationally prohibitive. In the next supappendix we will see that replacing entanglement with surrogate entanglement brings forth a practical implementation.

### J.3.2 Practical Algorithm via Surrogate for Entanglement

To efficiently implement the scheme from Appendix J.3.1, we replace entanglement with surrogate entanglement (Definition 4), which for $P$-dimensional data is straightforwardly defined as follows.

**Definition 8.** Given a set of $M \in \mathbb{N}$ instances $\mathcal{X} := \left\{ \{ \mathbf{x}^{((n_1,\ldots,n_P),m)} \in \mathbb{R}^D \}_{n_1,\ldots,n_P \in [N]} \right\}_{m=1}^{M}$, denote by $p_{(n_1,\ldots,n_P),(n_1',\ldots,n_P')}$ the multivariate Pearson correlation between features $(n_1,\ldots,n_P) \in [N]^P$ and $(n_1',\ldots,n_P') \in [N]^P$. For $\mathcal{K} \subseteq [N]^P$, the *surrogate entanglement* of $\mathcal{X}$ with respect to the partition $(\mathcal{K},\mathcal{K}^c)$, denoted $SE(\mathcal{X};\mathcal{K})$, is the sum of Pearson correlation coefficients between pairs of features, the first belonging to $\mathcal{K}$ and the second to $\mathcal{K}^c := [N]^P \setminus \mathcal{K}$. That is:

$$
SE\big(\mathcal{X};\mathcal{K}\big) = \sum_{(n_1,\ldots,n_P) \in \mathcal{K}, (n_1',\ldots,n_P') \in \mathcal{K}^c} p_{(n_1,\ldots,n_P),(n_1',\ldots,n_P')} \, .
$$

Analogously to the case of one-dimensional data (Appendix G), Proposition 5 shows that replacing the entanglement with surrogate entanglement in the scheme from Appendix J.3.1 converts each search for a balanced partition minimizing average entanglement into a *minimum balanced $2^P$-cut problem*. Although the minimum balanced $2^P$-cut problem is NP-hard (see, *e.g.*, [23]), similarly to the minimum balanced (2-) cut problem, it enjoys a wide array of well-established approximation implementations, particularly ones designed for large scale [29, 57]. We therefore obtain a practical algorithm for enhancing the suitability of a data distribution with $P$-dimensional instances to locally connected neural networks — see Algorithm 3.

**Proposition 5.** *For any $\bar{\mathcal{K}} \subseteq [N]^P$ of size divisible by $2^P$, the following optimization problem can be framed as a minimum balanced $2^P$-cut problem over a complete graph with $|\bar{\mathcal{K}}|$ vertices:*

$$
\min_{\substack{\{\mathcal{K}_{k_1,\ldots,k_P} \subset \bar{\mathcal{K}}\}_{k_1,\ldots,k_P \in [2]} \\ \text{s.t. } \cup_{k_1,\ldots,k_P \in [2]} \mathcal{K}_{k_1,\ldots,k_P} = \bar{\mathcal{K}}, \\ \forall k_1,\ldots,k_P,k_1',\ldots,k_P' \in [2] \; |\mathcal{K}_{k_1,\ldots,k_P}| = |\mathcal{K}_{k_1',\ldots,k_P'}|}} \frac{1}{2^P} \sum_{k_1,\ldots,k_P \in [2]} SE(\mathcal{X}; \mathcal{K}_{k_1,\ldots,k_P}). \qquad (10)
$$

*Specifically, there exists a complete undirected weighted graph with vertices $\bar{\mathcal{K}}$ and edge weights $w : \bar{\mathcal{K}} \times \bar{\mathcal{K}} \to \mathbb{R}$ such that, for any partition of $\bar{\mathcal{K}}$ into equally sized dis-*

Table 3: Arranging features of randomly permuted image data via Algorithm 3 significantly improves the prediction accuracies of convolutional neural networks. Reported are the results of experiments analogous to those of Table 1, carried out over the CIFAR10 image dataset (as opposed to an audio dataset) using Algorithm 3. For further details see caption of Table 1 as well as Appendix L.

|  | Randomly Permuted | Algorithm 3 | IGTD |
| --- | --- | --- | --- |
| CNN | $35.1 \pm _{0.5}$ | $\mathbf{38.2} \pm _{0.4}$ | $36.2 \pm _{0.7}$ |

Table 4: Our feature rearrangement method (Algorithm 2 in Appendix G) can be efficiently applied to data with a large number of features. Reported are the results of an experiment identical to that of Table 1, but over a version of the Speech Commands [64] dataset in which every instance has 50,000 features. Instances of the minimum balanced cut problem solved as part of Algorithm 2 entail graphs with up to $25 \cdot 10^8$ edges. They are solved using the well-known edge sparsification algorithm of [57] that preserves weights of cuts, allowing for configurable compute and memory consumption (the more resources are consumed, the more accurate the solution will be). As can be seen, Algorithm 2 leads to significant improvements in prediction accuracies. See Appendix L for further implementation details.

|  | Randomly Permuted | Our Method |
| --- | --- | --- |
| CNN | $15.0 \pm _{1.6}$ | $\mathbf{65.6} \pm _{1.1}$ |
| S4 | $18.2 \pm _{0.5}$ | $\mathbf{82.2} \pm _{0.4}$ |

*joint sets $\{\mathcal{K}_{k_1,\ldots,k_P}\}_{k_1,\ldots,k_P \in [2]}$, the weight of the $2^P$-cut in the graph induced by them —* $\frac{1}{2} \sum_{k_1,\ldots,k_P \in [2]} \sum_{(n_1,\ldots,n_P) \in \mathcal{K}_{k_1,\ldots,k_P},(n'_1,\ldots,n'_P) \in \bar{\mathcal{K}} \setminus \mathcal{K}_{k_1,\ldots,k_P}} w(\{(n_1,\ldots,n_P),(n'_1,\ldots,n'_P)\})$ — *is equal, up to multiplicative and additive constants, to the term minimized in Equation* (5).

*Proof.* Consider the complete undirected graph whose vertices are $\bar{\mathcal{K}}$ and where the weight of an edge $\{(n_1,\ldots,n_P),(n'_1,\ldots,n'_P)\} \in \bar{\mathcal{K}} \times \bar{\mathcal{K}}$ is:

$$w(\{(n_1,\ldots,n_P),(n'_1,\ldots,n'_P)\}) = p_{(n_1,\ldots,n_P),(n'_1,\ldots,n'_P)}$$

(recall that $p_{(n_1,\ldots,n_P),(n'_1,\ldots,n'_P)}$ stands for the multivariate Pearson correlation between features $(n_1,\ldots,n_P)$ and $(n'_1,\ldots,n'_P)$ in $\mathcal{X}$). For any partition of $\bar{\mathcal{K}}$ into equally sized disjoint sets $\{\mathcal{K}_{k_1,\ldots,k_P}\}_{k_1,\ldots,k_P \in [2]}$, it holds that:

$$\frac{1}{2} \sum_{k_1,\ldots,k_P \in [2]} \sum_{(n_1,\ldots,n_P) \in \mathcal{K}_{k_1,\ldots,k_P},(n'_1,\ldots,n'_P) \in \bar{\mathcal{K}} \setminus \mathcal{K}_{k_1,\ldots,k_P}} w(\{(n_1,\ldots,n_P),(n'_1,\ldots,n'_P)\})$$
$$= \frac{1}{2} \sum_{k_1,\ldots,k_P \in [2]} SE(\mathcal{X};\mathcal{K}_{k_1,\ldots,k_P}) - \frac{1}{2} SE(\mathcal{X};\bar{\mathcal{K}}),$$

where $\frac{1}{2} SE(\mathcal{X};\bar{\mathcal{K}})$ does not depend on $\{\mathcal{K}_{k_1,\ldots,k_P}\}_{k_1,\ldots,k_P \in [2]}$. This concludes the proof. □

### J.3.3 Experiments

We supplement the experiments from Section 5.2 for one-dimensional models and data, by empirically evaluating Algorithm 3 using two-dimensional convolutional neural networks over randomly permuted image data. Specifically, Table 3 presents experiments analogous to those of Table 1 from Section 5.2.1 over the CIFAR10 [33] image dataset. For brevity, we defer some implementation details to Appendix L.

## K   Further Experiments

Table 4 supplements the experiment of Table 1 from Section 5.2.1, demonstrating that Algorithm 2 can be applied to data with a large number of features (*e.g.* with tens of thousands of features). In the experiment, instances of the minimum balanced cut problem solved as part of Algorithm 2 entail graphs with up to $25 \cdot 10^8$ edges. They are solved using the well-known edge sparsification algorithm of [57] that preserves weights of cuts, allowing for configurable compute and memory consumption (the more resources are consumed, the more accurate the solution will be). In contrast, this approach is not directly applicable for improving the efficiency of IGTD.

Table 5: Arranging features of tabular datasets via our method (detailed in Appendix G) significantly improves the prediction accuracies of locally connected neural networks. Reported are results of experiments identical to those of Table 2, but over the "dna" tabular classification dataset [60]. For further details, see caption of Table 2 and Appendix L.

| | Dataset: dna | | |
| --- | --- | --- | --- |
| | Baseline | Our Method | IGTD |
| CNN | $82.5 \pm 1.7$ | $\mathbf{91.2} \pm 1.1$ | $87.4 \pm 1.1$ |
| S4 | $86.4 \pm 1.7$ | $89.1 \pm 3.7$ | $89.9 \pm 1.1$ |
| Local-Attention | $79.2 \pm 4.0$ | $85.7 \pm 4.5$ | $82.7 \pm 3.2$ |

Table 5 supplements Table 2 from Section 5.2.2 by reporting results of experiments with an additional tabular benchmark — "dna".

## L   Further Implementation Details

We provide implementation details omitted from our experimental reports (Section 4.3, Section 5, Appendix J.2.3, Appendix J.3.3 and Appendix K). Source code for reproducing our results and figures, based on the PyTorch [44] framework, can be found at `https://github.com/nmd95/data_suitable_lc_nn_code`. All experiments were run on a single Nvidia RTX A6000 GPU.

### L.1   Empirical Demonstration of Theoretical Analysis (Figures 3, 5, and 8)

#### L.1.1   Figures 3 and 5

**Dataset**: The SpeechCommands dataset [64] contains raw audio segments of length up to 16000, split into 84843 train and 11005 test segments. We zero-padded all audio segments to have a length of 16000 and resampled them using sinc interpolation (default PyTorch implementation). We allocated 11005 audio segments from the train set for validation, *i.e.* the dataset was split into 73838 train, 11005 validation and 11005 test audio segments, and created a binary one-vs-all classification version of SpeechCommands by taking all audio segments labeled by the class "33" (corresponding to the word "yes"), and sampling an equal amount of segments from the remaining classes (this process was done separately for the train, validation and test sets). The resulting balanced classification dataset had 5610 train, 846 validation and 838 test segments. Lastly, we resampled all audio segments in the dataset from 16000 to 4096 features using sinc interpolation.

**Random feature swaps:** Starting with the original order of features, we created increasingly "shuffled" versions of the dataset by randomly swapping the position of features. For each number of random position swaps $k \in \{0, 250, \ldots, 4000\}$, we created ten datasets, whose features were subject to $k$ random position swaps between features, using different random seeds.

**Quantum entanglement measurement**: Each reported value in the plot is the average of entanglements with respect to canonical partitions (Definition 2) corresponding to levels $l = 1, 2, 3, 4, 5, 6$. We used Algorithm 1 described in Appendix F on two mini-batches of size 500, randomly sampled from the train set. As customary (*cf.* [59]), every input feature $x$ was embedded using the following sine-cosine scheme:

$$\phi(x) := (\sin(\pi\theta x), \cos(\pi\theta x)) \in \mathbb{R}^2 \,,$$

with $\theta = 0.085$.

**Neural network architectures**:

- **CNN**: An adaptation of the M5 architecture from [17], which is designed for audio data. Our implementation is based on `https://github.com/danielajisafe/Audio_WaveForm_Paper_Implementation`.

- **S4**: Official implementation of [26] with a hidden dimension of 128 and 4 layers.

- **Local-Attention**: Adaptation of the local-attention model from [46], as implemented in `https://github.com/lucidrains/local-attention`. We use a multi-layer perceptron (MLP) for mapping continuous (raw) inputs to embeddings, which are fed as input the the local-attention model. For classification, we collapse the spatial dimension of the network's output using mean-pooling and pass the result into an MLP classification head. The network had attention dimension 128 (with 2 heads of dimension 64), depth 8, hidden dimension of MLP blocks 341

(computed automatically by the library based on the attention dimension) and local-attention window size 10.

**Training and evaluation:** The binary cross-entropy loss was minimized via the Adam optimizer [32] with default $\beta_1, \beta_2$ coefficients. Batch sizes were chosen to be the largest powers of two that fit in the GPU memory. The duration of optimization was 300 epochs for all models (number of epochs was taken large enough such that the train loss plateaued). After the last training epoch, the model which performed best on the validation set was chosen, and its average test accuracy is reported. Additional optimization hyperparameters are provided in Table 6. We note that for S4 [26], in accordance with its official implementation, we use a cosine annealing learning rate scheduler.

Table 6: Optimization hyperparameters for the experiments of Figure 3.

| Model | Optimizer | Learning Rate | Weight Decay | Batch Size |
|---|---|---|---|---|
| CNN | Adam | 0.001 | 0.0001 | 128 |
| S4 | AdamW | 0.001 | 0.01 | 64 |
| Local-Attention | Adam | 0.0005 | 0 | 32 |

## L.1.2 Figure 8

**Dataset**: We created one-vs-all binary classification datasets based on CIFAR10 [33] as follows. All images were converted to grayscale using the PyTorch default implementation. Then, we allocated 7469 images from the train set for validation, *i.e.* the dataset was split into 42531 train, 7469 validation and 1000 test images. We took all images labeled by the class "0" (corresponding to images of airplanes), and uniformly sampled an equal amount of images from the remaining classes (this process was done separately for the train, validation and test sets). The resulting balanced classification dataset had 8506 train, 1493 validation and 2001 test images.

**Random feature swaps:** We created increasingly "shuffled" versions of the dataset according to the protocol described in Appendix L.1.1.

**Quantum entanglement measurement**: Each reported value in the plot is the average entanglements with respect to canonical partitions (Definition 6) corresponding to levels $l = 1, 2$.

**Neural network architectures**:

- **CNN**: Same architecture used for the experiments of Table 2 (see Appendix L.2.3), but with one-dimensional convolutional layers replaced with two-dimensional convolutional layers. See Table 7 for the exact architectural hyperparameters.

- **VGG**: Adaptation of the VGG16 architecture, as implemented in the PyTorch framework, for binary classification tasks on grayscale images. Specifically, the number of input channels is set to one and the number of output classes is set to two.

- **MobileNet**: Adaptation of the MobileNet V2 architecture, as implemented in the PyTorch framework, for binary classification on grayscale images. Specifically, the number of input channels is set to one and the number of output classes is set to two.

Table 7: Architectural hyperparameters for the convolutional neural network (referred to as "CNN") used in the experiments of Figure 8 and Table 3.

| Hyperparameter | Value |
|---|---|
| Stride | (3, 3) |
| Kernel size | (3, 3) |
| Pooling window size | (3, 3) |
| Number of blocks | 8 |
| Hidden dimension | 32 |

**Training and evaluation**: The binary cross-entropy loss was minimized for 150 epochs via the Adam optimizer [32] with default $\beta_1, \beta_2$ coefficients (number of epochs was taken large enough such that the train loss plateaued). Batch size was chosen to be the largest power of two that fit in the

GPU memory. After the last training epoch, the model which performed best on the validation set was chosen, and its average test accuracy is reported. Additional optimization hyperparameters are provided in Table 8.

Table 8: Optimization hyperparameters for the experiments of Figure 8.

| Model | Optimizer | Learning Rate | Weight Decay | Batch Size |
|---|---|---|---|---|
| CNN | Adam | 0.001 | 0.0001 | 128 |
| VGG | Adam | 0.0001 | 0.0001 | 64 |
| MobileNet | Adam | 0.001 | 0.0001 | 64 |

## L.2 Enhancing Suitability of Data to Locally Connected Neural Networks (Tables 1, 2, 3, 4, and 5)

### L.2.1 Feature Rearrangement Algorithms

Algorithm 2, 3 and IGTD [67] were applied to the training set. Subsequently, the learned feature rearrangement was used for both the validation and test data. In instances where three-way cross-validation was used, distinct rearrangements were learned for each separately.

**Algorithm 2 and Algorithm 3**: Approximate solutions to the minimum balanced cut problems were obtained using the METIS graph partitioning algorithm [29], as implemented in `https://github.com/networkx/networkx-metis`.

**IGTD [67]**: The original IGTD implementation supports rearranging data only into two-dimensional images. We adapted its implementation to support one-dimensional (sequential) data for the experiments of Tables 1 and 2.

### L.2.2 Randomly Permuted Audio Datasets (Table 1)

**Dataset**: To facilitate efficient experimentation, we downsampled all audio segments in SpeechCommands to have 2048 features. Furthermore, for the train, validation and test sets separately, we used 20% of the audio segments available for each class.

**Neural network architectures**:

- **CNN**: Same architecture used for the experiments of Figure 3 (see Appendix L.1.1).
- **S4**: Same architecture used for the experiments of Figure 3 (see Appendix L.1.1).
- **Local-Attention**: Same architecture used in the experiments of Figure 3, but with a depth 4 network (we reduced the depth to allow for more efficient experimentation).

**Training and evaluation:** The cross-entropy loss was minimized via the Adam optimizer [32] with default $\beta_1, \beta_2$ coefficients. Batch sizes were chosen to be the largest powers of two that fit in the GPU memory. The duration of optimization was 200, 200 and 450 epochs for the CNN, S4 and Local-Attention models, respectively (number of epochs was taken large enough such that the train loss plateaued). After the last training epoch, the model which performed best on the validation set was chosen, and its average test accuracy is reported. Additional optimization hyperparameters are provided in Table 9. We note that for S4 [26], in accordance with its official implementation, we use a cosine annealing learning rate scheduler.

Table 9: Optimization hyperparameters for the experiments of Table 1.

| Model | Optimizer | Learning Rate | Weight Decay | Batch Size |
|---|---|---|---|---|
| CNN | Adam | 0.001 | 0.0001 | 128 |
| S4 | AdamW | 0.001 | 0.01 | 64 |
| Local-Attention | Adam | 0.0001 | 0 | 32 |

### L.2.3 Tabular Datasets (Tables 2 and 5)

**Datasets**: The datasets "dna", "semeion" and "isolet" are all from the OpenML repository [60]. For each dataset we split the samples intro three folds, which were used for evaluation according to a

standard three-way cross-validation protocol. That is, for each of the three folds, we used one third of the data as a test set and the remaining for train and validation. One third of the samples in the remaining folds (not used for testing) were allocated for the validation set.

**Neural network architectures**:

- **CNN**: We used a ResNet adapted for tabular data. It consisted of residual blocks of the following form:
$$\text{Block}(\mathbf{x}) = \text{dropout}(\mathbf{x} + \text{BN}(\text{maxpool}(\text{ReLU}(\text{conv}(\mathbf{x}))))) \, .$$

  After applying a predetermined amount of residual blocks, a global average pooling and fully connected layers were used to output the prediction. The architectural hyperparameters are specified in Table 10.

- **S4**: Same architecture used for the experiments of Figure 3 (see Appendix L.1.1), but with a hidden dimension of 64.

- **Local-Attention**: Same architecture used for the experiments of Figure 3 (see Appendix L.1.1), but with 4 attention heads of dimension 32 and a local-attention window size of 25.

Table 10: Architectural hyperparameters for the convolutional neural network used in the experiments of Tables 2 and 4.

| Hyperparameter | Value |
|---|---|
| Stride | 3 |
| Kernel size | 3 |
| Pooling window size | 3 |
| Number of blocks | 8 |
| Hidden dimension | 32 |

**Training and evaluation**: The cross-entropy loss was minimized for 300 epochs via the Adam optimizer [32] with default $\beta_1, \beta_2$ coefficients (number of epochs was taken large enough such that the train loss plateaued). Batch sizes were chosen to be the largest powers of two that fit in the GPU memory. After the last training epoch, the model which performed best according to the validation sets was chosen, and test accuracy was measured on the test set. The reported accuracy is the average over the three folds. Additional optimization hyperparameters are specified in Table 11. We note that for S4 [26], in accordance with its official implementation, we use a cosine annealing learning rate scheduler.

Table 11: Optimization hyperparameters for the experiments of Table 2.

| Model | Optimizer | Learning Rate | Weight Decay | Batch Size |
|---|---|---|---|---|
| CNN | Adam | 0.001 | 0.0001 | 64 |
| S4 | AdamW | 0.001 | 0.01 | 64 |
| Local-Attention | Adam | 0.00005 | 0 | 64 |

### L.2.4 Randomly Permuted Image Datasets (Table 3)

**Dataset**: The data acquisition process followed the protocol described in Appendix L.1.2, except that the data was not converted into a binary one-vs-all classification dataset.

**Neural network architectures**:

- **CNN**: Same architecture used for the experiments of Figure 8 (see Appendix L.1.2).

**Training and evaluation**: The cross-entropy loss was minimized for 500 epochs via the Adam optimizer [32] with default $\beta_1, \beta_2$ coefficients (number of epochs was taken large enough such that the train loss plateaued). Batch size was chosen to be the largest power of two that fit in the GPU memory. After the last training epoch, the model which performed best on the validation set was chosen, and its average test accuracy is reported. Additional optimization hyperparameters are provided in Table 12.

Table 12: Optimization hyperparameters for the experiments of Table 3.

| Model | Optimizer | Learning Rate | Weight Decay | Batch Size |
|-------|-----------|---------------|--------------|------------|
| CNN | Adam | 0.001 (multiplied by 0.1 after 300 epochs) | 0.0001 | 128 |

### L.2.5 Randomly Permuted Audio Datasets With a Large Number of Features (Table 4)

**Dataset**: The data acquisition process followed the protocol described in Appendix L.1.1, except that the data was not transformed into a binary one-vs-all classification dataset and the audio segments were upsampled from 16,000 to 50,000.

**Edge Sparsification**: To facilitate running the METIS graph partitioning algorithm over the minimum balanced cut problems encountered as part of Algorithm 2, we first removed edges from the graph using the spectral sparsification algorithm of [57]. Specifically, we used the official Julia implementation (https://github.com/danspielman/Laplacians.jl) with hyperparameter $\epsilon = 0.15$.

**Neural network architectures**:

- **CNN**: Same architecture used for the experiments of Table 2 (see Appendix L.2.3).
- **S4**: Same architecture used for the experiments of Table 1 (see Appendix L.1.1), but with a hidden dimension of 32 (we reduced the hidden dimension due to GPU memory considerations).

**Training and evaluation**: The cross-entropy loss was minimized for 200 epochs via the Adam optimizer [32] with default $\beta_1, \beta_2$ coefficients (number of epochs was taken large enough such that the train loss plateaued). Batch sizes were chosen to be the largest powers of two that fit in the GPU memory. After the last training epoch, the model which performed best on the validation set was chosen, and its average test accuracy is reported. Additional optimization hyperparameters are provided in Table 13. We note that for S4 [26], in accordance with its official implementation, we use a cosine annealing learning rate scheduler.

Table 13: Optimization hyperparameters for the experiments of Table 4.

| Model | Optimizer | Learning Rate | Weight Decay | Batch Size |
|-------|-----------|---------------|--------------|------------|
| CNN | Adam | 0.001 | 0.0001 | 64 |
| S4 | AdamW | 0.001 | 0.01 | 64 |

# M Deferred Proofs

## M.1 Useful Lemmas

Below we collect a few useful results, which we will use in our proofs.

**Lemma 2.** *We denote the vector of singular values of a matrix* $\mathbf{X}$ *(arranged in decreasing order) by* $S(\mathbf{X})$. *For any* $\mathbf{A}, \mathbf{B} \in \mathbb{R}^{D_1 \times D_2}$ *it holds that:*

$$\|S(\mathbf{A}) - S(\mathbf{B})\| \leq \|S(\mathbf{A} - \mathbf{B})\|$$

*Proof.* See Theorem III.4.4 of [4]. $\square$

**Lemma 3.** *Let* $\mathcal{P} = \{p_1, ..., p_N\}, \mathcal{Q} = \{q_1, ..., q_N\}$ *be two probability distributions supported on* $[N]$, *and denote by* $TV(\mathcal{P}, \mathcal{Q}) := \frac{1}{2} \sum_{n=1}^{N} |p_n - q_n|$ *their* total variation distance. *If for* $\epsilon \in (0, 1/2)$ *it holds that* $TV(\mathcal{P}, \mathcal{Q}) \leq \epsilon$, *then:*

$$|H(\mathcal{P}) - H(\mathcal{Q})| \leq H_b(\epsilon) + \epsilon \cdot \ln(N),$$

*where* $H(\mathcal{P}) := -\sum_{n=1}^{N} p_n \ln(p_n)$ *is the entropy of* $\mathcal{P}$, *and* $H_b(c) := -c \cdot \ln(c) - (1-c) \cdot \ln(1-c)$ *is the binary entropy of a Bernoulli distribution parameterized by* $c \in [0, 1]$.

*Proof.* See, *e.g.*, Theorem 11 of [27]. $\square$

**Lemma 4** (Hoeffding inequality in Hilbert space)**.** *Let $X_1, .., X_N$ be an i.i.d. sequence of random variables whose range is some separable Hilbert space $\mathcal{H}$. Suppose that $\mathbb{E}[X_n] = 0$ and $\|X_n\| \leq c$ for all $n \in [N]$. Then, for all $t \geq 0$:*

$$Pr\left(\left\|\frac{1}{n}\sum_{n=1}^{N}X_n\right\| \geq t\right) \leq 2\exp\left(-\frac{Nt^2}{2c^2}\right),$$

*where $\|\cdot\|$ refers to the Hilbert space norm.*

*Proof.* See Section 2.4 of [51]. $\qquad\square$

**Lemma 5** (adapted from [35])**.** *Let $G = (V, E)$ be the perfect binary tree graph, with vertices $V$ and edges $E$, underlying the locally connected tensor network that generates $\mathcal{W}_{\mathrm{TN}} \in \mathbb{R}^{D_1 \times \cdots \times D_N}$ (defined in Section 3.1). For $\mathcal{K} \subseteq [N]$, let $V_\mathcal{K} \subseteq V$ and $V_{K^c} \subseteq \mathcal{V}$ be the leaves in $G$ corresponding to axes indices $\mathcal{K}$ and $\mathcal{K}^c$ of $\mathcal{W}_{\mathrm{TN}}$, respectively. Lastly, given a cut $(A, B)$ of $V$, i.e. $A \subseteq V$ and $B \subseteq V$ are disjoint and $A \cup B = V$, denote by $C(A, B) := \{\{u, v\} \in E : u \in A, v \in B\}$ the edge-cut set. Then:*

$$\mathrm{rank}(\llbracket \mathcal{W}_{\mathrm{TN}}; \mathcal{K} \rrbracket) \leq \min{}_{\mathit{cut}\ (A,B)\ \mathit{of}\ V\ \mathit{s.t.}\ V_\mathcal{K} \subseteq A, V_{\mathcal{K}^c} \subseteq B}\ R^{|C(A,B)|}.$$

*In particular, if $(\mathcal{K}, \mathcal{K}^c) \in \mathcal{C}_N$, then:*

$$\mathrm{rank}(\llbracket \mathcal{W}_{\mathrm{TN}}; \mathcal{K} \rrbracket) \leq R.$$

*Proof.* See Claim 1 in [35] for the upper bound on the rank of $\llbracket \mathcal{W}_{\mathrm{TN}}; \mathcal{K} \rrbracket$ for any $\mathcal{K} \subseteq [N]$. If $(\mathcal{K}, \mathcal{K}^c) \in \mathcal{C}_N$, since there exists a cut $(A, B)$ of $V$ such that $V_\mathcal{K} \subseteq A$ and $V_{\mathcal{K}^c} \subseteq B$ whose edge-cut set is of a singleton, we get that $\mathrm{rank}(\llbracket \mathcal{W}_{\mathrm{TN}}; \mathcal{K} \rrbracket) \leq R$. $\qquad\square$

**Lemma 6** (adapted from [35])**.** *Let $G = (V, E)$ be the perfect $2^P$-ary tree graph, with vertices $V$ and edges $E$, underlying the locally connected tensor network that generates $\mathcal{W}_{TN}^P \in \mathbb{R}^{D_1 \times \cdots \times D_{N^P}}$ (defined in Appendix J.1.1). Furthermore, let $\mu : [N]^P \to [N^P]$ be a compatible map from $P$-dimensional coordinates to axes indices of $\mathcal{W}_{TN}^P$ (Definition 5). For $\mathcal{J} \subseteq [N^P]$, let $V_\mathcal{J} \subseteq V$ and $V_{\mathcal{J}^c} \subseteq \mathcal{V}$ be the leaves in $G$ corresponding to axes indices $\mathcal{J}$ and $\mathcal{J}^c$ of $\mathcal{W}_{TN}^P$, respectively. Lastly, given a cut $(A, B)$ of $V$, i.e. $A \subseteq V$ and $B \subseteq V$ are disjoint and $A \cup B = V$, denote by $C(A, B) := \{\{u, v\} \in E : u \in A, v \in B\}$ the edge-cut set. Then:*

$$\mathrm{rank}(\llbracket \mathcal{W}_{TN}^P; \mathcal{J} \rrbracket) \leq \min{}_{\mathit{cut}\ (A,B)\ \mathit{of}\ V\ \mathit{s.t.}\ V_\mathcal{J} \subseteq A, V_{\mathcal{J}^c} \subseteq B}\ R^{|C(A,B)|}.$$

*In particular, if $(\mathcal{K}, \mathcal{K}^c) \in \mathcal{C}_N^P$, then:*

$$\mathrm{rank}(\llbracket \mathcal{W}_{\mathrm{TN}}; \mu(\mathcal{K}) \rrbracket) \leq R.$$

*Proof.* See Claim 1 in [35] for the upper bound on the rank of $\llbracket \mathcal{W}_{TN}^P; \mathcal{J} \rrbracket$ for any $\mathcal{J} \subseteq [N^P]$. If $(\mathcal{K}, \mathcal{K}^c) \in \mathcal{C}_N^P$, since there exists a cut $(A, B)$ of $V$ such that $V_{\mu(\mathcal{K})} \subseteq A$ and $V_{\mu(K)^c} \subseteq B$ whose edge-cut set is of a singleton, we get that $\mathrm{rank}(\llbracket \mathcal{W}_{\mathrm{TN}}; \mu(\mathcal{K}) \rrbracket) \leq R$. $\qquad\square$

**Lemma 7** (adapted from Theorem 3.18 in [25])**.** *Let $\mathcal{A} \in \mathbb{R}^{D_1 \times \cdots \times D_N}$ and $\epsilon > 0$. For each canonical partition $(\mathcal{K}, \mathcal{K}^c) \in \mathcal{C}_N$, let $\sigma_{\mathcal{K},1} \geq \ldots \sigma_{K,D_\mathcal{K}}$ be the singular values of $\llbracket \mathcal{A}; \mathcal{K} \rrbracket$, where $D_\mathcal{K} := \min\{\prod_{n \in \mathcal{K}} D_n, \prod_{n \in \mathcal{K}^c} D_n\}$. Let $(n_\mathcal{K})_{\mathcal{K} \in \mathcal{C}_N} \in \mathbb{N}^{\mathcal{C}_N}$ be an assignment of an integer to each $\mathcal{K} \in \mathcal{C}_N$. For any such assignment, consider the set of tensors with* Hierarchical Tucker *(HT) rank at most $(n_\mathcal{K})_{\mathcal{K} \in \mathcal{C}_N}$ as follows:*

$$HT\left((n_\mathcal{K})_{\mathcal{K} \in \mathcal{C}_N}\right) := \left\{\mathcal{V} \in \mathbb{R}^{D_1 \times \cdots \times D_N} : \forall \mathcal{K} \in \mathcal{C}_N, \mathrm{rank}(\llbracket \mathcal{V}; \mathcal{K} \rrbracket) \leq n_\mathcal{K}\right\}.$$

*Suppose that for all $(\mathcal{K}, \mathcal{K}^c) \in \mathcal{C}_N$ it holds that $\sqrt{\sum_{d=n_\mathcal{K}+1}^{D_\mathcal{K}} \sigma_{\mathcal{K},d}^2} \leq \frac{\epsilon}{\sqrt{2N-3}}$. Then, there exists $\mathcal{W} \in HT\left((n_\mathcal{K})_{\mathcal{K} \in \mathcal{C}_N}\right)$ satisfying:*

$$\|\mathcal{W} - \mathcal{A}\| \leq \epsilon.$$

*In particular, if for all $(\mathcal{K}, \mathcal{K}^c) \in \mathcal{C}_N$ it holds that $\sqrt{\sum_{d=R+1}^{D_\mathcal{K}} \sigma_{\mathcal{K},d}^2} \leq \frac{\epsilon}{\sqrt{2N-3}}$, then there exists $\mathcal{W}_{\mathrm{TN}} \in \mathbb{R}^{D_1 \times \cdots \times D_N}$ generated by the locally connected tensor network (defined in Section 3.1) satisfying:*

$$\|\mathcal{W}_{\mathrm{TN}} - \mathcal{A}\| \leq \epsilon.$$

**Lemma 8** (adapted from Theorem 3.18 in [25])**.** *Let* $(n_\mathcal{K})_{\mathcal{K} \in \mathcal{C}_N} \in \mathbb{N}^{\mathcal{C}_N}$ *be an assignment of an integer to each* $\mathcal{K} \in \mathcal{C}_N$*. For any such assignment, consider the set of tensors with* Hierarchical Tucker *(HT) rank at most* $(n_\mathcal{K})_{\mathcal{K} \in \mathcal{C}_N}$ *as follows:*

$$HT\left((n_\mathcal{K})_{\mathcal{K} \in \mathcal{C}_N}\right) := \left\{ \mathcal{V} \in \mathbb{R}^{D_1 \times \cdots \times D_N} : \forall \mathcal{K} \in \mathcal{C}_N, \operatorname{rank}([\![\mathcal{V}; \mathcal{K}]\!]) \leq n_\mathcal{K} \right\}.$$

*Consider a locally connected tensor network with varying widths* $(n_\mathcal{K})_{\mathcal{K} \in \mathcal{C}_N}$*, i.e. a tensor network conforming to a perfect binary tree graph in which the lengths of inner axes are as follows (in contrast to all being equal to* $R$ *as in the locally connected tensor network defined in Section 3.1). An axis corresponding to an edge that connects a node with descendant leaves indexed by* $\mathcal{K}$ *to its parent is assigned the length* $n_\mathcal{K}$*. Then, every* $\mathcal{A} \in HT\left((n_\mathcal{K})_{\mathcal{K} \in \mathcal{C}_N}\right)$ *can be represented by said locally connected tensor network with varying widths, meaning there exists an assignment to the tensors of the tensor network such that it generates* $\mathcal{A}$*. In particular, if* $n_K = R$ *for all* $\mathcal{K} \in \mathcal{C}_N$*, then* $\mathcal{A}$ *can be generated by the locally connected tensor network with all inner axes being of length* $R$*.*

**Lemma 9.** *Let* $\mathcal{V}, \mathcal{W} \in \mathbb{R}^{D_1 \times \cdots \times D_N}$ *be tensors such that* $\|\mathcal{V}\| = \|\mathcal{W}\| = 1$ *and* $\|\mathcal{V} - \mathcal{W}\| < \frac{1}{2}$*. Then, for any* $(\mathcal{K}, \mathcal{K}^c) \in \mathcal{C}_N$ *it holds that:*

$$|QE(\mathcal{V}; \mathcal{K}) - QE(\mathcal{W}; \mathcal{K})| \leq H_b(\|\mathcal{V} - \mathcal{W}\|) + \|\mathcal{V} - \mathcal{W}\| \cdot \ln(\mathcal{D}_\mathcal{K}),$$

*where* $H_b(c) := -(c \cdot \ln(c) + (1 - c) \cdot \ln(1 - c))$ *is the binary entropy of a Bernoulli distribution parameterized by* $c \in [0, 1]$*, and* $\mathcal{D}_\mathcal{K} := \min\{\prod_{n \in \mathcal{K}} D_n, \prod_{n \in \mathcal{K}^c} D_n\}$*.*

*Proof.* For any matrix $\mathbf{M} \in \mathbb{R}^{D_1 \times D_2}$ with $D := \min\{D_1, D_2\}$, let $S(\mathbf{M}) = (\sigma_{\mathbf{M},1}, ..., \sigma_{\mathbf{M},D})$ be the vector consisting of its singular values. First note that

$$\|\mathcal{V} - \mathcal{W}\| = \|S([\![\mathcal{V} - \mathcal{W}; \mathcal{K}]\!])\| \leq \|\mathcal{V} - \mathcal{W}\|.$$

So by Lemma 2

$$\|S([\![\mathcal{V}; \mathcal{K}]\!]) - S([\![\mathcal{W}; \mathcal{K}]\!])\| \leq \|\mathcal{V} - \mathcal{W}\|.$$

Let $v_1 \geq \cdots \geq v_{D_\mathcal{K}}$ and $w_1 \geq \cdots \geq w_{D_\mathcal{K}}$ be the singular values of $[\![\mathcal{V}; \mathcal{K}]\!]$ and $[\![\mathcal{W}; \mathcal{K}]\!]$, respectively. We have by the Cauchy-Schwarz inequality that

$$\left(\sum_{d=1}^{D_\mathcal{K}} |w_d^2 - v_d^2|\right)^2 = \left(\sum_{d=1}^{D_\mathcal{K}} |w_d - v_d| \cdot |w_d + v_d|\right)^2 \leq \left(\sum_{d=1}^{D_\mathcal{K}} (w_d - v_d)^2\right)\left(\sum_{d=1}^{D_\mathcal{K}} (w_d + v_d)^2\right).$$

Now the first term is upper bounded by $\|\mathcal{V} - \mathcal{W}\|^2$, and for the second we have

$$\sum_{d=1}^{D_\mathcal{K}} (w_d + v_d)^2 = \sum_{d=1}^{D_\mathcal{K}} w_d^2 + \sum_{d=1}^{D_\mathcal{K}} v_d^2 + 2v_d w_d = 2 + 2\sum_{d=1}^{D_\mathcal{K}} v_d w_d \leq 4,$$

where we use the fact that $\|\mathcal{V}\| = \|\mathcal{W}\| = 1$, and again Cuachy-Schwarz. Overall we have:

$$\sum_{d=1}^{D_\mathcal{K}} |w_d^2 - v_d^2| \leq 2\|\mathcal{V} - \mathcal{W}\|.$$

Note that the left hand side of the inequality above equals twice the total variation distance between the distributions defined by $\{w_d^2\}_{d=1}^{D_\mathcal{K}}$ and $\{v_d^2\}_{d=1}^{D_\mathcal{K}}$. Therefore by Lemma 3 we have:

$$|QE(\mathcal{V}; \mathcal{K}) - QE(\mathcal{W}; \mathcal{K})| = |H(\{w_d^2\}) - H(\{v_d^2\})| \leq \|\mathcal{V} - \mathcal{W}\| \cdot \ln(\mathcal{D}_\mathcal{K}) + H_b(\|\mathcal{V} - \mathcal{W}\|).$$

$\square$

**Lemma 10.** *Let* $\mathcal{P} = \{p(x)\}_{x \in [S]}$*, where* $S \in \mathbb{N}$*, be a probability distribution, and denote its entropy by* $H(\mathcal{P}) := \mathbb{E}_{x \sim \mathcal{P}}[\ln(1/p(x))]$*. Then, for any* $0 < a < 1$*, there exists a subset* $T \subseteq [S]$ *such that* $Pr_{x \sim \mathcal{P}}(T^c) \leq a$ *and* $|T| \leq e^{\frac{H(\mathcal{P})}{a}}$*.*

*Proof.* By Markov's inequality we have for any $0 < a < 1$:

$$Pr_{x \sim \mathcal{P}}\left(\left\{x : e^{-\frac{H(P)}{a}} \geq p(x)\right\}\right) = Pr_{x \sim \mathcal{P}}\left(\left\{x : \ln\left(\frac{1}{p(x)}\right) \geq \frac{H(P)}{a}\right\}\right) \leq a.$$

Let $T := \left\{x : e^{-\frac{H(\mathcal{P})}{a}} \leq p(x)\right\} \subseteq [S]$. Note that

$$e^{-\frac{H(\mathcal{P})}{a}} |T| \leq \sum_{x \in T} p(x) \leq \sum_{x \in [S]} p(x) = 1,$$

and so $|T| \leq e^{\frac{H(\mathcal{P})}{a}}$ and $Pr_{x \sim \mathcal{P}}(T^c) \leq a$, as required. $\square$

## M.2 Proof of Theorem 1

If $\mathcal{A} = 0$ the theorem is trivial, since then $QE(\mathcal{A}; \mathcal{K}) = 0$ for all $(\mathcal{K}, \mathcal{K}^c) \in \mathcal{C}_N$, so we can assume $\mathcal{A} \neq 0$. We have:

$$
\begin{aligned}
\left\| \frac{\mathcal{W}_{\text{TN}}}{\|\mathcal{W}_{\text{TN}}\|} - \frac{\mathcal{A}}{\|\mathcal{A}\|} \right\| &= \frac{1}{\|\mathcal{A}\|} \left\| \frac{\|\mathcal{A}\|}{\|\mathcal{W}_{\text{TN}}\|} \cdot \mathcal{W}_{\text{TN}} - \mathcal{A} \right\| \\
&\leq \frac{1}{\|\mathcal{A}\|} \left( \left| \frac{\|\mathcal{A}\|}{\|\mathcal{W}_{\text{TN}}\|} - 1 \right| \cdot \|\mathcal{W}_{\text{TN}}\| + \|\mathcal{W}_{\text{TN}} - \mathcal{A}\| \right) \\
&= \frac{1}{\|\mathcal{A}\|} \left( |\|\mathcal{A}\| - \|\mathcal{W}_{\text{TN}}\|| + \|\mathcal{W}_{\text{TN}} - \mathcal{A}\| \right) \\
&\leq \frac{2\epsilon}{\|\mathcal{A}\|}.
\end{aligned}
$$

Now, let $\hat{\mathcal{A}} := \frac{\mathcal{A}}{\|\mathcal{A}\|}$ and $\hat{\mathcal{W}}_{\text{TN}} = \frac{\mathcal{W}_{\text{TN}}}{\|\mathcal{W}_{\text{TN}}\|}$ be normalized versions of $\mathcal{A}$ and $\mathcal{W}_{\text{TN}}$, respectively, and let $c = \frac{2\epsilon}{\|\mathcal{A}\|}$. Note that $c < \frac{2\|\mathcal{A}\|}{4} \frac{1}{\|\mathcal{A}\|} = \frac{1}{2}$, and therefore by Lemma 9 we have:

$$
\begin{aligned}
|QE(\mathcal{A}; \mathcal{K}) - QE(\mathcal{W}_{\text{TN}}; \mathcal{K})| &= \left| QE\left(\hat{\mathcal{A}}; \mathcal{K}\right) - QE\left(\hat{\mathcal{W}}_{\text{TN}}; \mathcal{K}\right) \right| \\
&\leq c \cdot \ln(D_{\mathcal{K}}) + H_b(c).
\end{aligned}
$$

By Lemma 5 we have that

$$
QE\left(\hat{\mathcal{W}}_{\text{TN}}; \mathcal{K}\right) \leq \ln(\text{rank}(\llbracket \mathcal{W}_{\text{TN}}; \mathcal{K} \rrbracket)) \leq \ln(R),
$$

and therefore

$$
QE\left(\hat{\mathcal{A}}; \mathcal{K}\right) \leq \ln(R) + c \ln(D_{\mathcal{K}}) + H_b(c).
$$

Substituting $c = \frac{2\epsilon}{\|\mathcal{A}\|}$ and invoking the elementary inequality $H_b(x) \leq 2\sqrt{x}$ we obtain

$$
QE(\mathcal{A}; \mathcal{K}) \leq \ln(R) + \frac{2\epsilon}{\|\mathcal{A}\|} \cdot \ln(D_{\mathcal{K}}) + 2\sqrt{\frac{2\epsilon}{\|\mathcal{A}\|}},
$$

as required.

As for the expected entanglements of a random tensor, let $\mathcal{A}' \in \mathbb{R}^{D_1 \times \cdots \times D_N}$ be drawn according to the uniform distribution over the set of unit norm tensors. According to [52], for any canonical partition $(\mathcal{K}, \mathcal{K}^c) \in \mathcal{C}_N$ it holds that:

$$
\mathbb{E}[QE(\mathcal{A}'; \mathcal{K})] = \sum_{k=D'_{\mathcal{K}}+1}^{D_{\mathcal{K}} \cdot D'_{\mathcal{K}}} \frac{1}{k} - \frac{D_{\mathcal{K}} - 1}{2D'_{\mathcal{K}}},
$$

where $D'_{\mathcal{K}} := \max\{\prod_{n \in \mathcal{K}} D_n, \prod_{n \in \mathcal{K}^c} D_n\}$ and recall that $D_{\mathcal{K}} := \min\{\prod_{n \in \mathcal{K}} D_n, \prod_{n \in \mathcal{K}^c} D_n\}$. By the elementary inequality of $\sum_{k=q}^{p} 1/k \geq \ln(p) - \ln(q)$ we therefore have that:

$$
\begin{aligned}
\mathbb{E}[QE(\mathcal{A}'; \mathcal{K})] &\geq \ln(D_{\mathcal{K}} \cdot D'_{\mathcal{K}}) - \ln(D'_{\mathcal{K}} + 1) - \frac{D_{\mathcal{K}} - 1}{2D'_{\mathcal{K}}} \\
&= \ln(D_{\mathcal{K}}) + \ln\left( \frac{D'_{\mathcal{K}}}{D'_{\mathcal{K}} + 1} \right) - \frac{D_{\mathcal{K}} - 1}{2D'_{\mathcal{K}}}.
\end{aligned}
$$

Since $D_{\mathcal{K}} \leq D'_{\mathcal{K}}$ and $D_{\mathcal{K}} \geq 1$, it holds that $(D_{\mathcal{K}} - 1)/2D'_{\mathcal{K}} \leq 1/2$ and $\ln(D'_{\mathcal{K}}/(D'_{\mathcal{K}} + 1)) \geq \ln(1/2)$. Thus:

$$
\mathbb{E}[QE(\mathcal{A}'; \mathcal{K})] \geq \ln(D_{\mathcal{K}}) + \ln\left(\frac{1}{2}\right) - \frac{1}{2} \geq \min\{|\mathcal{K}|, |\mathcal{K}^c|\} \cdot \ln(\min_{n \in [N]} D_n) + \ln\left(\frac{1}{2}\right) - \frac{1}{2}.
$$

$\square$

## M.3  Proof of Theorem 2

Consider, for each canonical partition $(\mathcal{K}, \mathcal{K}^c) \in \mathcal{C}_N$, the distribution

$$\mathcal{P}_\mathcal{K} = \left\{ p_\mathcal{K}(i) := \frac{\sigma^2_{\mathcal{K},i}}{\|\mathcal{A}\|^2} \right\}_{i \in [D_\mathcal{K}]},$$

where $\sigma_{\mathcal{K},1} \geq \sigma_{\mathcal{K},2} \geq \ldots \geq \sigma_{\mathcal{K},D_\mathcal{K}}$ are the singular values of $[\![\mathcal{A}; \mathcal{K}]\!]$ (note that $\frac{1}{\|\mathcal{A}\|^2} \sum_j \sigma^2_{\mathcal{K},j} = \frac{\|\mathcal{A}\|^2}{\|\mathcal{A}\|^2} = 1$ so $\mathcal{P}_\mathcal{K}$ is indeed a probability distribution). Denoting by $H(\mathcal{P}_\mathcal{K}) := \mathbb{E}_{i \sim \mathcal{P}_\mathcal{K}}[\ln(1/p_\mathcal{K}(i))]$ the entropy of $\mathcal{P}_\mathcal{K}$, by assumption:

$$QE(\mathcal{A}; \mathcal{K}) = H(\mathcal{P}_\mathcal{K}) \leq \frac{\epsilon^2}{\|\mathcal{A}\|^2(2N-3)} \ln(R),$$

for all $(\mathcal{K}, \mathcal{K}^c) \in \mathcal{C}_N$. Thus, taking $a = \frac{\epsilon^2}{\|\mathcal{A}\|^2(2N-3)}$ we obtain by Lemma 10 that there exists a subset $T_\mathcal{K} \subseteq [D_\mathcal{K}]$ such that

$$\mathcal{P}_\mathcal{K}(T_\mathcal{K}^c) \leq \frac{\epsilon^2}{(2N-3)\|\mathcal{A}\|^2},$$

and $|T_\mathcal{K}| \leq e^{\frac{H(\mathcal{P}_\mathcal{K})}{c}} = e^{\ln(R)} = R$. Note that

$$\mathcal{P}_\mathcal{K}(T_\mathcal{K}) \leq \sum_{i=1}^{R} \frac{\sigma_i^2}{\|\mathcal{A}\|^2}.$$

Since this holds for any subset of cardinality at most $R$. Taking complements we obtain

$$\sum_{i=R+1}^{D_\mathcal{K}} \frac{\sigma_i^2}{\|\mathcal{A}\|^2} \leq \mathcal{P}_\mathcal{K}(T_\mathcal{K}^c),$$

so

$$\sqrt{\sum_{i=R+1}^{D_\mathcal{K}} \sigma^2_{\mathcal{K},i}} \leq \frac{\epsilon}{\sqrt{(2N-3)}}.$$

We can now invoke Lemma 7, which implies that there exists some $\mathcal{W}_{\mathrm{TN}} \in \mathbb{R}^{D_1 \times \cdots \times D_N}$ generated by the locally connected tensor network satisfying:

$$\|\mathcal{W}_{\mathrm{TN}} - \mathcal{A}\| \leq \epsilon.$$

$\square$

## M.4  Proof of Corollary 1

Notice that the entanglements of a tensor are invariant to multiplication by a constant. In particular, $QE(\mathcal{D}_{\mathrm{pop}}; \mathcal{K}) = QE(\mathcal{D}_{\mathrm{pop}}/\|\mathcal{D}_{\mathrm{pop}}\|; \mathcal{K})$ for any $(\mathcal{K}, \mathcal{K}^c) \in \mathcal{C}_N$. Hence, if there exists a canonical partition $(\mathcal{K}, \mathcal{K}^c) \in \mathcal{C}_N$ under which $QE(\mathcal{D}_{\mathrm{pop}}; \mathcal{K}) > \ln(R) + 2\epsilon \cdot \ln(D_\mathcal{K}) + 2\sqrt{2}\epsilon$, then Theorem 1 implies that $\min_{\mathcal{W}_{\mathrm{TN}}} \|\mathcal{W}_{\mathrm{TN}} - \mathcal{D}_{\mathrm{pop}}/\|\mathcal{D}_{\mathrm{pop}}\|\| > \epsilon$. Now, for any non-zero $\mathcal{W} \in \mathbb{R}^{D_1 \times \cdots D_N}$ generated by the locally connected tensor network, one can also represent $\mathcal{W}/\|\mathcal{W}\|$ by multiplying any of the tensors constituting the tensor network by $1/\|\mathcal{W}\|$ (contraction is a multilinear operation). Thus:

$$\mathrm{SubOpt} := \min_{\mathcal{W}_{\mathrm{TN}}} \left\| \frac{\mathcal{W}_{\mathrm{TN}}}{\|\mathcal{W}_{\mathrm{TN}}\|} - \frac{\mathcal{D}_{\mathrm{pop}}}{\|\mathcal{D}_{\mathrm{pop}}\|} \right\| \geq \min_{\mathcal{W}_{\mathrm{TN}}} \left\| \mathcal{W}_{\mathrm{TN}} - \frac{\mathcal{D}_{\mathrm{pop}}}{\|\mathcal{D}_{\mathrm{pop}}\|} \right\| > \epsilon,$$

which concludes the first part of the claim, *i.e.* the necessary condition for low suboptimality in achievable accuracy.

For the sufficient condition, if for all $(\mathcal{K}, \mathcal{K}^c) \in \mathcal{C}_N$ it holds that $QE(\mathcal{D}_{\mathrm{pop}}; \mathcal{K}) \leq \frac{\epsilon^2}{8N-12} \cdot \ln(R)$, then by Theorem 2 there exists an assignment for the locally connected tensor network such that $\|\mathcal{W}_{\mathrm{TN}} - \mathcal{D}_{\mathrm{pop}}/\|\mathcal{D}_{\mathrm{pop}}\|\| \leq \epsilon/2$. From the triangle inequality we obtain:

$$\left\| \frac{\mathcal{W}_{\mathrm{TN}}}{\|\mathcal{W}_{\mathrm{TN}}\|} - \frac{\mathcal{D}_{\mathrm{pop}}}{\|\mathcal{D}_{\mathrm{pop}}\|} \right\| \leq \left\| \mathcal{W}_{\mathrm{TN}} - \frac{\mathcal{D}_{\mathrm{pop}}}{\|\mathcal{D}_{\mathrm{pop}}\|} \right\| + \left\| \mathcal{W}_{\mathrm{TN}} - \frac{\mathcal{W}_{\mathrm{TN}}}{\|\mathcal{W}_{\mathrm{TN}}\|} \right\| \leq \frac{\epsilon}{2} + \left\| \mathcal{W}_{\mathrm{TN}} - \frac{\mathcal{W}_{\mathrm{TN}}}{\|\mathcal{W}_{\mathrm{TN}}\|} \right\|. \tag{11}$$

Since $\|\mathcal{W}_{\mathrm{TN}} - \mathcal{D}_{\mathrm{pop}}/\|\mathcal{D}_{\mathrm{pop}}\|\| \le \epsilon/2$ it holds that $\|\mathcal{W}_{\mathrm{TN}}\| \le 1 + \epsilon/2$. Combined with the fact that $\left\|\mathcal{W}_{\mathrm{TN}} - \frac{\mathcal{W}_{\mathrm{TN}}}{\|\mathcal{W}_{\mathrm{TN}}\|}\right\| = \|\mathcal{W}_{\mathrm{TN}}\| - 1$, we get that $\left\|\mathcal{W}_{\mathrm{TN}} - \frac{\mathcal{W}_{\mathrm{TN}}}{\|\mathcal{W}_{\mathrm{TN}}\|}\right\| \le \epsilon/2$. Plugging this into Equation (11) yields:

$$\left\| \frac{\mathcal{W}_{\mathrm{TN}}}{\|\mathcal{W}_{\mathrm{TN}}\|} - \frac{\mathcal{D}_{\mathrm{pop}}}{\|\mathcal{D}_{\mathrm{pop}}\|} \right\| \le \epsilon,$$

and so $\mathrm{SubOpt} := \min_{\mathcal{W}_{\mathrm{TN}}} \left\| \frac{\mathcal{W}_{\mathrm{TN}}}{\|\mathcal{W}_{\mathrm{TN}}\|} - \frac{\mathcal{D}_{\mathrm{pop}}}{\|\mathcal{D}_{\mathrm{pop}}\|} \right\| \le \epsilon$. $\qquad\square$

## M.5 Proof of Proposition 2

We have the identity

$$\left\| \frac{\mathcal{D}_{\mathrm{pop}}}{\|\mathcal{D}_{\mathrm{pop}}\|} - \frac{\mathcal{D}_{\mathrm{emp}}}{\|\mathcal{D}_{\mathrm{emp}}\|} \right\| = \left\| \frac{\mathcal{D}_{pop}\|\mathcal{D}_{\mathrm{emp}}\| - \mathcal{D}_{\mathrm{emp}}\|\mathcal{D}_{\mathrm{pop}}\|}{\|\mathcal{D}_{\mathrm{pop}}\|\|\mathcal{D}_{\mathrm{emp}}\|} \right\| =$$

$$\left\| \frac{\mathcal{D}_{\mathrm{pop}}\|\mathcal{D}_{\mathrm{emp}}\| - \|\mathcal{D}_{\mathrm{emp}}\|\mathcal{D}_{\mathrm{emp}} + \mathcal{D}_{\mathrm{emp}}\|\mathcal{D}_{\mathrm{emp}}\| - \mathcal{D}_{\mathrm{emp}}\|\mathcal{D}_{\mathrm{pop}}\|}{\|\mathcal{D}_{\mathrm{pop}}\|\|\mathcal{D}_{\mathrm{emp}}\|} \right\|.$$

By the triangle inequality the above is bounded by

$$\frac{\|\mathcal{D}_{\mathrm{pop}} - \mathcal{D}_{\mathrm{emp}}\|}{\|\mathcal{D}_{\mathrm{pop}}\|} + \frac{|\|\mathcal{D}_{\mathrm{pop}}\| - \|\mathcal{D}_{\mathrm{emp}}\||}{\|\mathcal{D}_{\mathrm{pop}}\|}.$$

For $m \in [M]$, let $\mathcal{X}^{(m)} = y^{(m)} \cdot \otimes_{n \in [N]} \mathbf{x}^{(n,m)} - \mathcal{D}_{\mathrm{pop}}$. These are i.i.d. random variables with $\mathbb{E}[\mathcal{X}^{(m)}] = 0$ and $\|\mathcal{X}^{(m)}\| \le 2$ for all $m \in [M]$. Note that

$$\left\| \frac{1}{M} \sum_{m=1}^{M} \mathcal{X}^{(m)} \right\| = \|\mathcal{D}_{\mathrm{emp}} - \mathcal{D}_{\mathrm{pop}}\|,$$

so by Lemma 4 with $c = 2, t = \frac{\|\mathcal{D}_{\mathrm{pop}}\|\gamma}{2}$, assuming $M \ge \frac{2\ln(\frac{2}{\delta})}{\|\mathcal{D}_{\mathrm{pop}}\|^2\gamma^2}$ we have with probability at least $1 - \delta$

$$|\|\mathcal{D}_{\mathrm{pop}}\| - \|\mathcal{D}_{\mathrm{emp}}\|| \le \|\mathcal{D}_{pop} - \mathcal{D}_{\mathrm{emp}}\| \le \frac{\|\mathcal{D}_{\mathrm{pop}}\|\gamma}{2},$$

and therefore

$$\frac{\|\mathcal{D}_{\mathrm{pop}} - \mathcal{D}_{\mathrm{emp}}\|}{\|\mathcal{D}_{\mathrm{pop}}\|} + \frac{|\|\mathcal{D}_{\mathrm{pop}}\| - \|\mathcal{D}_{\mathrm{emp}}\||}{\|\mathcal{D}_{\mathrm{pop}}\|} \le \gamma.$$

$\qquad\square$

## M.6 Proof of Corollary 2

First, by Proposition 2, for $M \ge \frac{8\ln(\frac{2}{\delta})}{\|\mathcal{D}_{\mathrm{pop}}\|^2\epsilon^2}$ we have that with probability at least $1 - \delta$:

$$\left\| \frac{\mathcal{D}_{\mathrm{pop}}}{\|\mathcal{D}_{\mathrm{pop}}\|} - \frac{\mathcal{D}_{\mathrm{emp}}}{\|\mathcal{D}_{\mathrm{emp}}\|} \right\| \le \frac{\epsilon}{2}. \tag{12}$$

Now, we establish the necessary condition on the entanglements of $\mathcal{D}_{\mathrm{emp}}$. Assume that there exists a canonical partition $(\mathcal{K}, \mathcal{K}^c) \in \mathcal{C}_N$ under which

$$QE(\mathcal{D}_{\mathrm{emp}}; \mathcal{K}) > \ln(R) + 3\epsilon \cdot \ln(D_{\mathcal{K}}) + 2\sqrt{3\epsilon}.$$

We may view $\mathcal{D}_{\mathrm{emp}}$ as the population data tensor for the uniform data distribution over the training instances $\left\{ \left( (\mathbf{x}^{(1,m)}, \ldots, \mathbf{x}^{(N,m)}), y^{(m)} \right) \right\}_{m=1}^{M}$. Thus, Corollary 1 implies that

$$\min_{\mathcal{W}_{\mathrm{TN}}} \left\| \frac{\mathcal{W}_{\mathrm{TN}}}{\|\mathcal{W}_{\mathrm{TN}}\|} - \frac{\mathcal{D}_{\mathrm{emp}}}{\|\mathcal{D}_{\mathrm{emp}}\|} \right\| > 1.5\epsilon.$$

Using the triangle inequality after adding and subtracting $\frac{\mathcal{D}_{\mathrm{pop}}}{\|\mathcal{D}_{\mathrm{pop}}\|}$, it follows that

$$\min_{\mathcal{W}_{\mathrm{TN}}} \left\{ \left\| \frac{\mathcal{W}_{\mathrm{TN}}}{\|\mathcal{W}_{\mathrm{TN}}\|} - \frac{\mathcal{D}_{\mathrm{pop}}}{\|\mathcal{D}_{\mathrm{pop}}\|} \right\| + \left\| \frac{\mathcal{D}_{\mathrm{pop}}}{\|\mathcal{D}_{\mathrm{pop}}\|} - \frac{\mathcal{D}_{\mathrm{emp}}}{\|\mathcal{D}_{\mathrm{emp}}\|} \right\| \right\} > 1.5\epsilon.$$

Hence, combined with Equation (12) this concludes the current part of the proof:

$$\text{SubOpt} := \min_{\mathcal{W}_{\text{TN}}} \left\| \frac{\mathcal{W}_{\text{TN}}}{\|\mathcal{W}_{\text{TN}}\|} - \frac{\mathcal{D}_{\text{pop}}}{\|\mathcal{D}_{\text{pop}}\|} \right\| > 1.5\epsilon - \left\| \frac{\mathcal{D}_{\text{pop}}}{\|\mathcal{D}_{\text{pop}}\|} - \frac{\mathcal{D}_{\text{emp}}}{\|\mathcal{D}_{\text{emp}}\|} \right\| \geq 1.5\epsilon - \frac{\epsilon}{2} = \epsilon.$$

We turn our attention to the sufficient condition on the entanglements of $\mathcal{D}_{\text{emp}}$. Suppose that $QE(\mathcal{D}_{\text{emp}}; \mathcal{K}) \leq \frac{\epsilon^2}{32N-48} \cdot \ln(R)$ for all canonical partitions $(\mathcal{K}, \mathcal{K}^c) \in \mathcal{C}_N$. Invoking Corollary 1 while viewing $\mathcal{D}_{\text{emp}}$ as the population data tensor for the uniform data distribution over the training instances $\left\{ \left( (\mathbf{x}^{(1,m)}, \ldots, \mathbf{x}^{(N,m)}), y^{(m)} \right) \right\}_{m=1}^{M}$, we get that

$$\min_{\mathcal{W}_{\text{TN}}} \left\| \frac{\mathcal{W}_{\text{TN}}}{\|\mathcal{W}_{\text{TN}}\|} - \frac{\mathcal{D}_{\text{emp}}}{\|\mathcal{D}_{\text{emp}}\|} \right\| \leq \frac{\epsilon}{2}.$$

Combined with Equation (12), and by the triangle inequality, we conclude:

$$\begin{aligned}
\text{SubOpt} := \min_{\mathcal{W}_{\text{TN}}} & \left\| \frac{\mathcal{W}_{\text{TN}}}{\|\mathcal{W}_{\text{TN}}\|} - \frac{\mathcal{D}_{\text{pop}}}{\|\mathcal{D}_{\text{pop}}\|} \right\| \\
\leq \min_{\mathcal{W}_{\text{TN}}} & \left\| \frac{\mathcal{W}_{\text{TN}}}{\|\mathcal{W}_{\text{TN}}\|} - \frac{\mathcal{D}_{\text{emp}}}{\|\mathcal{D}_{\text{emp}}\|} \right\| + \left\| \frac{\mathcal{D}_{\text{pop}}}{\|\mathcal{D}_{\text{pop}}\|} - \frac{\mathcal{D}_{\text{emp}}}{\|\mathcal{D}_{\text{emp}}\|} \right\| \\
\leq \epsilon \, .
\end{aligned}$$

Lastly, the fact that the entanglements of $\mathcal{D}_{\text{emp}}$ can be evaluated efficiently is discussed in Appendix F.
□

### M.7    Proof of Theorem 3

If $\mathcal{A} = 0$ the theorem is trivial, since then $QE(\mathcal{A}; \mu(\mathcal{K})) = 0$ for all $(\mathcal{K}, \mathcal{K}^c) \in \mathcal{C}_N^P$, so we can assume $\mathcal{A} \neq 0$. We have:

$$\begin{aligned}
\left\| \frac{\mathcal{W}_{\text{TN}}^P}{\|\mathcal{W}_{\text{TN}}^P\|} - \frac{\mathcal{A}}{\|\mathcal{A}\|} \right\| &= \frac{1}{\|\mathcal{A}\|} \left\| \frac{\|\mathcal{A}\|}{\|\mathcal{W}_{\text{TN}}^P\|} \cdot \mathcal{W}_{\text{TN}}^P - \mathcal{A} \right\| \\
&\leq \frac{1}{\|\mathcal{A}\|} \left( \left| \frac{\|\mathcal{A}\|}{\|\mathcal{W}_{\text{TN}}^P\|} - 1 \right| \cdot \|\mathcal{W}_{\text{TN}}^P\| + \|\mathcal{W}_{\text{TN}}^P - \mathcal{A}\| \right) \\
&= \frac{1}{\|\mathcal{A}\|} \left( \left| \|\mathcal{A}\| - \|\mathcal{W}_{\text{TN}}^P\| \right| + \|\mathcal{W}_{\text{TN}}^P - \mathcal{A}\| \right) \\
&\leq \frac{2\epsilon}{\|\mathcal{A}\|} \, .
\end{aligned}$$

Now, let $\hat{\mathcal{A}} := \frac{\mathcal{A}}{\|\mathcal{A}\|}$ and $\hat{\mathcal{W}}_{\text{TN}}^P = \frac{\mathcal{W}_{\text{TN}}^P}{\|\mathcal{W}_{\text{TN}}^P\|}$ be normalized versions of $\mathcal{A}$ and $\mathcal{W}_{\text{TN}}^P$, respectively, and let $c = \frac{2\epsilon}{\|A\|}$. Note that $c < \frac{2\|\mathcal{A}\|}{4} \frac{1}{\|\mathcal{A}\|} = \frac{1}{2}$, and therefore by Lemma 9 we have:

$$\begin{aligned}
|QE(\mathcal{A}; \mu(\mathcal{K})) - QE(\mathcal{W}_{\text{TN}}^P; \mu(\mathcal{K}))| &= \left| QE\left( \hat{\mathcal{A}}; \mu(\mathcal{K}) \right) - QE\left( \hat{\mathcal{W}}_{\text{TN}}^P; \mu(\mathcal{K}) \right) \right| \\
&\leq c \cdot \ln(D_{\mu(\mathcal{K})}) + H_b(c) \, .
\end{aligned}$$

By Lemma 6 we have that

$$QE\left( \hat{\mathcal{W}}_{\text{TN}}^P; \mu(\mathcal{K}) \right) \leq \ln(\text{rank}(\llbracket \mathcal{W}_{\text{TN}}^P; \mu(\mathcal{K}) \rrbracket)) \leq \ln(R) \, ,$$

and therefore

$$QE\left( \hat{\mathcal{A}}; \mu(\mathcal{K}) \right) \leq \ln(R) + c \ln(D_{\mu(\mathcal{K})}) + H_b(c) \, .$$

Substituting $c = \frac{2\epsilon}{\|A\|}$ and invoking the elementary inequality $H_b(x) \leq 2\sqrt{x}$ we obtain

$$QE(\mathcal{A}; \mu(\mathcal{K})) \leq \ln(R) + \frac{2\epsilon}{\|\mathcal{A}\|} \cdot \ln(D_{\mu(\mathcal{K})}) + 2\sqrt{\frac{2\epsilon}{\|\mathcal{A}\|}} \, ,$$

as required.

As for Equation (7), let $\mathcal{A}' \in \mathbb{R}^{D_1 \times \cdots \times D_{N^P}}$ be drawn according to the uniform distribution over the set of unit norm tensors. According to [52], for any canonical partition $(\mathcal{K}, \mathcal{K}^c) \in \mathcal{C}_N^P$ it holds that:

$$\mathbb{E}[QE(\mathcal{A}'; \mu(\mathcal{K}))] = \sum_{k=D'_{\mu(\mathcal{K})}+1}^{D_{\mu(\mathcal{K})} \cdot D'_{\mu(\mathcal{K})}} \frac{1}{k} - \frac{D_{\mu(\mathcal{K})} - 1}{2D'_{\mu(\mathcal{K})}},$$

where we denote $D'_{\mu(\mathcal{K})} := \max\{\prod_{n \in \mu(\mathcal{K})} D_n, \prod_{n \in \mu(\mathcal{K})^c} D_n\}$ and recall that $D_{\mu(\mathcal{K})} := \min\{\prod_{n \in \mu(\mathcal{K})} D_n, \prod_{n \in \mu(\mathcal{K})^c} D_n\}$. By the elementary inequality of $\sum_{k=q}^p 1/k \geq \ln(p) - \ln(q)$ we therefore have that:

$$\mathbb{E}[QE(\mathcal{A}'; \mu(\mathcal{K}))] \geq \ln(D_{\mu(\mathcal{K})} \cdot D'_{\mu(\mathcal{K})}) - \ln(D'_{\mu(\mathcal{K})} + 1) - \frac{D_{\mu(\mathcal{K})} - 1}{2D'_{\mu(\mathcal{K})}}$$

$$= \ln(D_{\mu(\mathcal{K})}) + \ln\left(\frac{D'_{\mu(\mathcal{K})}}{D'_{\mu(\mathcal{K})} + 1}\right) - \frac{D_{\mu(\mathcal{K})} - 1}{2D'_{\mu(\mathcal{K})}}.$$

Since $D_{\mu(\mathcal{K})} \leq D'_{\mu(\mathcal{K})}$ and $D_{\mu(\mathcal{K})} \geq 1$, we have that:

$$\frac{D_{\mu(\mathcal{K})} - 1}{2D'_{\mu(\mathcal{K})}} \leq \frac{1}{2} \quad , \quad \ln\left(\frac{D'_{\mu(\mathcal{K})}}{D'_{\mu(\mathcal{K})} + 1}\right) \geq \ln\left(\frac{1}{2}\right).$$

Thus:

$$\mathbb{E}[QE(\mathcal{A}'; \mu(\mathcal{K}))] \geq \ln(D_{\mu(\mathcal{K})}) + \ln\left(\frac{1}{2}\right) - \frac{1}{2}$$

$$\geq \min\{|\mathcal{K}|, |\mathcal{K}^c|\} \cdot \ln(\min_{n \in [N^P]} D_n) + \ln\left(\frac{1}{2}\right) - \frac{1}{2}.$$

$\square$

## M.8 Proof of Theorem 4

Let $\mathcal{C}_{N^P}$ be the one-dimensional canonical partitions of $[N^P]$ (Definition 2). Note that $\mu(\mathcal{C}_N^P) := \{\mu(\mathcal{K}) : \mathcal{K} \in \mathcal{C}_N^P\} \subseteq \mathcal{C}_{N^P}$. For an assignment $(n_{\mathcal{K}})_{\mathcal{K} \in \mathcal{C}_{N^P}} \in \mathbb{N}^{\mathcal{C}_{N^P}}$ of integers to one-dimensional canonical partitions $\mathcal{K} \in \mathcal{C}_{N^P}$, we consider the set of tensors whose matricization with respect to each $\mathcal{K} \in \mathcal{C}_{N^P}$ has rank at most $n_{\mathcal{K}}$. This set is also known in the literature as the set of tensors with *Hierarchical Tucker (HT) rank* at most $(n_{\mathcal{K}})_{\mathcal{K} \in \mathcal{C}_{N^P}}$ (*cf.* [25]). Accordingly, we denote it by:

$$HT\left((n_{\mathcal{K}})_{\mathcal{K} \in \mathcal{C}_{N^P}}\right) := \left\{\mathcal{V} \in \mathbb{R}^{D_1 \times \cdots \times D_{N^P}} : \forall \mathcal{K} \in \mathcal{C}_{N^P}, \operatorname{rank}(\llbracket \mathcal{V}; \mathcal{K} \rrbracket) \leq n_{\mathcal{K}}\right\}.$$

Now, define $(n_{\mathcal{K}}^*)_{\mathcal{K} \in \mathcal{C}_{N^P}} \in \mathbb{N}^{\mathcal{C}_{N^P}}$ by:

$$\forall \mathcal{K} \in \mathcal{C}_{N^P} : n_{\mathcal{K}}^* = \begin{cases} R & \text{if } \mu^{-1}(\mathcal{K}) \in \mathcal{C}_N^P \\ D_{\mathcal{K}} & \text{if } \mu^{-1}(\mathcal{K}) \notin \mathcal{C}_N^P \end{cases},$$

where $D_{\mathcal{K}} := \min\{\prod_{n \in \mathcal{K}} D_n, \prod_{n \in \mathcal{K}^c} D_n\}$. We show that for any tensor $\mathcal{A}$ that satisfies for all $\mathcal{K} \in \mathcal{C}_N^P$:

$$QE(\mathcal{A}; \mu(\mathcal{K})) \leq \frac{\epsilon^2}{(2N^P - 3)\|\mathcal{A}\|^2} \cdot \ln(R),$$

there exists a tensor $\mathcal{V} \in HT\left((n_{\mathcal{K}}^*)_{\mathcal{K} \in \mathcal{C}_{N^P}}\right)$ such that $\|\mathcal{A} - \mathcal{V}\| \leq \epsilon$. Consider, for each canonical partition $(\mathcal{K}, \mathcal{K}^c) \in \mathcal{C}_N^P$, the distribution

$$\mathcal{P}_{\mathcal{K}} = \left\{p_{\mathcal{K}}(i) := \frac{\sigma_{\mathcal{K},i}^2}{\|\mathcal{A}\|^2}\right\}_{i \in [D_{\mathcal{K}}]},$$

where $\sigma_{\mathcal{K},1} \geq \sigma_{\mathcal{K},2} \geq \ldots \geq \sigma_{\mathcal{K},D_{\mathcal{K}}}$ are the singular values of $\llbracket \mathcal{A}; \mu(\mathcal{K}) \rrbracket$ (note that $\frac{1}{\|\mathcal{A}\|^2} \sum_j \sigma_{\mathcal{K},j}^2 = \frac{\|\mathcal{A}\|^2}{\|\mathcal{A}\|^2} = 1$ so $\mathcal{P}_{\mathcal{K}}$ is indeed a probability distribution). Denoting by $H(\mathcal{P}_{\mathcal{K}}) := \mathbb{E}_{i \sim \mathcal{P}_{\mathcal{K}}}[\ln(1/p_{\mathcal{K}}(i))]$ the entropy of $\mathcal{P}_{\mathcal{K}}$, by assumption:

$$QE(\mathcal{A}; \mu(\mathcal{K})) = H(\mathcal{P}_{\mathcal{K}}) \leq \frac{\epsilon^2}{\|\mathcal{A}\|^2(2N^P - 3)} \ln(R),$$

for all $(\mathcal{K}, \mathcal{K}^c) \in \mathcal{C}_N^P$. Thus, taking $a = \frac{\epsilon^2}{\|\mathcal{A}\|^2 (2N^P - 3)}$ we get by Lemma 10 that there exists a subset $T_\mathcal{K} \subseteq [D_\mathcal{K}]$ such that

$$\mathcal{P}_\mathcal{K}(T_\mathcal{K}^c) \leq \frac{\epsilon^2}{(2N^P - 3)\|\mathcal{A}\|^2},$$

and $|T_\mathcal{K}| \leq e^{\frac{H(\mathcal{P}_\mathcal{K})}{a}} = e^{\ln(R)} = R$. Note that

$$\mathcal{P}_\mathcal{K}(T_\mathcal{K}) \leq \sum\nolimits_{i=1}^{R} \frac{\sigma_i^2}{\|\mathcal{A}\|^2}.$$

Since this holds for any subset of cardinality at most $R$. Taking complements we obtain

$$\sum\nolimits_{i=R+1}^{D_\mathcal{K}} \frac{\sigma_i^2}{\|\mathcal{A}\|^2} \leq \mathcal{P}_\mathcal{K}(T_\mathcal{K}^c),$$

so

$$\sqrt{\sum\nolimits_{i=R+1}^{D_\mathcal{K}} \sigma_{\mathcal{K},i}^2} \leq \frac{\epsilon}{\sqrt{(2N^P - 3)}}.$$

We can now invoke Lemma 7 (note that if $\mu^{-1}(\mathcal{K}) \notin \mathcal{C}_N^P$, then the requirements of Lemma 7 are trivially fullfiled with respect to the partition $(\mathcal{K}, \mathcal{K}^c)$ since $n_\mathcal{K}^* = D_\mathcal{K}$), which implies that there exists some $\mathcal{W} \in HT\big((n_\mathcal{K}^*)_{\mathcal{K} \in \mathcal{C}_{N^P}}\big)$ satisfying:

$$\|\mathcal{W} - \mathcal{A}\| \leq \epsilon,$$

as required.

The proof concludes by establishing that for any tensor $\mathcal{W} \in HT\big((n_\mathcal{K}^*)_{\mathcal{K} \in \mathcal{C}_{N^P}}\big)$, there exists assignment for the tensors constituting the $P$-dimensional locally connected tensor network (defined in Figure 6) such that it generates $\mathcal{W}$.

To see why this is the case, note that by Lemma 8 any tensor $\mathcal{W} \in HT\big((n_\mathcal{K}^*)_{\mathcal{K} \in \mathcal{C}_{N^P}}\big)$ can be represented by a (one-dimensional) locally connected tensor network with varying widths $(n_\mathcal{K}^*)_{\mathcal{K} \in \mathcal{C}_{N^P}}$, *i.e.* a tensor network conforming to a perfect binary tree graph in which the lengths of inner axes are as follows: an axis corresponding to an edge that connects a node with descendant leaves indexed by $\mathcal{K}$ to its parent is assigned the length $n_\mathcal{K}^*$. We can obtain an equivalent representation of any such tensor as a $P$-dimensional locally connected tensor network (described in Appendix J.1.1) via the following procedure. For each node at level $l \in \{0, P, 2P, \ldots, (L-1)P\}$ of the tree (recall $N = 2^L$ for $L \in \mathbb{N}$), contract it with its descendant nodes at levels $\{l+1, \ldots, l+(P-1)\}$.[12] This results in a new tensor network whose underlying graph is a perfect $2^P$-ary tree and the remaining edges all correspond to inner axes of lengths equal to $n_\mathcal{K}^* = R$ for $\mathcal{K} \in \mathcal{C}_N^P$, *i.e.* in a representation of $\mathcal{W}$ as a $P$-dimensional locally connected tensor network. $\qquad\square$

---

[12]For a concrete example, let $N = 2^L = 4$ and $P = 2$ (*i.e.* $L = 2$). In this case, the perfect binary tree underlying the one-dimensional locally connected tensor network of varying widths is of height $L \cdot P = 4$ and has $N^P = 16$ leaves. It is converted into a perfect 4-ary tree tensor network of height $L = 2$ by contracting the root with its two children and the nodes at level two with their children.

