# OpenReview forum: "What Makes Data Suitable for a Locally Connected Neural Network? A Necessary and Sufficient Condition Based on Quantum Entanglement."
_NeurIPS.cc/2023/Conference — NeurIPS 2023 spotlight_

### Official Review · Reviewer_AvgR · 2023-07-05

**Soundness:** 2 fair
**Presentation:** 3 good
**Contribution:** 3 good
**Rating:** 6
**Confidence:** 3

**Summary:**

By utilizing theoretical tools from quantum physics, the authors propose that a locally connected neural network can accurately predict data if and only if the data distribution exhibits low quantum entanglement under certain feature partitions. Based on this result, they develop a preprocessing method to enhance the compatibility between data distributions and locally connected neural networks. Experimental evaluations using various datasets and models validate their findings.

**Strengths:**

It investigates what makes a data distribution suitable for machine learning from the theories of tensor network and quantum entanglement, which provides a new perspective and tools from other fields to explore the learning theory.

**Weaknesses:**

The numerical experiments are insufficient, it is only applied to randomly arranged data instead of original data which might be more convincing to support the argument.

**Questions:**

1. In corollary 1(line 223), does it still work under other partitions? and what's the motivation for choosing "canonical partition"?
2. It demonstrates that the locally connected neural network is capable of accurate prediction over distribution if only if it admits low entanglements. In the experiments, it increases the entanglement of data via randomly swapping the feature and provides the comparison between a random permutation of features and proposed methods. Why the proposed method, in the numerics (Table 1, 2), is applied to random arrangement data instead of original data? After we manually randomly arrange the data feature, the data might be meaningless which is expected to get bad performance. Thus, is it appropriate to support the argument? If I miss some thing, please correct me.


**Limitations:**

---

> ### Author Rebuttal · Authors · 2023-08-08
>
> Thank you for your feedback. We respond to your comments and questions below. If our response is satisfactory, we would greatly appreciate it if you would consider raising your score.
>
> > *The numerical experiments are insufficient, it is only applied to randomly arranged data instead of original data which might be more convincing to support the argument.*
>
> > *It demonstrates that the locally connected neural network is capable of accurate prediction over distribution if only if it admits low entanglements. In the experiments, it increases the entanglement of data via randomly swapping the feature and provides the comparison between a random permutation of features and proposed methods. Why the proposed method, in the numerics (Table 1, 2), is applied to random arrangement data instead of original data? After we manually randomly arrange the data feature, the data might be meaningless which is expected to get bad performance. Thus, is it appropriate to support the argument? If I miss some thing, please correct me.*
>
> Our experimentation (Figures 3 and 8, and Tables 1 to 5) establishes the following:
> * Audio and image datasets, on which locally connected neural networks achieve high prediction accuracies, satisfy the necessary and sufficient condition we derive — low entanglement under canonical partitions.
> * Randomly permuting features in the above datasets leads the condition to be violated, i.e. the entanglement under canonical partitions to be higher, and accordingly prediction accuracies of locally connected neural networks deteriorate.
> * Applying our preprocessing algorithm (which is designed to ensure that the condition is met, i.e. that entanglement under canonical partitions is low) to the permuted datasets recovers a significant portion of the performance lost.
> * Applying our preprocessing algorithm to tabular datasets, on which the prediction accuracies of locally connected neural networks are known to be subpar, leads to significant gains in performance.
>
> It is important to note that in tabular data, features are by definition arranged arbitrarily, so randomly permuting them (as we did) prior to applying a non-permutation invariant learning architecture (e.g. a locally connected neural network) is the correct thing to do (otherwise, results may be skewed by implicit structure in the default feature arrangement, which is not supposed to exist).
>
> Note also that randomly permuting the features of a dataset does not render it meaningless. Indeed, learning architectures that are permutation invariant (for example fully connected neural networks) are completely unaffected by such permutation, and their prediction accuracies are often high, and in particular far better than chance.
>
> > *In corollary 1(line 223), does it still work under other partitions? and what's the motivation for choosing "canonical partition"?*
>
> The motivation for defining canonical partitions (Definition 2) is that, per our analysis (Corollary 1 in particular), low entanglement under these partitions characterizes the ability of a locally connected neural network to achieve high prediction accuracy. If you are asking about the intuition behind canonical partitions, note that, in general, the entanglement under a partition $( \mathcal{K} , \mathcal{J} )$ characterizes the dependence between $\mathcal{K}$ and $\mathcal{J}$. This, together with the fact that in canonical partitions $\mathcal{K}$ consists of contiguous indices, imply that low entanglement under canonical partitions can be viewed as a formalization of locality.
>
> With regards to extension of Corollary 1 to other partitions:
> It is oftentimes the case that entanglement is low under canonical partitions, while being high under other partitions. Accordingly, Corollary 1 will not hold true if one replaces the canonical partitions with an arbitrary set of partitions. Nonetheless, if the architecture of the locally connected neural network is modified, then in order for Corollary 1 to hold, the definition of canonical partitions (Definition 2) needs to be modified as well. In that sense, Corollary 1 can be extended to account for other partitions.
>
> We hope the above discussion adequately addresses your questions (please let us know if not and we will happily elaborate). It is planned to be included in the camera-ready version. Thank you!

---

> > ### Comment · Reviewer_AvgR · 2023-08-20
> >
> > Thank you for your response that helps me better understand, I have no further questions.

---

### Official Review · Reviewer_tJGs · 2023-07-18

**Soundness:** 4 excellent
**Presentation:** 3 good
**Contribution:** 4 excellent
**Rating:** 7
**Confidence:** 3

**Summary:**

The paper investigates criterion that make data distributions suitable for being accurately fit by neural networks using tools from tensor networks. Specifically, it shows that some locally connected neural networks (with polynomial activations) fit a data distribution accurately if and only if the quantum entanglement of the data tensor is sufficiently low under all canonical partitions of the axes. The argument uses the previously established fact that a certain tensor network (locally connected tensor network) is equivalent to a locally connected neural network. The main theoretical contribution is to show that low quantum entanglement across canonical partitions is a necessary and sufficient condition for an arbitrary tensor $T$ to be approximated by the locally connected tensor network $\mathcal{W}_\mathrm{TN}$. The problem of a locally connected neural network fitting a distribution is then identified with the corresponding data tensor being well-approximated by $\mathcal{W}_\mathrm{TN}$, to establish the main claim.

Numerical simulations are performed to show that under random swaps of the features of standard datasets (that are known to be well fit by local neural networks), more swaps results in more entanglement and this corresponds to the predictive performance being correspondingly lower. The authors also suggest a heuristic method for improving the compatibility of data to a local neural network architecture, by searching for permutations of the axes that result in a low entanglement.

**Strengths:**

The main result is compelling, providing a clean and computationally efficient measure that may be used to quantify the suitability of a data distribution to certain local neural network architectures. Despite the minor limitations (such as polynomial activations) placed on the corresponding network, this still seems to be significant progress on an important and difficult question.  The potential of systematically searching for feature permutations to improve the predictive power of locally connected networks is also an intriguing possibility.

The theoretical results seem sound and correct to the best of my knowledge, although I did not check every detail.

The numerical results do provide evidence of an increase in predictive power from searching for low entanglement feature permutations.

**Weaknesses:**

1. The presentation of the paper is rather dense in some places, which is understandable due to the breadth of material covered, but makes the exposition hard to follow in places. It would be helpful to have some intuition about how the notions defined in the paper correspond to standard ones in deep learning. For example, the definition of canonical partitions seems unmotivated at first and it may be helpful to the reader if the connection to locality is highlighted.

As a minor comment, there are a few terms that are defined once and then used in later statements without context: for example, it would be helpful to mention in Definition 2 that $N = 2^L$, and to have a reminder about $R$ near Corollary 1.

2. Section 5.2.2 only presents numerical results for the proposed preprocessing pipeline against a baseline of randomly permuted features. It is not clear to me whether these improvements manifest for real data ie. data already suited to a locally connected network, with the goal being to improve predictive accuracy. It is unlikely in practice that a preprocessing step would be used to improve the performance of a model with very low predictive accuracy.

**Questions:**

Does the suggested pipeline lead to any accuracy improvements for natural datasets (without randomly permuted features) and local networks that already achieve good performance on them?

**Limitations:**

Yes

---

> ### Author Rebuttal · Authors · 2023-08-08
>
> Thank you for your positive feedback, and in particular for describing our contributions as “significant progress on an important and difficult question”! We respond to your comments and questions below.
>
> > *The presentation of the paper is rather dense in some places, which is understandable due to the breadth of material covered, but makes the exposition hard to follow in places. It would be helpful to have some intuition about how the notions defined in the paper correspond to standard ones in deep learning. For example, the definition of canonical partitions seems unmotivated at first and it may be helpful to the reader if the connection to locality is highlighted.*
>
> Thank you for raising this point! In line with your suggestion, we plan to use additional space in the camera-ready version for discussions providing more intuition behind the notions we use and their relation to deep learning. Below is an initial discussion about canonical partitions.
>
> The motivation for defining canonical partitions (Definition 2) is that, per Theorem 1, low entanglement under these partitions characterizes the ability of the locally connected tensor network to fit a given tensor. In general, the entanglement under a partition $( \mathcal{K} , \mathcal{J} )$ characterizes the dependence between $\mathcal{K}$ and $\mathcal{J}$. This, together with the fact that in canonical partitions $\mathcal{K}$ consists of contiguous indices, imply that low entanglement under canonical partitions can indeed be viewed as a formalization of locality.
>
> > *As a minor comment, there are a few terms that are defined once and then used in later statements without context: for example, it would be helpful to mention in Definition 2 that
> \(N=2^{L}\), and to have a reminder about R near Corollary 1.*
>
> Thank you for the suggestions! They will be incorporated into the text.
>
> > *Section 5.2.2 only presents numerical results for the proposed preprocessing pipeline against a baseline of randomly permuted features. It is not clear to me whether these improvements manifest for real data ie. data already suited to a locally connected network, with the goal being to improve predictive accuracy. It is unlikely in practice that a preprocessing step would be used to improve the performance of a model with very low predictive accuracy.*
>
> > *Does the suggested pipeline lead to any accuracy improvements for natural datasets (without randomly permuted features) and local networks that already achieve good performance on them?*
>
> The preprocessing algorithm we propose (in Section 5.1) is not designed for data on which locally connected neural networks already achieve high prediction accuracies. Indeed, we have shown (in Figures 3 and 8) that on such data, namely on audio and image datasets, the criterion sought after by the algorithm — low entanglement under canonical partitions — is satisfied to begin with. In contrast, on tabular data (which may also be real data) the prediction accuracies of locally connected neural networks are known to be subpar (see lines 328-329 of the paper), and our preprocessing algorithm may lead to significant improvements. We demonstrated this in Section 5.2.2 with standard datasets. Note that in tabular data features are by definition arranged arbitrarily, so randomly permuting them (as we did) prior to applying a non-permutation invariant learning architecture (e.g. a locally connected neural network) is the correct thing to do (otherwise, results may be skewed by implicit structure in the default feature arrangement, which is not supposed to exist).

---

### Official Review · Reviewer_jhrZ · 2023-07-26

**Soundness:** 2 fair
**Presentation:** 3 good
**Contribution:** 3 good
**Rating:** 6
**Confidence:** 3

**Summary:**

This paper focuses on the problem that which data distribution is more learnable by locally-connected neural networks such as CNN, RNN, and local-attention. The paper introduces the notation of quantum entanglement, theoretically proves and empirically verifies that the network can achieve accurate predictions if and only if the data distribution exhibits low quantum entanglement under canonical partitions of features.

**Strengths:**

1. This paper introduces the notation of quantum entanglement from physics, which provides new insights into the problem of which data distribution is more learnable by a specific family of neural networks.

2. The paper provides a comprehensive theoretical analysis of the above problem.

3. Overall speaking, the paper is easy to read and follow.


**Weaknesses:**

1. The notion of entanglement is not very intuitive. Although the authors formally introduce the mathematical definition of quantum entanglement, it is still not clear what characteristics a data distribution would exhibit if the distribution has low entanglement. I suggest the authors illustrate this low-entanglement property on toy datasets and/or textual data. This intuitive illustration might provide valuable insights to the machine learning community.

2. The authors are encouraged to conduct comprehensive experiments on more architectures and datasets. For example, for image classification, it is suggested to test on classical CNN architectures such as AlexNet, VGG, and ResNet. Besides, textual data should also be investigated.


**Questions:**

1. In Theorem 1, the authors derive an upper bound for quantum entanglement. However, it is not clear whether this upper bound truly ensures *low* entanglement. There are mainly two terms in the bound, $\ln(R)$ and $\ln(D_\mathcal{K})$. The magnitude of the first term is already addressed by the authors, but it is not clear whether the second term can be extremely large. The authors are encouraged to have a discussion on this issue and show that this upper bound is indeed much smaller than the entanglement of a random tensor (as a baseline).

2. How will the network architecture affect the main theoretical results in Section 3.1, 4.1, and 4.2? For example, I wonder if the results still hold if the neural network has skip connections and batch normalization operations.

3. What does feature (re)arrangement mean in Section 5.1? Let us take textual data as an example and consider a sentence with N words. Does a feature rearrangement mean that the words (as well as their embeddings) in the sentence are randomly permuted? The authors are suggested to make this clearer, and better with examples.


**Limitations:**

Yes

---

> ### Author Rebuttal · Authors · 2023-08-08
>
> Thank you for your feedback, and specifically for noting the soundness of our theory and the clarity of our presentation. We respond to your comments and questions below. If our response is satisfactory, we would greatly appreciate it if you would consider raising your score.
>
> > *The notion of entanglement is not very intuitive. … it is still not clear what characteristics a data distribution would exhibit if the distribution has low entanglement. I suggest the authors illustrate this low-entanglement property … . This intuitive illustration might provide valuable insights … .*
>
> Thank you for raising this! In line with your suggestion, we will add to the text more intuition behind entanglement, in general and in the context of data distributions. Below is an initial discussion along this line.
>
> Generally, entanglement is a concept which, given a collection of elements $[N] := \\{ 1, 2, … , N \\}$ partitioned into two sets $\mathcal{K}$ and $\mathcal{J} := [N] \setminus K$, quantifies how dependent $\mathcal{K}$ and $\mathcal{J}$ are. In quantum physics the elements in $[N]$ are particles, and the entanglement quantifies the dependence (quantum interaction) between the particles in $\mathcal{K}$ and those in $\mathcal{J}$. In the context of data distributions, the elements in $[N]$ are features (e.g. pixels, text tokens or audio samples), and the entanglement is a measure of dependence between the features in $\mathcal{K}$ and those in $\mathcal{J}$.
>
> To gain some intuition on entanglement as a measure of dependence between the features in $\mathcal{K}$ and those in $\mathcal{J}$, consider the case of entanglement equal to zero. There, the population data tensor (Eq. (3)) can be written as an outer product between two tensors, one corresponding to $\mathcal{K}$ and the other to $\mathcal{J}$. This may be interpreted as zero correlation between the features in $\mathcal{K}$ and those in $\mathcal{J}$ (indeed, by Eq. (3), the population data tensor holds expectations of products of features, and the expectation of a product being equal to a product of expectations is the definition of zero correlation). Moving to the case where entanglement is non-zero, one may view it as a measure of distance from zero correlation — the higher the entanglement, the farther we are from this state, and vice versa.
>
> > *The authors are encouraged to conduct comprehensive experiments on more architectures and datasets ... .*
>
> As stated in the paper (experimental sections and Appendix J), our experiments currently include the following architectures:
>
> - M5 and ResNet CNNs;
> - S4 RNN; and
> - local self-attention,
>
> and the following datasets:
>
> - Speech Commands audio;
> - CIFAR10 images; and
> - semeion, isolet and dna tabular benchmarks.
>
> To our knowledge, for a paper whose nature is primarily theoretical, this empirical evaluation is relatively broad.
>
> Notwithstanding the above, following your comment, we began conducting additional experiments with more architectures and more datasets (including textual data). These experiments will take several weeks to run on our hardware. We did however manage to obtain complete results for VGG CNN architecture, and these are qualitatively identical to the CNN results reported in the paper. We will include a full account of the additional experiments in the text once they are complete.
>
> > *In Theorem 1 … it is not clear whether this upper bound truly ensures low entanglement. There are mainly two terms in the bound, $\ln (R)$  and $\ln (D_{\mathcal{K}})$. The magnitude of the first term is already addressed by the authors, but it is not clear whether the second term can be extremely large. The authors are encouraged to have a discussion on this issue and show that this upper bound is indeed much smaller than the entanglement of a random tensor … .*
>
> Note that in Theorem 1, the term $\ln ( D_{\mathcal{K}} )$ (which up to a logarithmic factor is on the order of the number of axes $N$) is **multiplied by $\epsilon$**, the desired degree of approximation. Accordingly, in the regime of interest (low $\epsilon$), the upper bound is indeed small. In contrast, the theorem shows that there exist tensors for which entanglements are on the order of $\ln ( D_{\mathcal{K}} )$. This in fact holds for random tensors as well; a point that will be clarified in the text. Thank you for the question!
>
> > *How will the network architecture affect the main theoretical results … ? For example, I wonder if the results still hold if the neural network has skip connections and batch normalization … .*
>
> It is possible to extend our analysis to tensor networks, and corresponding neural networks, with connectivities beyond those considered (e.g. connectivities that are non-local, or ones involving skip connections). Such extensions can follow the lines of Appendix I (which lifts the one-dimensional analysis in paper body to arbitrary dimensions). In particular, under such extensions, our theoretical results will still hold, but the definition of canonical partitions (Def. 2) will require adaptation. With regards to batch normalization, as long as it maintains representational capacity (as is typically the case), it will be automatically accounted for by our theory. We will add to the text a discussion including the above, as well as architectural extensions that require further research. Thank you for the question!
>
> > *What does feature (re)arrangement mean in Section 5.1? Let us take textual data as an example … . Does a feature rearrangement mean that the words … are randomly permuted? … .*
>
> In our context, feature (re)arrangement means applying a permutation to the features in each data instance (e.g. the word embeddings in each sentence, the pixels in each image, or the samples in each audio recording). Note that the permutation need not be random. In fact, Section 5.1 concerns searching for a specific permutation which satisfies a specific property (low entanglement under canonical partitions).

---

> > ### Comment · Reviewer_jhrZ · 2023-08-16
> >
> > The authors' rebuttal addresses most of my concerns. I would like to keep my original rating toward accepting this paper.

---

### Official Review · Reviewer_CJsr · 2023-07-26

**Soundness:** 3 good
**Presentation:** 3 good
**Contribution:** 4 excellent
**Rating:** 8
**Confidence:** 3

**Summary:**

This paper investigates the representation power of locally connected neural networks, a prevalent family of deep learning architectures, using tools from quantum physics. In particular, following the established equivalence between locally connected neural network and locally connected tensor network, the authors showed that the necessary and sufficient condition of a locally connected neural network fitting an objective tensor is the entanglements of the tensor admits under canonical partitions being small enough. As an application, the authors discuss the condition of making accurate predictions using locally connected neural networks, accompanied by empirical demonstrations. Notably, the findings suggest that the representation power of locally connected neural networks can potentially be enhanced by reorganizing the features to achieve reduced entanglements under canonical partitions.

**Strengths:**

The authors successfully extend the connection between deep learning and quantum physics by revealing a compelling link between representation power and entanglement – two fundamental concepts in both domains. From my perspective, the results are not only theoretically elegant but also closely related to illustrations of practical situations.

**Weaknesses:**

The paper's presentation could benefit from further improvement, especially in providing more qualitative discussions to enhance readers' intuition regarding the technical correspondence between representation power and entanglement. While the link between these two concepts is established, the paper lacks sufficient qualitative explanations to make the connection more accessible.

Furthermore, the tensor network's width, denoted as $R$ in the paper, seems to be crucial in the derived bounds. However, the definition of this parameter remains somewhat unclear, so I'm not fully convinced on why it is usually small.

Furthermore, the current results solely focus on the representation of tensors using locally connected neural networks. It would be good to include an explanation as to why other types of underlying functions for representation were not considered.

If these questions are addressed, I'm happy to further increase my evaluation.

**Questions:**

I don't have more questions other than the existing ones above.

---

> ### Author Rebuttal · Authors · 2023-08-08
>
> Thank you for your positive feedback and support! We greatly appreciate your willingness to further increase your evaluation if your questions are addressed. We treat them below.
>
> > *The paper's presentation could benefit from further improvement, especially in providing more qualitative discussions to enhance readers' intuition regarding the technical correspondence between representation power and entanglement. While the link between these two concepts is established, the paper lacks sufficient qualitative explanations to make the connection more accessible.*
>
> Thank you for raising this point! In line with your suggestion, we plan to use additional space in the camera-ready version for qualitative discussions aimed to enhance readers’ intuition. Below is an initial discussion on the technical correspondence between representation power and entanglement.
>
> Roughly speaking, entanglement is a concept which, given a collection of elements $[N] := \\{ 1, 2, … , N \\}$ partitioned into two sets $\\mathcal{K}$ and $\\mathcal{J} := [N] \\setminus K$, quantifies how dependent $\\mathcal{K}$ and $\\mathcal{J}$ are. In quantum physics the elements in $[N]$ are particles, and the entanglement quantifies the dependence, i.e. the quantum interaction, between the particles in $\\mathcal{K}$ and those in $\\mathcal{J}$. In the context of neural networks, the elements in $[N]$ are input variables (e.g. pixels, text tokens or audio samples), and the entanglement quantifies the dependence that a neural network can represent between the variables in $\\mathcal{K}$ and those in $\\mathcal{J}$.
>
> To gain some intuition on the latter (entanglement quantifying the dependence that a neural network can represent between the variables in $\\mathcal{K}$ and those in $\\mathcal{J}$), consider the case of entanglement being equal to zero. There, a function $f ( \\cdot )$ realized by the neural network must be separable with respect to $\\mathcal{K}$ and $\\mathcal{J}$, meaning it can be written as $f ( X ) = g ( X_\\mathcal{K} ) h ( X_\\mathcal{J} )$, i.e. as a product of two functions, one that intakes only variables in $\\mathcal{K}$, and another that intakes only variables in $\\mathcal{J}$. This means that there is no dependence between $\\mathcal{K}$ and $\\mathcal{J}$. Indeed, in a statistical setting, where $f ( \\cdot )$, $g ( \\cdot )$ and $h ( \\cdot )$ are probability density functions, this is the definition of $\\mathcal{K}$ and $\\mathcal{J}$ being statistically independent. Moving to the general case (where the entanglement is not necessarily zero), one may view the entanglement as the distance from the above-described independence, i.e. the distance $f ( \\cdot )$ can have from the closest function which is separable with respect to $\\mathcal{K}$ and $\\mathcal{J}$. The higher the entanglement, the more dependence can be represented, and vice versa.
>
> > *Furthermore, the tensor network's width, denoted as R  in the paper, seems to be crucial in the derived bounds. However, the definition of this parameter remains somewhat unclear, so I'm not fully convinced on why it is usually small.*
>
> The width of the tensor network $R$ corresponds to the width of hidden layers in the equivalent neural network, thus in practice (i.e. in any situation where the neural network is to be implemented) $R$ must be of moderate size (typically no more than a few hundreds or thousands), and in particular $\ln ( R )$ is much smaller than the number of input elements $N$. This is discussed in lines 124-127 of the paper. We will broaden that portion of the text to further clarify. Thank you for bringing this up!
>
> > *Furthermore, the current results solely focus on the representation of tensors using locally connected neural networks. It would be good to include an explanation as to why other types of underlying functions for representation were not considered.*
>
> In Section 4.1 we show that, for the analyzed model (locally connected neural network) over a standard SVM objective, accurate prediction is equivalent to fitting (representing) a tensor. Extending this equivalence to other types of objectives, e.g. multi-class SVM, is viewed as an interesting direction for future work. We will mention this in the conclusion; thank you!
>
> *P.S.*
> $\\,$ If we misunderstood the intent behind “other types of underlying functions for representation” please let us know and we will gladly elaborate.

---

> > ### Comment · Reviewer_CJsr · 2023-08-15
> >
> > I would like to thank the authors for addressing the questions I have raised in a satisfactory manner. Accordingly, I have decided to raise my rating to 8.

---

### Official Review · Reviewer_SJKu · 2023-07-27

**Soundness:** 3 good
**Presentation:** 2 fair
**Contribution:** 3 good
**Rating:** 6
**Confidence:** 3

**Summary:**

The fundamental question of what makes a data distribution suitable for deep learning is addressed in this study, focusing on locally connected neural networks. The study uses theoretical tools from quantum physics to tackle this problem. The main theoretical finding is that a specific type of locally connected neural network can accurately predict over a data distribution if, and only if, the data distribution shows low quantum entanglement under specific feature partitions. The study suggests that using quantum entanglement could inspire further use of physics tools to understand the relationship between deep learning and real-world data.

**Strengths:**

- The paper technically sounds correct and claims well supported by theoretical analysis and experimental results.
- Related works and background knowledge are covered and discussed.
- Experiments are conducted extensively in different locally connected neural networks (CNN, RNN, local self-attention), and experimental results are thoroughly discussed.

**Weaknesses:**

- Need more discussion on the motivation of using quantum entanglement.
- Need more discussion on the limitations of this study.

**Questions:**

- The whole study targtes at locally connected neural networks. What happens to a NN with high connectivity? Does entanglement help the learning task?
- Entanglement entropy is a typical metric. But what does entanglement mean in the dataset? If it is used to quantify the temporal/spatial non-locality in the data, why not use other metrics such as autocorrelation or other metrics from information theory? Why choose entanglement specifically?
- Why use a tensor network as the equivalent model instead of directly evaluating on locally connected neural networks?
- Will the code be available to reproduce the findings?

**Limitations:**

Not found.

---

> ### Author Rebuttal · Authors · 2023-08-08
>
> Thank you for your feedback, and specifically for noting the soundness of our theory and experiments, as well as our account for background and related work. We respond to your comments and questions below. If our response is satisfactory, we would greatly appreciate it if you would consider raising your score.
>
> > *Need more discussion on the motivation of using quantum entanglement.*
>
> The motivation for using quantum entanglement is discussed in the introduction (Section 1). As stated there, quantum entanglement facilitates a widely accepted theory that allows for assessing the suitability of a distribution to a computational model, where distributions are described by tensors and the computational models are tensor networks. This, along with a known equivalence between tensor networks and certain deep neural networks, allows us to make progress on the question of what makes a (data) distribution suitable for deep learning.
>
> We hope the above addresses your comment. If not, and the intent behind “motivation of using quantum entanglement” is for an intuition behind its connection to neural networks, then please see our response to the first point raised by Reviewer CJsr.
>
> > *Need more discussion on the limitations of this study.*
>
> Limitations are currently discussed throughout the paper (for example, the fact that, as you mention, our theory is restricted to a specific type of locally connected neural network, is explicitly conveyed in Section 3.1). Following your comment, we plan to use additional space in the camera-ready version for a concentrated account of limitations.
>
> > *The whole study targtes at locally connected neural networks. What happens to a NN with high connectivity? Does entanglement help the learning task?*
>
> Our study is indeed limited to locally connected neural networks, which, while prevalent in practice, do not account for all deep learning architectures being used. When adding connectivity to a locally connected neural network its representational capacity is generally enhanced, so the condition we derived (which is both necessary and sufficient before connectivity is added) remains sufficient. We do not expect it to remain necessary though. Investigation of necessary conditions for a data distribution to be suitable to a neural network with high connectivity is a promising direction for future research; we hope that our work will inspire such progress. We will mention the above in the camera-ready version of the paper; thank you!
>
> > *Entanglement entropy is a typical metric. But what does entanglement mean in the dataset? If it is used to quantify the temporal/spatial non-locality in the data, why not use other metrics such as autocorrelation or other metrics from information theory? Why choose entanglement specifically?*
>
> In general, entanglement is a concept which, given a collection of elements $[N] := \\{ 1, 2, … , N \\}$ partitioned into two sets $\\mathcal{K}$ and $\\mathcal{J}:= [N] \\setminus K$, quantifies how dependent $\\mathcal{K}$ and $\mathcal{J}$ are. When the elements $[N]$ are features of a dataset ordered by time/space, and the partition is such that $\\mathcal{K}$ is contiguous (this is the case for all canonical partitions), then the entanglement can indeed be viewed as quantifying the temporal/spatial non-locality in the data.
>
> The reason we use entanglement and not other measures as you suggest is laid out in our response to your first comment. Namely, entanglement facilitates a widely accepted theory in quantum physics, which allows us to make progress on the question of what makes a data distribution suitable for deep learning.
>
> > *Why use a tensor network as the equivalent model instead of directly evaluating on locally connected neural networks?*
>
> As discussed in the introduction (Section 1), tensor networks tie to a widely accepted theory from quantum physics, which we imported for addressing our question of study (what makes a data distribution suitable for deep learning). Although it is possible to present our analysis solely through the lens of locally connected neural networks, we chose to include the tensor network formalism since it reflects the connection to physics — a branch of science we believe will be key to formally reasoning about the relation between deep learning and real-world data.
>
> We note that tensor networks were central to many past studies of expressiveness and generalization in deep learning. See lines 128-138 of the paper for further details.
>
> > *Will the code be available to reproduce the findings?*
>
> Code for reproducing our experiments is available in the supplementary material. A public repository holding this code will be referenced in the camera-ready version of the text.

---

> > ### Comment · Reviewer_SJKu · 2023-08-16
> >
> > The authors' rebuttal addresses my concerns. I would like to raise my rating to 6.

---

### Decision · Program_Chairs · 2023-09-21

**Decision:**

Accept (spotlight)

**Comment:**

The paper investigates criteria that make data distributions suitable for being accurately fit by neural networks using tools from tensor networks. The major technical contribution is the establishment of the connection between locally connected neural networks and locally connected tensor networks. Based on that, the authors identified the conditions for whether a locally connected network fitting an objective tensor could be derived from the entanglement (a concept in quantum physics) when applied to the corresponding tensor network.  As a result, the paper provides a mathematically elegant and computationally efficient measure, inspired by quantum physics, to characterize the suitability of a data distribution to certain local neural network architectures.  There is a consensus that this is significant progress on an important and difficult problem, with many potentials yet unexplored, which justifies the recommendation.

However, there is also a consensus that the paper would benefit from a better presentation of the key ideas. Especially, some conceptual discussions about quantum entanglement, and about the built connection between the expressive power of neural networks and the entanglement, would greatly improve the accessibility of the contribution.  The authors are also advised to include more comprehensive empirical results, as well as other changes,  as promised in the rebuttal.